# Time-variability and uncertainty of the fraction of young water in a small headwater catchment

Michael Paul Stockinger[1,2], Heye Reemt Bogena[1], Andreas Lücke[1], Christine Stumpp[2], Harry Vereecken[1]

[1]Forschungszentrum Jülich GmbH, Agrosphere Institute (IBG-3), Wilhelm-Johnen-Straße, 52425 Jülich, Germany
[2]University of Natural Resources and Life Sciences Vienna, Institute of Hydraulics and Rural Water Management, Muthgasse 18, 1190 Vienna, Austria.

*Correspondence to*: Michael Paul Stockinger (michael_stockinger@boku.ac.at)

**Abstract.** The time precipitation needs to travel through a catchment to its outlet is an important descriptor of a catchment's susceptibility to pollutant contamination, nutrient loss and hydrological functioning. The fast component of total water flow
can be estimated by the fraction of young water (Fyw) which is the percentage of streamflow younger than three months. Fyw is calculated by comparing the amplitudes of sine waves fitted to seasonal precipitation and streamflow tracer signals. This is usually done for the complete tracer time series available neglecting annual differences in the amplitudes of longer time series. Considering inter-annual amplitude differences, we employed a moving time window of one-year length in weekly time steps over a 4.5-years $\delta^{18}$O tracer time series to calculate 189 Fyw estimates and their uncertainty. They were then tested against the
following null hypotheses, defining a difference of 0.04 in Fyw (4% young water) as significant based on data-inherent uncertainty: (1) At least 90% of Fyw results do not deviate more than ±0.04 from the mean of all Fyw results indicating long-term invariance. Larger deviations would indicate changes in the relative contribution of different flow paths; (2) for any four-week window Fyw does not change more than ±0.04 indicating short-term invariance. Larger deviations would indicate a high sensitivity of Fyw to a 1-4 weeks shift in the start of a one-year sampling campaign; (3) for a given calendar month Fyw does
not change more than ±0.04 indicating seasonal invariance of Fyw. In our study, all three null hypotheses were rejected. Thus, the Fyw results were time-variable, showed variability in the chosen sampling time and had no pronounced seasonality. We furthermore found evidence that the 2015 European heat wave and including two winters into a one-year sampling campaign increased the uncertainty of Fyw calculation. Based on an increase of Fyw uncertainty when the mean adjusted R² was below 0.2 we recommend further investigations into the dependence of Fyw and its uncertainty to goodness-of-fit measures.
Furthermore, while investigated individual meteorological factors did not sufficiently explain variations of Fyw, the runoff coefficient showed a moderate negative correlation of r = -0.50 with Fyw. The results of this study suggest that care must be taken when comparing Fyw of catchments that were based on different calculation periods and that the influence of extreme events and snow must be considered.

## 1 Introduction

Precipitation water uses slow and fast flow paths on its way through a catchment to the outlet where it becomes streamwater [*Tsuboyama et al.*, 1994]. Slow flow paths are for example the saturated and unsaturated flow through the soil matrix [*Gannon et al.*, 2017] while fast flow paths include preferential flow [*Wiekenkamp et al.*, 2016a] and overland flow [*Miyata et al.*, 2009].

The distribution of slow and fast flow paths varies in time and depends on a catchment's spatiotemporal characteristics [*Harman*, 2015; *Heidbüchel et al.*, 2013; *Stockinger et al.*, 2014; *Tetzlaff et al.*, 2009a; *Tetzlaff et al.*, 2009b]. Knowledge of this distribution helps in assessing the risk of streamflow contamination with pollutants or nutrient loss since nutrients and pollutants are transported through the soil by hydrological pathways [*Bourgault et al.*, 2017; *Gottselig et al.*, 2014].

The water stable isotopes ($\delta^{18}$O and $\delta^2$H) are widely applied in the study of flow paths and transit times of precipitation through a catchment [*McGuire and McDonnell, 2006*]. One method that utilizes the water stable isotopes for investigating fast flow paths is the fraction of young water (Fyw). Developed by *Kirchner* [2016a], Fyw estimates the streamflow fraction that is younger than three months since entering the catchment as meteoric water. It does so by comparing the amplitudes of sine waves fitted to the seasonally-varying isotope tracer signals of precipitation and streamflow. The seasonally-varying isotope

signal in precipitation is caused by different evaporation/condensation temperatures, vapor source areas and evaporation amounts of falling rain droplets during warmer and colder seasons, leading on average to higher $\delta^{18}$O values in summer and lower ones in winter [*Dansgaard*, 1964]. As rainfall passes through a catchment to reach the outlet, this signal is attenuated and shifted in time, leading to a much smoother but still seasonally-varying isotope signal in streamflow. The ratio of the fitted streamflow sine wave's amplitude $A_S$ divided by the fitted precipitation sine wave's amplitude $A_P$ equals the percentage of

water in streamflow younger than three months. *Kirchner* [2016a,b] showed the robustness of Fyw against spatial catchment heterogeneities (aggregation bias error) where previous methods of transit time estimation by sine wave fitting produced highly uncertain results.

Catchment influences on Fyw were, e.g., investigated globally by *Jasechko et al.* [2016]. They calculated Fyw for 254

catchments and concluded that one third of global streamflow consists of water younger than three months with catchments in steeper terrains having smaller contributions of young water to their runoff. *Wilusz et al.* [2017] coupled a rainfall generator with rainfall-runoff and time-varying transit time models to determine the young water fraction. They found an increase of annual rainfall amounts of 1 mm/d led to an increase of 0.03-0.04 in the modeled Fyw (percentage point increase of 3-4%, from here on written as 0.03-0.04, where the value 1 would mean that 100% of streamflow is younger than three months).

Similar to this, *von Freyberg et al.* [2018] found a positive correlation between Fyw and high-intensity precipitation events. This dependence of Fyw on precipitation characteristics could lead to long-term changes in Fyw due to global warming. Global warming was found to increase precipitation intensity and the frequency of droughts [*Pendergrass and Hartmann*, 2014; *Trenberth*, 2011]. For Europe, the chance of extreme heat waves and thus dry conditions has substantially increased since 2003

[*Christidis et al*., 2015]. Previous studies highlighted that the distribution of fast and slow flow paths is time-variable [*Harman*, 2015; *Heidbüchel et al*., 2013]. Since Fyw focuses on fast flow paths we expect it to be variable in time as well. However, so far previous studies focused on comparing Fyw between different catchments to derive relationships between catchment characteristics and Fyw, but no study investigated the temporal variability of Fyw for a given catchment yet.

Besides catchment characteristics, the conditions and conceptualizations of the Fyw calculation also influenced results in past studies. The effect of varying sampling frequencies of tracer data was investigated by *Stockinger et al*. [2016]. A higher sampling frequency led to higher Fyw highlighting the sensitivity of Fyw to the temporal resolution of the available tracer data. *Lutz et al*. [2018] investigated 24 catchments in Germany and used 10,000 Monte Carlo simulations with random errors

10   in the isotope data of precipitation and streamflow to derive the 95% confidence intervals of Fyw. Their confidence intervals indicated a robustness of Fyw against random errors in input data. The study of *von Freyberg et al*. [2018] focused on three influences on Fyw: (a) spatially interpolating precipitation isotopes, (b) including snow pack and (c) weighing streamflow in fitting sine waves. They found that weighing streamflow led to significant changes in Fyw while the other factors had a negligible effect.

The mentioned studies highlight the current research interest in the new measure of Fyw. For this reason, it is necessary to investigate the sensitivity of Fyw and its uncertainty to different datasets. This is especially important for catchment comparison studies where the conceptualization of calculating Fyw might vary between catchments or datasets of different catchments may vary in quality. The question to answer is how much of the difference between individual Fyw estimates stems

from actual, catchment-borne differences in flow path distributions and which part is merely based on e.g., different data quality or quantity.

The present study aims at answering one aspect of this open research question by focusing on the time-variance of Fyw and its associated uncertainty. Past studies fitted one sine wave to the complete time series available, varying from less than a year

to several decades [*Ogrinc et al*., 2008; *Song et al*., 2017; *von Freyberg et al*., 2018]. To our knowledge, only the study of *Stockinger et al*. [2017] calculated Fyw for two different 1-year periods of a multi-year time series but did not test the temporal variability of Fyw nor influencing factors on it or its uncertainty. Thus, the sensitivity of the Fyw method towards the timing and the length of the available data remains to be tested in detail. The present study investigated the temporal variability of Fyw when different calculation periods of a multi-year isotope data set are used. We used a one-year time window which was

shifted in 7-days steps to calculate 189 Fyw estimates over a 4.5-year time series of isotope data. The 189 Fyw results were tested against the following null hypotheses:

(1)  Fyw estimates do not change over time (time-invariance)
(2)  Short-term changes in the start of a tracer sampling campaign do not influence the Fyw estimate (sampling-invariance)

(3) Fyw estimates are similar for calculation years that are centered around a given calendar month (seasonal-invariance)

The three hypotheses were tested against whether Fyw differences exceeded a threshold value of ±0.04 which is the Fyw uncertainty when fitting a single sine wave to the 4.5-years time series (data-inherent uncertainty derived by Gauß error propagation, see results). We used hydrometeorological and isotopic data to investigate possible influences on time-variable Fyw results and their associated uncertainties and, where applicable, to reduce uncertainty. In conclusion of this study we recommend a tracer sampling design that reduces Fyw uncertainty.

## 2 Material and methods

### 2.1 Study site

The Wüstebach headwater catchment (38.5 ha) is located in the Eifel National Park (Germany, Figure 1). It is also part of the Lower Rhine/Eifel Observatory of the Terrestrial Environmental Observatories (TERENO) network [*Bogena et al.*, 2018]. The mean annual precipitation amounts to 1107 mm (1961 – 1990) with a mean annual temperature of 7°C [*Zacharias et al.*, 2011]. Soils are up to 2 m deep with an average depth of 1.6 m [*Graf et al.*, 2014]. Soil types of cambisol and planosol/cambisol are found on hillslopes, whereas gleysols, histosols and planosols are found in the riparian zone. The catchment is mostly covered with Norway spruce (Picea abies) and Sitka spruce (Picea sitchensis) [*Etmann*, 2009]. Eight ha (~21%) of the forest were clear-cut in August/September 2013 [*Wiekenkamp et al.*, 2016b]. A severe heat wave occurred in the Wüstebach during summer 2015 [*Duchez et al.*, 2016].

### 2.2 Data preparation

We used hourly hydrometric and weekly $\delta^{18}O$ isotope data of precipitation (composite sample) and streamflow (grab sample) from October 2012 to June 2017. We did not use $\delta^2H$ due to the strong correlation of $\delta^{18}O$ and $\delta^2H$ ($R^2$ = 0.97 for throughfall and 0.87 for streamflow) and therefor redundancy of information content. Precipitation depths were measured hourly in 0.1 mm increments for rainfall and daily in 1 cm increments for snowfall at the meteorological station Monschau-Kalterherberg of the German Weather Service (Deutscher Wetterdienst DWD station 3339, 535 m asl), located 9 km northwest of the catchment. Runoff was measured at the outlet by a V-notch weir for lower and a Parshall flume for higher runoff depths in 10-minute intervals. We collected throughfall samples for isotopic analysis as the Wüstebach catchment is forested and canopy-passage of precipitation influences Fyw [*Stockinger et al.*, 2017]. The samples were collected with six RS200 samplers (UMS GmbH, Germany) with a distance of 2 m to each other and to trees. The samplers consisted of a 50 cm long, 20 cm diameter plastic pipe which was buried in the ground. On top of it a 100 cm long plastic pipe with the same diameter was installed. An HDPE sample bottle (max. volume of 5000 ml) was placed inside the buried pipe and connected with plastic tubing to a funnel on top of the 100 cm long pipe. The funnel had a collecting area of 314 cm² and was protected by a wire mesh against foliage

and a table tennis ball in the funnel served as an additional evaporation barrier. Tests of the system showed the reliability in protecting the collected water from evaporation and in consequence isotopic fractionation for several weeks [*Stockinger et al.*, 2015]. Two samplers of the same design were placed in a clearing of the Wüstebach catchment to sample open precipitation, i.e., precipitation that has not passed through the spruce canopy. Streamflow samples for isotopic analysis were collected

weekly as grab samples in HDPE bottles at the outlet of the catchment.

Isotopic analysis was carried out using laser-based cavity ringdown spectrometers (models L2120-i and L2130-i, Picarro Inc., USA). Internal standards calibrated against VSMOW, Standard Light Antarctic Precipitation (SLAP2) and Greenland Ice Sheet Precipitation (GISP) were used for calibration and to ensure long-term stability of analyses [*Brand et al.*, 2014]. The

long-term precision of the analytical system was $\leq 0.1‰$ for $\delta^{18}O$.

We calculated weekly volume-weighed means of $\delta^{18}O$ for throughfall and open precipitation, which were further weighed according to the respective land-use percentage of spruce forest (79%) and clear-cut (21%) areas to generate a time series of precipitation $\delta^{18}O$ for the whole catchment. The derived precipitation isotope time series was then used together with the

weekly streamwater grab samples to calculate Fyw. While streamflow never ceased and thus a time series of weekly isotope values was available for the whole time series, there were weeks of no precipitation and thus gaps in the time series. Because of this for a 1-year calculation window on average 43 precipitation isotope values compared to 53 streamflow values were available. The total number of isotope values amounted to 156 for precipitation and 195 for streamflow. We could not always sample precipitation in weekly intervals, leading to bulk samples of 2-3 weeks on occasion. In this case, we assigned the

measured bulk isotope value to each week, while the measured bulk precipitation depth was proportionally assigned to each week according to the distribution of hourly precipitation measured at the meteorological station Kalterherberg.

For further hydro-meteorological and isotopic analyses several additional data were collected: we measured air temperature and relative humidity in 10-minute intervals at the TERENO meteorological station Schleiden-Schöneseiffen (Meteomedia

station, 572 m asl), located 3 km northeast of the catchment. We also calculated the runoff coefficient from runoff (Q) and open precipitation (P) as Q/P and used it for further analysis. Isotope data was complemented by $\delta^{18}O$ values of groundwater sampled in four different locations in weekly intervals since 2009. Groundwater was sampled by pumping first to avoid sampling stagnant water. Lastly, we calculated the d-excess of the precipitation samples using the slope and intercept of the global meteoric water line (d-excess = $\delta^2H – 8*\delta^{18}O$) [*Craig*, 1961; *Merlivat and Jouzel*, 1979].

**2.3 Fraction of young water**

We used a one-year time window which was moved in 7-days steps to calculate 189 Fyw estimates over the 4.5-year time series. A minimum time window length of one year was chosen to fully capture the annual isotope signal. Fyw is calculated

by fitting sine waves to both the seasonally-varying precipitation and streamflow isotope signals, respectively. We used the multiple regression algorithm IRLS (iteratively reweighted least squares, available in the software R) to minimize the influence of outliers:

$$C_P(t) = a_P \cos(2\pi f t) + b_P \sin(2\pi f t) + k_P,$$
$$C_S(t) = a_S \cos(2\pi f t) + b_S \sin(2\pi f t) + k_S \qquad (1)$$

with $C_P(t)$ and $C_S(t)$ the simulated precipitation and streamflow isotope values of time $t$, $a$ and $b$ regression coefficients, and $k$ and $f$ the vertical shift and frequency of the sine wave. The difference of $C_P(t)$ and $C_S(t)$ to the measured isotope time series in
precipitation and streamflow is minimized to fit the parameters a, b and k, while the frequency $f$ of the sine wave is known due to its annual character (i.e., if $C_P(t)$ and $C_S(t)$ are calculated in hourly time steps then the frequency $f$ is 1/8766; once per 24 x 365.25 hours). Precipitation isotope values were weighed using collected precipitation volumes, while streamflow was weighed using runoff volumes. The goodness-of-fit of the sine waves are expressed as the adjusted coefficient of determination $R^2$ ($R^2_{\text{adj}}$). If not otherwise stated we will use the mean of the streamflow and precipitation $R^2_{\text{adj}}$, as both sine waves are needed
to estimate the fraction of young water. After fitting the multiple regression equations, the amplitudes $A_P$ and $A_S$ and Fyw can be calculated:

$$A_P = \sqrt{a_P^2 + b_P^2}, \qquad A_S = \sqrt{a_S^2 + b_S^2},$$
$$F_{yw} = \frac{A_S}{A_P} \qquad (2)$$

Shifting the calculation window in 7-days steps resulted in a time series of varying Fyw estimates. Of course, the Fyw estimates cannot be considered independent from each other precluding the use of regression analysis to derive predictor variables (e.g., temperature, relative humidity) for the independent variable (Fyw). However, we used regression analysis to describe the average meteorological conditions during each Fyw time window. The thus derived "predictor" variables may have influenced
Fyw and could be investigated in future studies that use independent Fyw estimates.

Fyw calculation was done in a two-step process as the initial 189 Fyw results had large uncertainties that originated from a strong influence of the 2015 European heat wave (see results and supplementary material). Thus, in a second step we considered its influence and recalculated results while omitting precipitation isotope data of summer 2015. This greatly reduced
uncertainty. Apart from the 189 Fyw results we also calculated Fyw for the whole time series with one sine wave as was the standard of previous studies. We compared its peak timing and amplitude to the timing of peaks and amplitudes of the 189 sine waves.

**2.4 Hypotheses testing**

For clarity we want to highlight that each Fyw result was placed in the midpoint of the year it represents. That is, a data point located at any date represents the value for the six months before and six months after this date. For example, a Fyw result of 0.2 on 6th August 2013 means that between 5th February 2013 to 4th February 2014 on average 20% of runoff consisted of water younger than three months. The same logic applies to $R^2_{adj}$ values, amplitudes, phase shifts and hydrometeorological data if not explicitly stated otherwise. The hydrometeorological data was calculated as mean values for the 189 individual calculation years to facilitate comparison to the Fyw results that are averages valid for the respective calculation time frame.

Prior studies in the Wüstebach catchment identified changes of Fyw between 0.02-0.04 as significant [*Stockinger et al.*, 2016; *Stockinger et al.*, 2017]. Here, we employed Gauß error propagation on the sine wave fit parameters to carry their respective standard errors through to the Fyw results. Doing this resulted in the uncertainty of the 189 Fyw results as well as the uncertainty of Fyw calculated with the complete time series. We used the latter as the threshold value for testing the null hypothesis. In doing so, the time-variable Fyw results were tested against the data-inherent uncertainty of the complete time series. In our study we found that a threshold value of 0.04.

Based on this definition of a significant change in Fyw, three hypotheses were tested according to the following rules of acceptance:

1) Fyw estimates do not change over time (time-invariance)

This hypothesis is accepted if more than 90% of Fyw values are within ±0.04 of the mean value of all Fyw results. We chose a minimum percentage of 90% to ensure that the long-term time-invariance is captured. Larger changes of Fyw over time would indicate either flow path changes or a change in the relative contribution of different flow paths.

2) Short-term changes in the start of a tracer sampling campaign do not influence Fyw estimate (sampling-invariance)

This hypothesis is accepted if four consecutive Fyw results (i.e., four weekly shifts of the one-year time window) do not differ more than ±0.04. We thus investigated 186 four-week time windows of the in total 189 Fyw estimates. The short time span of four weeks ensures that the influence of possible long-term changes in catchment flow paths are not captured and only the influence of the start and end time of sampling one year of isotope data is investigated. In the case that Fyw shows stronger variations, the sampling time will likely have influenced Fyw results. Patterns to help identify such situations beforehand are then searched by analyzing the time of occurrence of these situations.

3) Fyw estimates are similar for calculation years that are centered around a given calendar month (seasonal-invariance)

This hypothesis is accepted if the Fyw results centered around a specific month do not differ more than ±0.04 within this month. To clarify, we did not calculate Fyw on a monthly basis but simply sorted the 189 Fyw results by the month they were assigned to (midpoint of the calculation year, see also explanation above). If the hypothesis is accepted it would indicate seasonal changes in the Fyw result as a function of the start date of a one-year sampling campaign. This would allow the pre-planning of sampling campaigns to establish comparable Fyw results. However, it is also possible that the hypothesis is accepted if Fyw is constant for all 189 results, as only the intra-month variance matters with this hypothesis. Contrary to the acceptance of the hypothesis, rejecting it for most months would indicate that there are no distinct seasonal patterns imprinted on Fyw.

An example of a theoretical Fyw time series is given in Figure 2. Despite it having a time-variant young water fraction, all three hypotheses are accepted. On a long term basis, the young water fraction does not deviate significantly from its overall mean value (time-invariance), choosing to start a one-year long sampling camping on a specific date or e.g., two weeks later would not significantly alter the result (sampling-invariance) and results show a seasonal behavior that is stable over longer time frames (seasonal-invariance). Therefore, these results would represent a runoff with a fraction of young water that systematically varies with the start of the sampling campaign, from a catchment with stable environmental conditions and water transport properties, and low sampling uncertainties.

## 3 Results

### 3.1 Isotopic and hydrometric data

Precipitation isotope ratios ranged from -3.04 to -17.80‰, spanning a range of 14.76‰ in $\delta^{18}O$ values. In comparison, streamflow values ranged from -7.78 to -8.74‰ with a range of 0.96‰ or only 1/15th of precipitation values. The volume-weighed groundwater isotope value was -8.43 ± 0.17‰. The maximum and minimum air temperatures were 27.0 and -7.4 °C, respectively, with a mean value of 7.6 °C. Relative humidity ranged from 96.8 to 32.3% with a mean of 82.2%. All the sampling years except winter season 2013/14 experienced a build-up of snow pack with a mean height of 15 cm. The absence of snow in 2013/14 correlated with on average higher temperatures (3.5 times the average temperature of the other years) and lower relative humidity (5% lower average relative humidity compared to the other years). The hydrometeorological and isotope data are presented in more detail in section 3.3.

### 3.2 Climatological influence on preliminary data set analysis

Before presenting final Fyw estimates we briefly introduce the detection and subsequent remedy of a climatological influence on the initial Fyw results and their uncertainty: the initial 189 Fyw estimates and their uncertainty significantly increased from July 2014 to December 2015 (supplementary Figure S1). The uncertainty of Fyw reached peak values of ±0.43. Concurrent with this, $R^2_{adj}$ values dropped close to 0 while being above 0.2 for most other Fyw results. The low goodness-of-fit and the

consequential large uncertainty could have been caused by outlier values or extraordinary catchment conditions in the Wüstebach. The hydrometeorological and isotopic data pointed to an influence of the 2015 European heat wave (see supplementary material). The heat wave was detectable in the Wüstebach catchment by the lowest relative air humidity, second lowest rainfall amounts, lowest runoff coefficient, high temperatures, and the complete disconnection of precipitation and

streamflow amplitudes (supplementary Figure S2). In addition, the 2015 European heat wave coincided with the lowest surface water temperatures of the North Atlantic since 1948 [*Duchez et al.*, 2016] which were visible by the loss of the seasonal d-excess signal. This created a situation where several months of precipitation isotope signal did not reach streamflow in the Wüstebach. The Fyw methods depends on comparable signals in precipitation and streamflow. Consequently, this disconnection of precipitation and streamflow added uncertainty to Fyw estimation. Therefore, we decided to omit the

precipitation isotope values between April to July 2015 (11 out of 156 precipitation isotope data; 7% of the measurements; Figure 3a) resulting in less Fyw uncertainty (average: 0.08, maximum: 0.31). We did not omit streamflow data during the same period as it contained Fyw information of the previous three months of precipitation and streamflow sine wave fitting had no impact on Fyw uncertainty (see results of Figure 4b below).

### 3.3 Isotopic and hydrometric data

After omitting summer 2015 precipitation data the sine waves for the whole study period had an $R^2_{adj}$ of 0.09 for precipitation and 0.23 for streamflow, respectively (Figure 3). The precipitation amplitude $A_P = 0.72‰$ and the streamflow amplitude $A_S = 0.08‰$ resulted in a Fyw of $0.12 \pm 0.04$. Thus, the threshold value for hypothesis testing was chosen as the absolute value 0.04. The 189 fitted sine waves had a wide range of $R^2_{adj}$ values: precipitation ranged from -0.02 to 0.63 with a mean of 0.22 and streamflow ranged from 0.00 to 0.55 with a mean of 0.25. The mean $R^2_{adj}$ (arithmetic average of precipitation $R^2_{adj}$ and

streamflow $R^2_{adj}$) for each calculation year ranged from 0.03 to 0.59 with a mean of 0.24. The sine waves showed strong variations in terms of amplitudes and phase shifts leading to distinct deviations from the sine wave fitted to the whole time series (Figure 3). Precipitation amplitudes ranged between 0.35 to 2.60‰ with a mean value of 1.26‰ while streamflow amplitudes ranged between 0.03 to 0.19‰ with a mean value of 0.10‰. The mean of all streamflow amplitudes was closer to the single sine wave amplitude (0.10‰ vs. 0.08‰) than those for precipitation (1.26‰ vs. 0.72‰). If we use the averages of

the 189 sine wave amplitudes to calculate Fyw, the result would be 0.08 instead of 0.12 of the single sine wave. This is less than the 0.04 difference in Fyw defined by this study as the data-driven threshold value for significant differences. Leaving out the period of low $R^2_{adj}$ values the single sine wave and the average of 189 amplitudes would both yield approximately 0.07. The overall pattern of the individual peaks was similar to the single sine wave peaks, except for the period of the 2015 European heat wave when between June to October 2015 a distinct double-peak in precipitation was visible. The individual sine waves

followed the general pattern of enriched isotopic values during summer months and depleted values in winter.

The mean $R^2_{adj}$ showed a marked decrease during July 2014 to October 2015 with values falling well below 0.2 (Figure 4a). Approximately at the same time the Fyw results varied strongly (mean and maximum change of Fyw between consecutive

one-year windows: 0.02 and 0.12) and the uncertainty was large (mean uncertainty: ±0.11). Contrary to this, during periods of larger $R^2_{adj}$ the change in Fyw was more modest (mean and maximum change of Fyw between consecutive one-year windows: 0.01 and 0.05) with lower uncertainty (mean uncertainty: ±0.04). To find possible modeling influences on the Fyw uncertainty we first compared the mean $R^2_{adj}$ with it and found that they were correlated (Figure 4b inset, $R^2 = 0.65$). Following this we further investigated relationships between Fyw uncertainty and the amplitudes, phase shifts and vertical shifts of the 189 sine waves but only show results for throughfall amplitudes, as the other parameters had no correlation (Figure 4b). The throughfall amplitudes were correlated with an $R^2 = 0.79$ while contrary to this streamflow amplitudes had an $R^2 = 0.04$. Thus, the Fyw uncertainty was strongly controlled by the amplitudes of the precipitation sine waves while the streamflow sine waves barely influenced it.

The baseline for Fyw was around 0.05 (Figure 4). Before the low $R^2_{adj}$ period Fyw was around 0.05, increased to about 0.1 for a short time and then fell back to 0.05. After the low $R^2_{adj}$ period Fyw also fell to about 0.05, before rising in the end. Thus, during the 4.5-years Fyw seldom fell below the baseline of 0.05 and we assumed that during any one-year period the Wüstebach catchment will have at least 0.05 Fyw. Overall, the 189 Fyw results were positively skewed (Figure 5). Around 30% of results indicated a Fyw of 0.06, followed by 55% of results indicating a Fyw up to 0.08. Few Fyw values are higher than 0.16 with possible outliers between 0.26 to 0.28. Leaving out the period of low $R^2_{adj}$ values does not change the skewness of the histogram. However, values of Fyw larger than 0.16 disappeared in favor of 0.06 that shifted from 30% to 40% relative frequency.

### 3.4 Hypothesis 1: Time-invariance

The mean value of all Fyw results was 0.09. Consequently, 90% of all Fyw results must lie within 0.05 to 0.13 to accept hypothesis 1. Out of the 189 Fyw results 159, i.e. 84%, were within those boundaries (Figure 6a). It could be possible that the period between July 2014 and October 2015 with low $R^2_{adj}$ values and erratic Fyw behavior significantly influenced the rejection of the hypothesis. Therefore, in a second step we excluded this period, calculated the mean for those values and evaluated Fyw results again (Figure 6b). The new mean Fyw was 0.07 with 93% of results found between 0.03 to 0.11. Thus, contrary to using all data the hypothesis could be accepted if the period of large uncertainty was left out. We then compared the time-variable Fyw to hydrometeorological measurements (Figure 7) and found that neither temperature nor relative humidity were correlated with Fyw (not shown). While throughfall volume, runoff volume and snow height were also not correlated (Figure 7a-c) the runoff coefficient (Q/P) was negatively correlated with $R^2 = 0.25$ and p-value = 1.7E-11 (Figure 7d). Leaving out again the period from July 2014 to October 2015 reduced the correlation to $R^2 = 0.08$ and p-value = 9.8E-4.

### 3.5 Hypothesis 2: Sampling-invariance

Here we tested if short-term changes in the start of a one-year sampling campaign could significantly influence Fyw. The hypothesis is accepted if during any consecutive four weeks Fyw did not differ more than 0.04. On multiple occasions this rule

was violated for the full data set, as well as for the reduced one (discounting the low $R^2_{adj}$ period), so we rejected hypothesis 2 (Figure 6). Thus, the start time of a one-year long sampling campaign could significantly influence Fyw. The periods when hypothesis 2 was violated were neither equally spaced in time (Figure 6) nor did they show significant correlations to hydrometric (Figure 7) or meteorological (not shown) variables. The only observation made was that hypothesis 2 seems to have preferentially failed around the 2015 European heat wave.

## 3.6 Hypothesis 3: Seasonal-invariance

As mentioned in the methods, the Fyw results were put in the middle of the one-year calculation period (calculating from February 2016 to February 2017, the result would be displayed as a data point in August 2016). We grouped together all Fyw results that were assigned to a specific calendar month and used a box plot to detect possible seasonality (Figure 8). Only in January and February was the difference in Fyw below 0.04. When leaving out the period with low $R^2_{adj}$, January to August stayed within ±0.04. Thus, we also rejected hypothesis 3 based on all data as our results did not indicate pronounced seasonality. Nonetheless, a trend of declining Fyw from January to June was visible that reversed from July onwards. Additionally, the standard deviation of Fyw, the interquartile range of the boxplots and the number of outliers increased starting with June until October/November. We compared this behavior qualitatively to the start and end time of snow influence in the Wüstebach, which usually started in December and the last melt event happened in February. Since the influence of this delayed signal transmission from precipitation to streamflow does not immediately end with the final snowmelt in February, we assumed that snowmelt still influenced streamflow for the following two months, i.e., until April. This comparison showed that calculation years that included one year's winter had lower interquartile ranges, a lower number of outliers and smaller standard deviations. On the other hand, calculation years that included winters of two different years (e.g., a calculation year starting and ending in December) matched the boxplot results with increased uncertainty (Table 1).

## 4 Discussion

Judging by the isotope data, we generally expect that groundwater was recharged locally from precipitation as the long-term, volume-weighed $\delta^{18}O$ of precipitation with -8.53‰ was close to the quasi-constant $\delta^{18}O$ of groundwater with a 5-year mean of -8.43 ± 0.17‰. Streamflow was substantially comprised of groundwater as its volume-weighed $\delta^{18}O$ was -8.40‰. The study by *Weigand et al.* [2017] came to the same conclusion for the Wüstebach catchment using wavelet analysis of nitrate and DOC data collected at mainstream and tributary locations. While lower altitude locations of the catchment near the outlet were dominated by groundwater, higher altitude areas were less affected. This finding was additionally supported by field observations of shallow groundwater.

## 4.1 Sine wave fits

The single sine wave fits to all data had low $R^2_{adj}$ values (0.09 for throughfall and 0.23 for streamflow). Compared to this, the 189 individual sine waves reached a maximum $R^2_{adj}$ of 0.63 and were often larger than 0.2. This indicated that the single wave fit to multi-year data is an oversimplification of the inter-annual variability in meteoric and streamflow isotope data and annual sine waves better capture the variability. One might argue that sine waves are a non-adequate function to describe the data variability if their $R^2$ is low. However, Fyw estimation is based on comparing sine wave amplitudes [*Kirchner*, 2016a] and no similar method exists to calculate it with different functions.

Completely undetectable by a single sine wave fit, the 189 sine waves highlighted a hydrologic change in the Wüstebach catchment caused by the 2015 European heat wave: the disconnection of precipitation and runoff. First, the general shapes of the 189 precipitation and 189 streamflow sine waves were similar (Figure 3), which can be seen, e.g., in the positive and negative peaks occurring around September 2014 and 2016 and February 2013 and 2014, respectively. Additionally, throughfall and streamflow amplitudes generally matched each other (supplementary Figure S2a). This indicated that throughout the 4.5-year time series the characteristic of the precipitation $\delta^{18}O$ signal was for the most part consistently and quickly transferred to the streamflow $\delta^{18}O$ signal within a year. However, the relationship between precipitation and streamflow considerably changed due to the influence of the 2015 European heat wave: while the sine wave double-peak of precipitation in summer 2015 was not transferred to streamflow (Figure 3), the amplitudes of both lost their close relationship at the same time (supplementary Figure S2a). After the heat wave the general shape of precipitation and streamflow sine waves matched each other again while their respective amplitudes regained their former relationship, albeit weakened: the large amplitude peak in throughfall in April 2016 again led to increasing streamflow peaks. Thus, considering the general hydrological observations obtained from the isotope data discussed above, we conclude that a certain percentage of precipitation became groundwater while another percentage that might or might not be Fyw quickly generated runoff, conserving the precipitation $\delta^{18}O$ signal in streamflow and resulting in the similar shapes of the 189 sine wave pairs. The 2015 European heat wave greatly disturbed the usually occurring runoff-generation process in the Wüstebach, leading to a disconnection of precipitation and streamflow signal.

A fast transmission of precipitation to streamflow was also found by *Jasechko et al.* [2016], and the fact that a part of precipitation quickly becomes streamflow is already inherent in Fyw. The new insight of the present study is the unexpected close resemblance of the 189 sine waves for precipitation and streamflow although the groundwater influence seems to have dominated in the Wüstebach. The simultaneous strong attenuation of the $\delta^{18}O$ streamflow signal while at the same time retaining much of the precipitation $\delta^{18}O$ signal characteristics can be explained by mixing with a quasi-constant $\delta^{18}O$ source, e.g., with groundwater. This would not alter the pattern but only attenuate the signal. Thus, the 189 sine waves gave a strong indication that streamflow in the Wüstebach consisted of precipitation and groundwater with no additional, unaccounted

sources of runoff such as subsurface flows from outside the catchment boundaries. This supports a previous study that closed the water-balance for the Wüstebach catchment using only precipitation, evapotranspiration and runoff data [*Graf et al.*, 2014] and is essential information for e.g., endmember-mixing analysis [*Barthold et al.*, 2011; *Katsuyama et al.*, 2001] or isotope hydrograph separation [*Klaus and McDonnell*, 2013]. Similar to before, this hydrological information about the Wüstebach
catchment would have been impossible to detect with a single sine wave fit.

## 4.2 Fraction of young water

The fact that Fyw calculated with the average amplitudes of 189 precipitation and streamflow sine waves was within the $\pm0.04$ boundary to Fyw calculated with a single sine wave (0.08 vs. 0.12) indicated that the single sine wave generally averaged the behavior of the 189 ones. If the isotope data and Fyw results of the period of low $R^2_{adj}$ values was left out, the average Fyw of
the 189 sine waves compared even better to the single Fyw (approximately 0.07 in both cases). Thus, if a study is interested in the overall behavior of a multi-year time series, a single sine wave fit would seem sufficient. Nevertheless, hypothesis 1 was rejected for both cases as Fyw varied significantly within this multi-year time series (Figure 6). Using a moving time window to calculate a host of Fyw values ensures that the entire range of possible Fyw estimates is considered with an average estimate and most importantly its uncertainty.

Most of the isotope data between 7-day calculation window shifts were the same. Still, during the low $R^2_{adj}$ period Fyw occasionally fluctuated in the order of 0.12 between one-week shifts. From a hydrological standpoint it is difficult to imagine a short-term change in flow paths of this magnitude for annual averages. Given that the Fyw calculation is based on comparing the amplitudes of precipitation and streamflow and a low $R^2_{adj}$ indicates a weak fit to a sine wave shape, we assumed that in
our case the Fyw calculation method reached its limit below an average $R^2_{adj} = 0.2$. Fyw became highly sensitive to a small change in input data and in consequence highly uncertain. We recommend further investigations of the sensitivity of Fyw to the goodness-of-fit (not necessarily only measured with $R^2_{adj}$) for future studies. It remains to be seen if a value of 0.2 for $R^2_{adj}$ is a general critical threshold for Fyw or if different catchments show varying results. Such studies should consider that the Fyw uncertainty was correlated with throughfall amplitudes (Figure 4b), raising the question if a curve fit with $R^2_{adj} = 0.6$ is
objectively better than a fit with $R^2_{adj} = 0.3$ when the underlying isotope data have completely different amplitudes. A decrease in the goodness-of-fit of the sine wave when amplitudes are low was also found by *Lutz et al.* [2018].

A difference of $\pm0.04$ Fyw was defined as the data-driven threshold value for significant differences in Fyw by this study. The acceptance or rejection of our null hypotheses will thus inform if the time-variability of Fyw is large in comparison to the
averaged Fyw value and its uncertainty. We recommend using different thresholds that are suited to the purpose of calculating a Fyw estimate. Purposes can range from any application of the method to answer questions about the quantity and quality of water resources for various industrial, touristic or infrastructural uses. First, a critical difference in Fyw should be defined by each application that reflects e.g., the vulnerability of aquatic ecosystems to certain pollutant loads. If an increase or decrease

by less than this value does not impact the results of an, e.g., risk assessment, then these Fyw changes are non-significant for the practical purpose at hand. The present study did not aim to answer any specific question related to Fyw that would justify setting a threshold value a priori but investigated the time-variability of Fyw and used the data-inherent uncertainty as its threshold value. Thus, while our hypotheses are accepted or rejected, the results of the hypothesis tests might change

completely if we would answer practical questions about the Wüstebach such as the vulnerability to pollutant loads of a certain chemical substance.

The 2015 European heat wave was among the top ten heat waves of the past 65 years and was accompanied by the lowest surface water temperatures of the North Atlantic in the period of 1948 to 2015 [*Duchez et al.*, 2016]. The North Atlantic

influences the European summer climate [*Ghosh et al.*, 2017] and is an important vapor source for precipitation over Europe [*Hurrell*, 1995; *Trigo et al.*, 2004]. The combined effects of low ocean water temperatures and high air temperatures in Europe were visible in the d-excess that lost its clear seasonal signal in summer 2015 (supplementary Figure S2d). The d-excess of precipitation samples is strongly controlled by the relative humidity of the moisture source [*Pfahl and Sodemann*, 2014; *Steen-Larsen et al.*, 2014] which in turn would change with changing surface water temperatures and thus changing evaporation

rates. Additionally, the increased European air temperatures during the heat wave would increase secondary evaporation of falling raindrops, further altering the d-excess of precipitation samples. The North Atlantic and European temperature anomalies of 2015 explain the behavior of the d-excess as well as the unusual double-peak of the 189 sine waves that was observed for summer 2015 in the Wüstebach.

Apart from affecting the isotopic input signal into the Wüstebach catchment, the temperature anomalies of 2015 also changed the hydrological behavior of the Wüstebach: precipitation was largely disconnected from streamflow and the isotopic signal was not transferred (supplementary Figure S2a-c). This directly increased Fyw uncertainty during this period. Future studies must be careful in comparing Fyw estimates of different time periods, especially if a heat wave occurred during those periods. We assume that mostly small headwater catchment with shallow soils are strongly affected by this effect but do not exclude

the possibility of other catchments being affected in varying degrees too. It is highly advisable to investigate further in this direction, as the probability of heat waves in the period from 2021 to 2040 is poised to increase [*Russo et al.*, 2015]. This, in extension, means that the probability of getting highly uncertain Fyw results will increase too. We argue that heat waves are actively disturbing the estimation of Fyw by potentially decoupling the input from the output isotope signal. This can be more clearly illustrated by the theoretical worst-case scenario: the decoupling of precipitation and streamflow signal for a full year

and streamflow being solely fed by another source, e.g., groundwater. Why, in this case, would we trust the Fyw result, no matter the magnitude of the uncertainty and goodness of sine wave fit? Thus, it is reasonable to assume that any amount of decoupling will add uncertainty to Fyw, as demonstrated by our data and results. Only by comparison to other time frames where the uncertainty was smaller was it possible for us to detect that the uncertainties for summer 2015 were unusually large.

## 4.3 Hypothesis 1 – Fyw is time-variant

Hypothesis 1 was rejected because the Fyw varied in the long-term. For example, in December 2013 Fyw was 0.06 while two months later it increased to 0.1, almost doubling. From summer 2016 to the end of the time series Fyw even tripled from 0.06 to 0.15. These differences in Fyw results complicate catchment comparisons as the result does not only depend on catchment characteristics but also on when isotope data was collected. As far as we can tell, the recent Fyw catchment comparison study of *Lutz et al.* [2018] used the same sampling period for precipitation and streamflow for all 24 investigated catchments. In contrast, the studies of *Jasechko et al.* [2016] and *von Freyberg et al.* [2018] had isotope sampling periods varying in start date and overall length for the 254 and 22 investigated catchments, respectively, potentially influencing the uncertainty for the inter-catchment comparison according to the results of our study.

In the Wüstebach catchment the baseline for Fyw was around 0.05. This lower boundary is useful in assessing pollutant risk and nutrient loss in the catchment as it defines a minimum expected load that will quickly appear in the stream if combined with precipitation volumes and chemical substance concentrations. Using a single sine wave would not have revealed this lower boundary.

The variability in Fyw of this study could not be explained by meteorological or hydrometric variables. *Lutz et al.* [2018] found a negative correlation between annual precipitation and Fyw. The study of 22 Swiss catchments by *von Freyberg et al.* [2018] found significant positive correlations between Fyw and mean monthly discharge and precipitation volumes. Fyw of this study neither correlated with precipitation nor with runoff (Figure 7a and Figure 7b). Such contradictions could be explained by the different sampling periods of our study and the mentioned studies but also by differing catchment characteristics. Additionally, the present study investigated the same catchment temporally while the other studies investigated spatially different catchments. Furthermore, *Lutz et al.* [2018] found complex interactions between several catchment characteristics and Fyw, possibly resulting in nonsignificant linear regressions between Fyw and individual catchment characteristics. However, the runoff coefficient Q/P was negatively correlated with Fyw (Figure 7d). Physically, this could be explained by the fact that if annual runoff volumes increase per annual precipitation volume then the additional runoff volumes were provided by catchment storage. This increased the percentage of old water in streamflow and relatively decreased the Fyw since catchment storage consists of old water [*Gabrielli et al.*, 2018].

## 4.4 Hypothesis 2 & 3 – Fyw is sensitive to sampling and has no clear seasonal pattern

While hypothesis 1 concentrated on long-term changes, hypotheses 2 focused on short-term changes where choosing to start a one-year sampling campaign by one to four weeks later could lead to significantly different results. On several occasion Fyw differed more than ±0.04 within four weeks (Figure 6). This means that the choice of the sampling period has a large potential for uncertainty in the Fyw estimates for studies that can monitor the water stable isotopes in precipitation and streamflow for

only one year. The obtained Fyw could be a potential outlier, a larger value or part of the Fyw baseline around 0.05 in the present study. As the timing of the violation of hypothesis 2 did not correlate with any meteorological or hydrometric data it was not possible to determine the conditions under which the sampling period led to higher Fyw uncertainty. A relationship with the 2015 European heat wave is possible, albeit not fully evident. Nonetheless, as discussed above, the choice of another threshold value beside the data-inherent ±0.04 may lead to an increase in the number of significant short-term Fyw changes. The results of this study indicate that estimating Fyw with data of a single year might not be enough for fully understanding catchment behavior. Quoting Kirchner et al. [2004]: "If we want to understand the full symphony of catchment hydrochemical behavior, then we need to be able to hear every note.". A single Fyw result is one note in the symphony of potential Fyw results slumbering in multi-year data sets.

Fyw did not have a clear seasonal pattern in that not all the months had Fyw differences of less than ±0.04 (Figure 8). A pattern was visible with larger Fyw with less uncertainty when the sampling campaign was centered around winter months compared to lower Fyw with larger uncertainties when the campaign was centered around summer months. The behavior of Fyw uncertainty can potentially be explained by the influence of snow and is similar to the proposed problem that the 2015 European heat wave introduced: a tracer signal in precipitation/streamflow that does not have any instantaneous connection with its counterpart streamflow/precipitation. This disconnection by snow could be explained by the longer delay in signal transmission of snowfall compared to rainfall due to snow blanket build-up. Consider a winter at the start of a sampling campaign: it is likely that streamflow will feature the snowmelt isotope signal originating from snowfall of e.g., several weeks ago that is not featured in the precipitation isotope data of this calculation year. Furthermore, snow blankets also change the isotopic signal potentially to a degree that obscures seasonal isotope patterns [*Cooper*, 2006]. This in turn would affect the Fyw estimate and its associated uncertainty. Currently, we recommend that if studies can only sample one year of data in snow-influenced catchments to not sample winters of two different calendar years and to design the sampling such that only one year's winter is in the time series. Future studies should provide more evidence if Fyw calculated by one year of isotope data shows a seasonal behavior or not and how snow influences the uncertainty. We highly recommend calculating a time series of Fyw, e.g., with the method of this study, to understand the temporal behavior of Fyw for the investigated catchment and to be able to evaluate possible uncertainties for Fyw estimation.

A difference in Fyw when only one year of isotope data is available was also observed by *Stockinger et al*. [2017] for the same catchment using only two calculation years without any further investigations in this direction, as it was not the main objective of their study to investigate Fyw time-variability and uncertainty. Only two Fyw were calculated in contrast to the 189 results of the present study (approximately 1%), making insights into the possible causes and a judgement if varying Fyw results are an isolated result or the rule impossible. Fyw for these years were 0.06 and 0.13, respectively. The authors assumed that using the complete time series averages sub-sets of the time series as the Fyw for the whole time series was approximately 0.13, so in between 0.06 and 0.13. However, this happened by coincidence. The present study shows that the two Fyw could have been

very different, e.g., both near 0.05. Then, Fyw of the whole time series would not have averaged the results of the two individual years. Thus, only the complete picture of all 189 individual Fyw results allowed a better judgment of Fyw time-variability and uncertainty. With knowledge from the current study, we would even consider one of the hydrological calculation years of *Stockinger et al.* [2017] as highly uncertain and possibly influenced by the 2015 European heat wave.

## 5 Conclusions

The fraction of young water (Fyw) is a promising new measure to estimate the fast transport of precipitation through a catchment to the stream. To calculate Fyw, sine waves are fitted to the water stable isotopes in precipitation and streamflow and their respective amplitudes compared. This is usually done for the complete time series available, ranging from less than a year to multiple years. This study used a moving one-year window to investigate the temporal variance of Fyw and its uncertainty for a 4.5-year long time series. Using 189 Fyw results instead of a single multi-year one, we were able to increase our hydrometeorological knowledge about the study catchment: (1) a potential strong influence of the 2015 European heat wave on Fyw estimates and uncertainties was discovered, which is a problem which could magnify in the future considering global warming; (2) precipitation and groundwater seemed to be the only end-members in streamflow which is information that isotope hydrograph separation studies can greatly benefit from; (3) a lower boundary of 0.05 Fyw was found, aiding e.g. pollutant risk studies in calculating minimum expected loads. Testing three hypotheses about the time-variability of Fyw we found that both in the long and short term Fyw is time-variable as defined by this study by the data-inherent ±0.04 threshold, while showing no clear seasonal pattern. The long-term variability has implications for catchment comparison studies when different time periods are investigated. Short-term variability indicated a potentially high sensitivity to the sampling period, where a shift of 1-4 weeks in the start of a one-year long sampling campaign significantly influenced Fyw. No pronounced seasonality of Fyw could be derived. However, a possible influence of snow pack led to the recommendation of sampling one year's winter and avoiding sampling the winters of two different years. If feasible, we recommend investigating a multi-year time series of tracer data with the method suggested in this study to enhance our knowledge of the sensitivity of Fyw to the chosen time frame in different catchment situations and the behavior of its uncertainty. That is, to use a one-year moving time window and estimate an ensemble of Fyw results and its uncertainty. Based on the goodness-of-fit for all 189 calculated sine waves and the corresponding Fyw behavior, we recommend considering that Fyw based on $R^2_{adj}$ below 0.2 might be highly uncertain. This must be verified by other dedicated studies of different catchments and would allow for a better comparability of Fyw results with various goodness-of-fits. The present study shows the importance of considering inter-annual fluctuations in the amplitudes of isotope tracer data and consequently of derived Fyw estimates in further learning about the uncertainty of Fyw and in aiding in catchment comparison studies.

**Acknowledgements**

We gratefully acknowledge the support by the SFB-TR32 ''Patterns in Soil-Vegetation-Atmosphere Systems: Monitoring, Modelling, and Data Assimilation'' funded by the Deutsche Forschungsgemeinschaft (DFG) and TERENO (Terrestrial Environmental Observatories) funded by the Helmholtz-Gemeinschaft. Holger Wissel, Werner Küpper, Rainer Harms, Ferdinand Engels, Leander Fürst, Sebastian Linke and Isabelle Fischer are thanked for supporting the isotope analysis, sample collection and the ongoing maintenance of the experimental setup. We appreciate the helpful comments of three anonymous reviewers that greatly improved the present study and the work of editor Patricia Saco. The data used in this study can be acquired from the corresponding author.

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

**Months from January (01) to December (12)**

| | | | | | | | | | | | | |
|---|---|---|---|---|---|---|---|---|---|---|---|---|
| 07 | 08 | 09 | 10 | 11 | 12 | **01** | 02 | 03 | 04 | 05 | 06 | 07 |
| 08 | 09 | 10 | 11 | 12 | 01 | **02** | 03 | 04 | 05 | 06 | 07 | 08 |
| 09 | 10 | 11 | 12 | 01 | 02 | **03** | 04 | 05 | 06 | 07 | 08 | 09 |
| 10 | 11 | 12 | 01 | 02 | 03 | **04** | 05 | 06 | 07 | 08 | 09 | 10 |
| 11 | 12 | 01 | 02 | 03 | 04 | **05** | 06 | 07 | 08 | 09 | 10 | 11 |
| 12 | 01 | 02 | 03 | 04 | 05 | **06** | 07 | 08 | 09 | 10 | 11 | 12 |
| 01 | 02 | 03 | 04 | 05 | 06 | **07** | 08 | 09 | 10 | 11 | 12 | 01 |
| 02 | 03 | 04 | 05 | 06 | 07 | **08** | 09 | 10 | 11 | 12 | 01 | 02 |
| 03 | 04 | 05 | 06 | 07 | 08 | **09** | 10 | 11 | 12 | 01 | 02 | 03 |
| 04 | 05 | 06 | 07 | 08 | 09 | **10** | 11 | 12 | 01 | 02 | 03 | 04 |
| 05 | 06 | 07 | 08 | 09 | 10 | **11** | 12 | 01 | 02 | 03 | 04 | 05 |
| 06 | 07 | 08 | 09 | 10 | 11 | **12** | 01 | 02 | 03 | 04 | 05 | 06 |

Table 1. The calculation years used for the boxplots of Figure 8. For example, the first row shows a calculation year starting in July and ending in July, where the Fyw result was assigned to January. Grey shaded areas are the usual beginning of snowfall and the final snowmelt (Dec to Feb, dark shaded) with an assumed prolonged influence of snowmelt on streamflow until April (light-shaded).

5  Green coloured calculation years highlight snow influence of only one winter within this year, while red coloured calculation years highlight influence of two different winters.

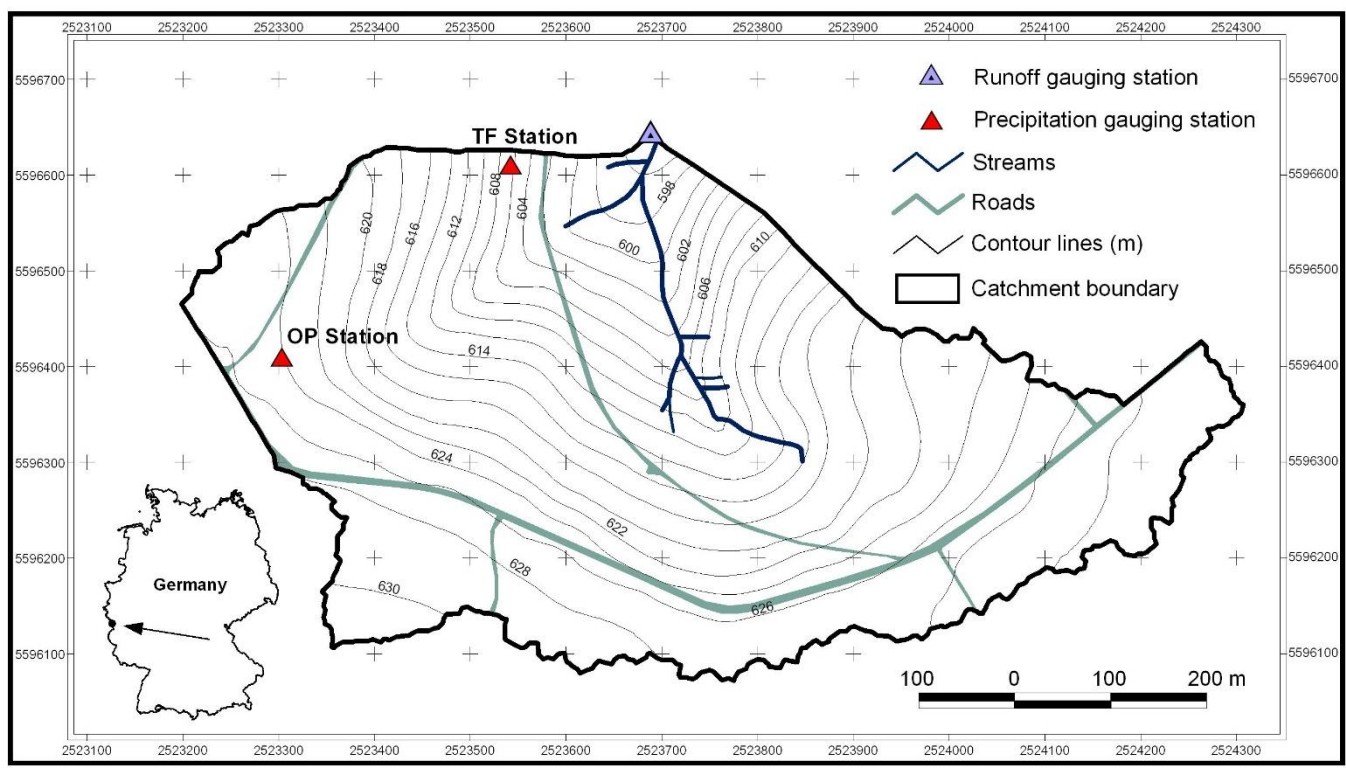

**Figure 1. Map showing the Wüstebach catchment and the used monitoring stations. OP Station is the open precipitation collection site, while TF Station is the throughfall station.**

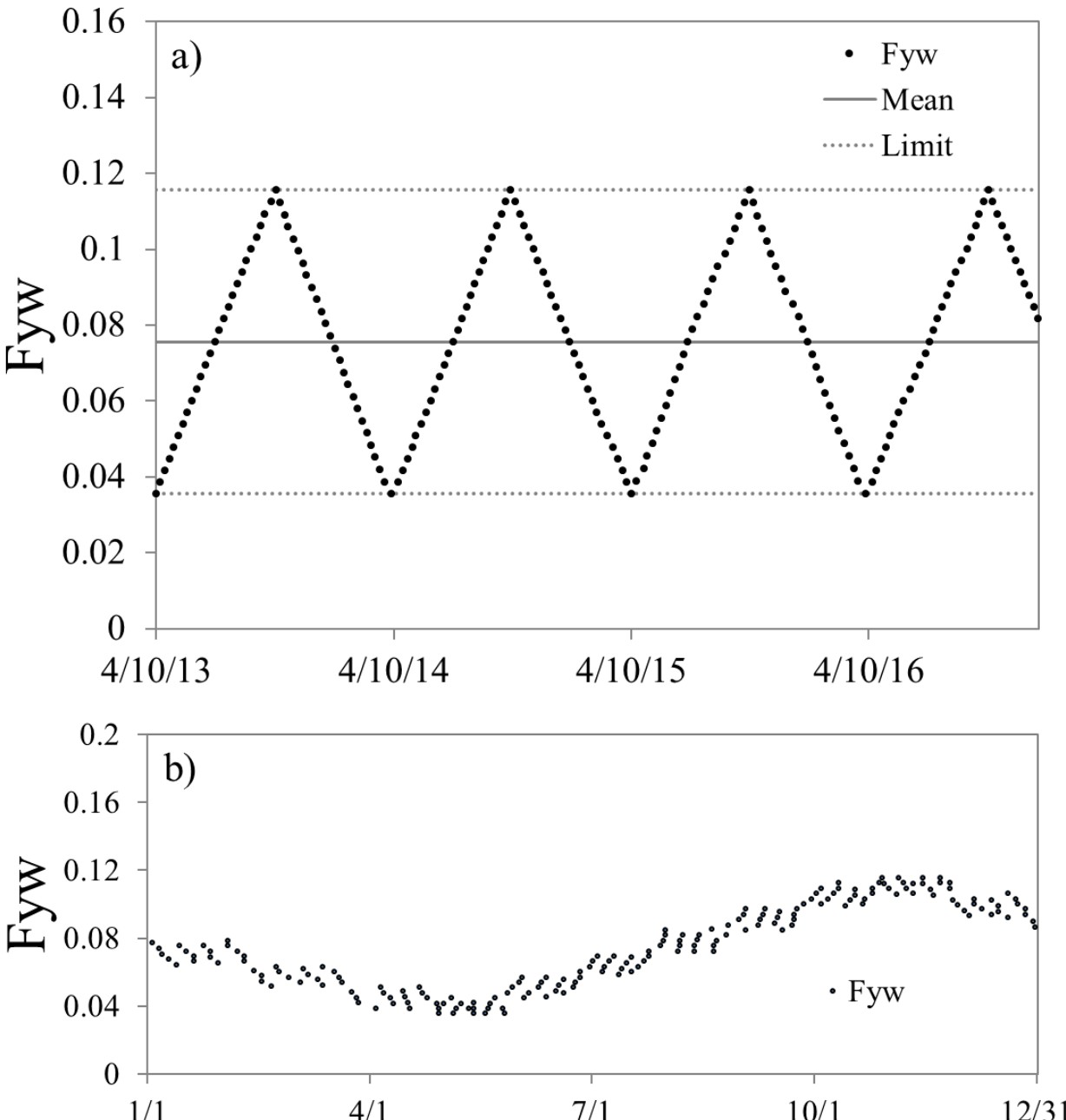

**Figure 2. Panel (a): Example of a theoretical Fyw time series where despite the time-variance all three null hypotheses are accepted: (1) more than 90% of Fyw values lie within ±0.04 of the mean of all values; (2) Fyw does not change more than ±0.04 over the course of four weeks; (3) Fyw for each month does not change more than ±0.04 within a month (panel (b)).**

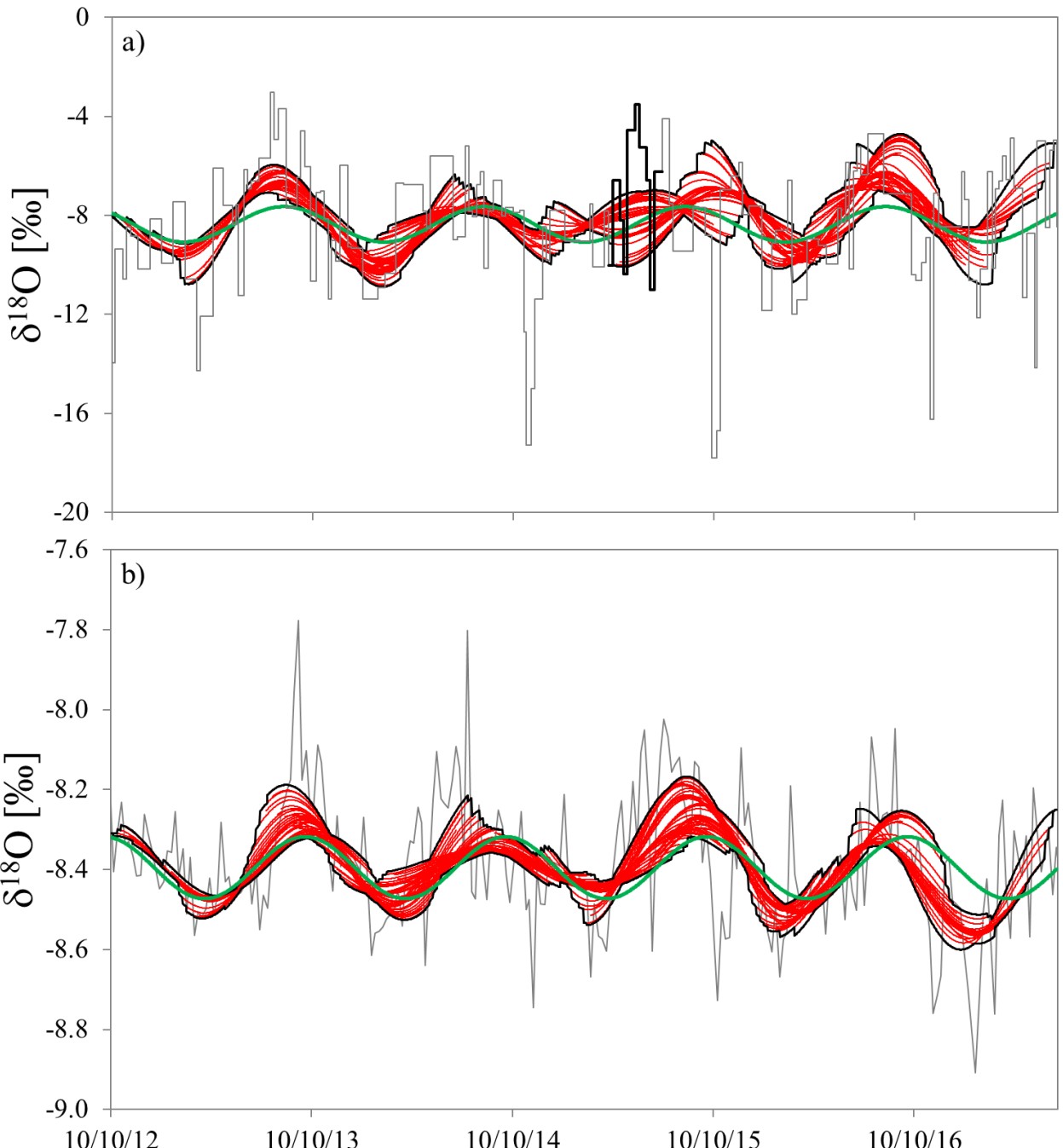

**Figure 3.** Sine waves (red lines) were fitted to (a) throughfall and (b) streamflow stable isotope data (grey line) with maximum and minimum values at each point in time (black enveloping curve). In comparison a single sine wave was fitted to the complete data set for both throughfall and streamflow (green lines). The omitted precipitation isotope values of the 2015 European summer heat wave are shown in panel (a) with bold black lines.

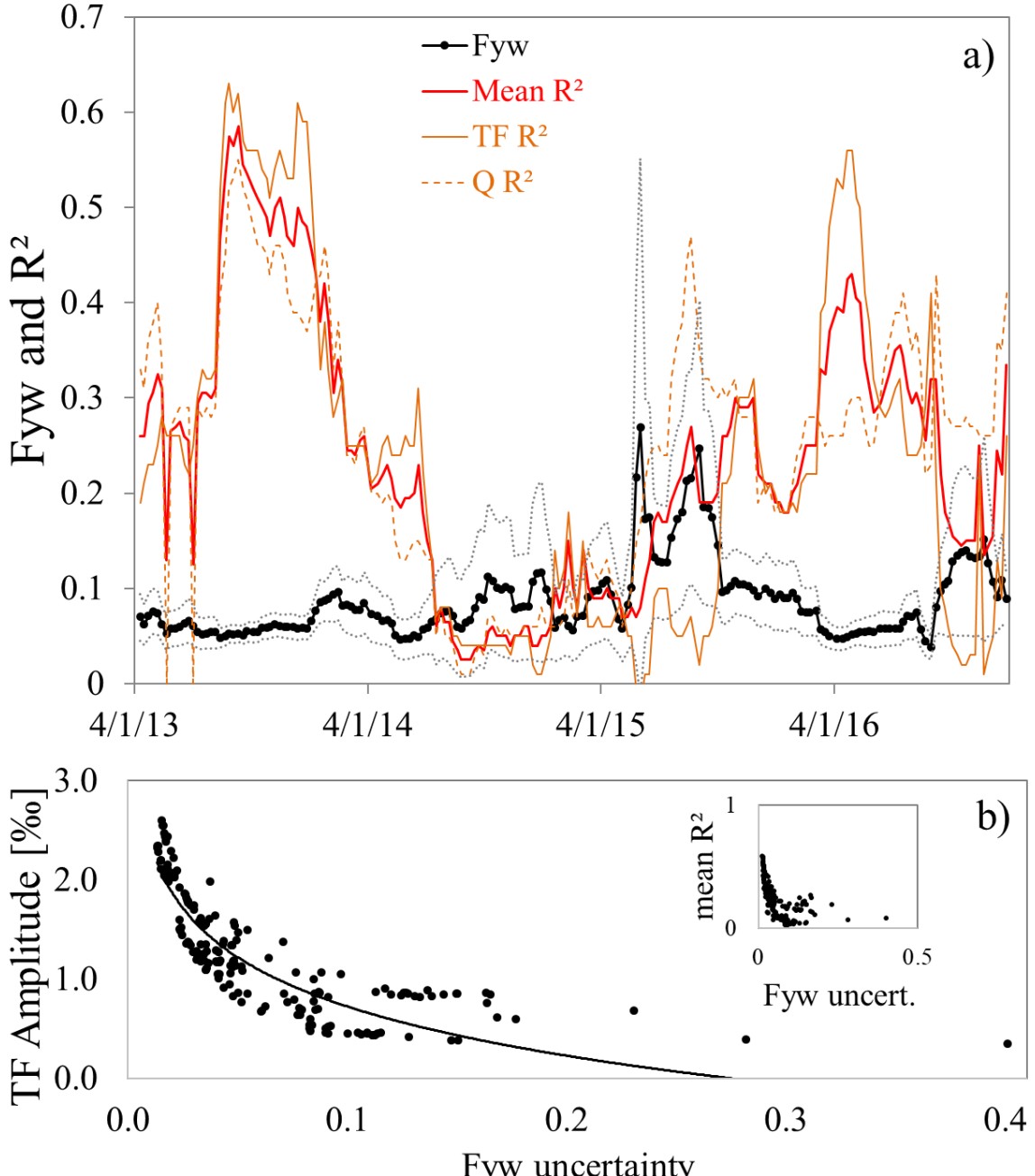

**Figure 4. (a) Fyw results and their uncertainty (black and grey lines) plotted against $R^2_{adj}$ for throughfall (TF $R^2$, solid orange line) and runoff (Q $R^2$, dashed orange line) sine wave fits and their average (Mean $R^2$, red line). All values are shown at the midpoint of the respective year they are valid for. Panel (b) shows throughfall amplitudes (TF Amplitude) versus the Fyw uncertainty. The regression equation is TF Amplitude = -0.716 ln(Fyw uncertainty) – 0.9236 with an $R^2$ of 0.79. A similar comparison between runoff amplitudes and Fyw uncertainty showed no relationship ($R^2$ of 0.04, not shown). The inset shows the Fyw uncertainty against mean $R^2_{adj}$ values of streamflow and precipitation.**

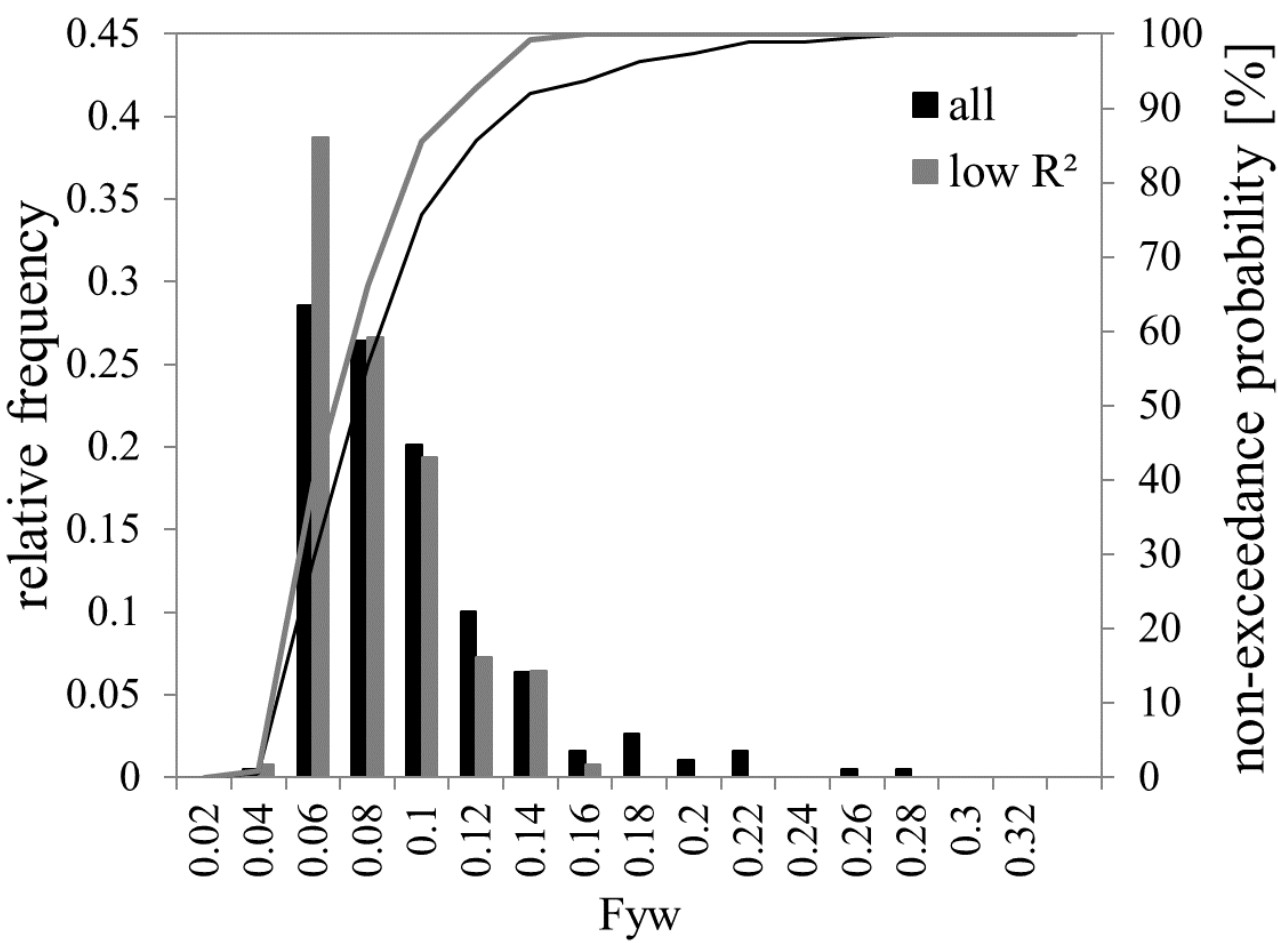

**Figure 5. Histograms and cumulative distribution functions of all Fyw results (black) and of the results when the low R²ₐdⱼ period is left out (low R², grey).**

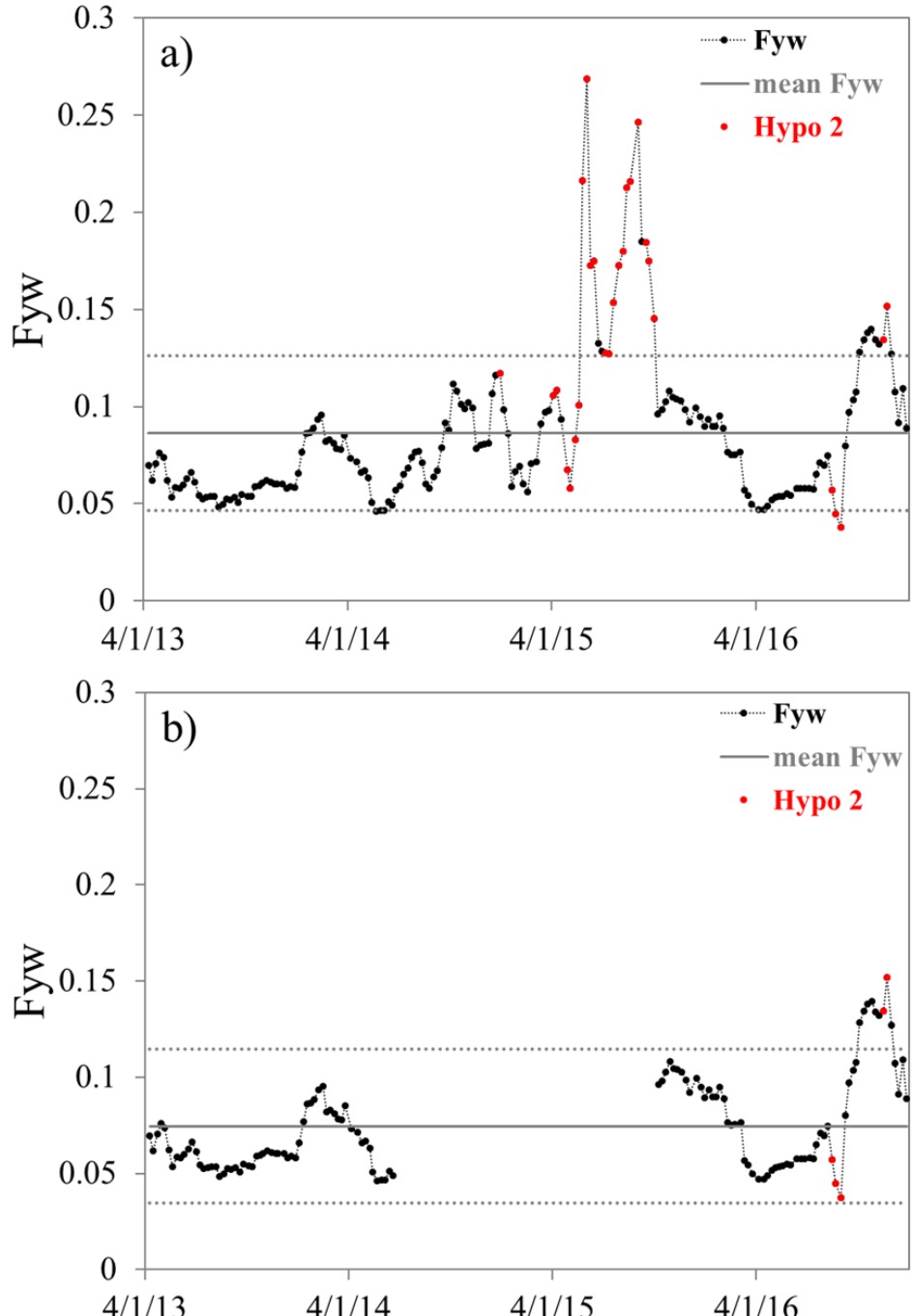

**Figure 6. Fyw compared to the mean Fyw (solid grey line) and a ±0.04 margin around it (dotted grey lines) to test hypothesis 1 (90% of all Fyw results are within the mean Fyw ±0.04). Red data points are periods where within four weeks Fyw differed more than 0.04 (testing hypothesis 2). Once all data was used (panel a) and subsequently data of the low $R^2_{adj}$ period between July 2014 to October 2015 was left out (panel b).**

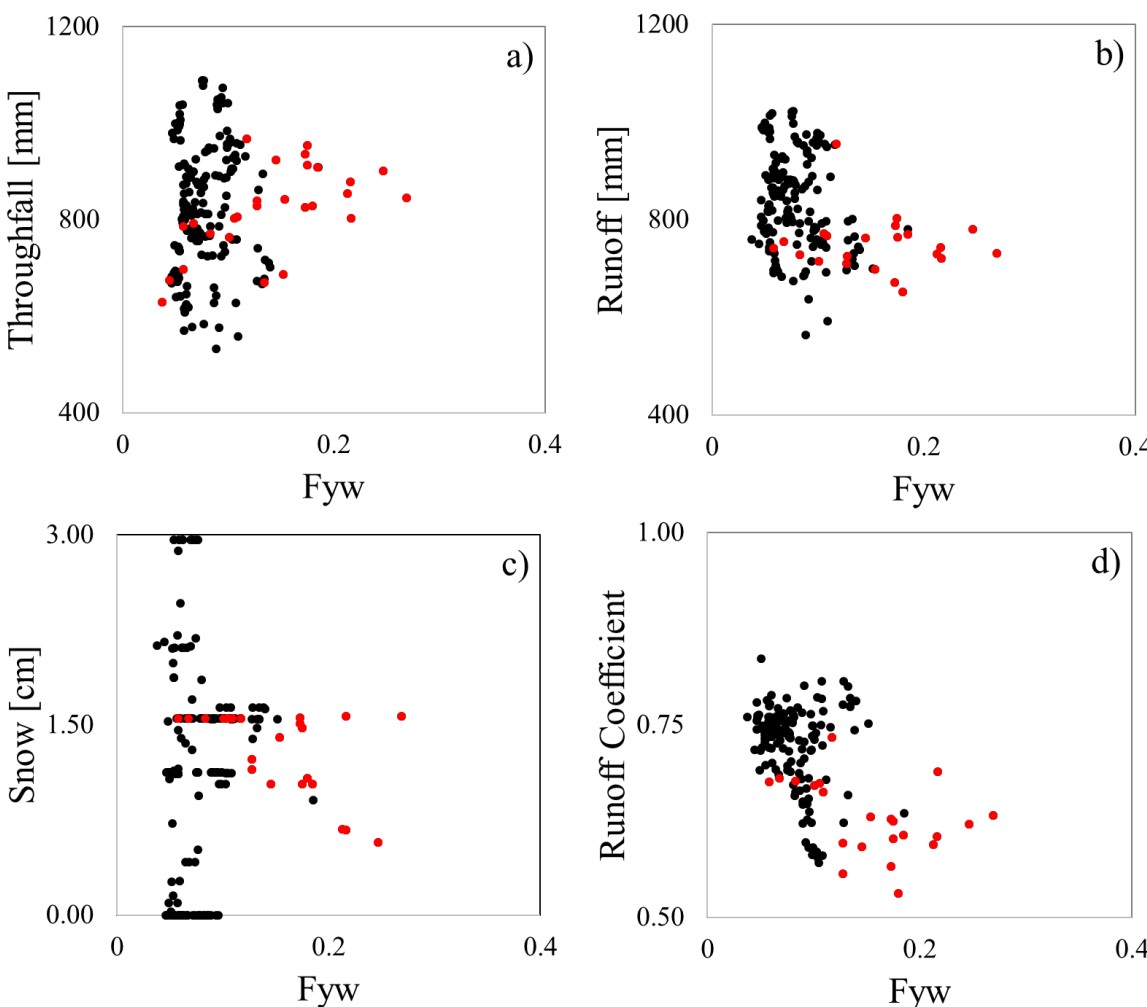

**Figure 7. Fyw plotted against hydrometric data (red and black dots): a) throughfall volumes, b) runoff volumes, c) snow height, d) the runoff coefficient. Red dots are data points where hypothesis 2 was rejected (Fyw does not differ more than ±0.04 within four consecutive weeks).**

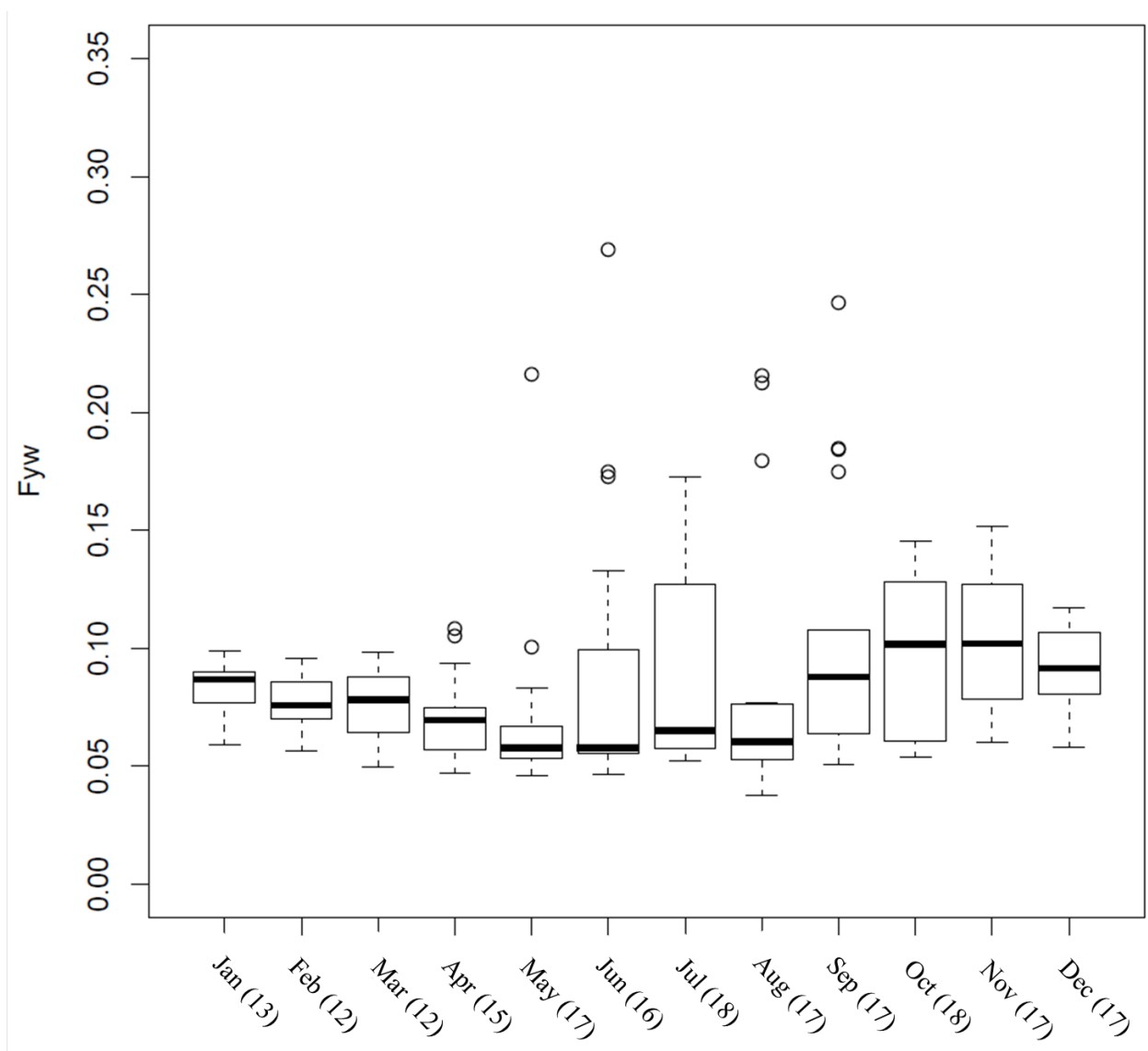

**Figure 8. Testing hypothesis 3 (Fyw centred around a specific month does not differ more than ±0.04 within this month): Boxplot of all Fyw results of a specific month. Whiskers are the upper and lower 1.5 interquartile range and circles are outlier values. The number of data points for each month is given in the brackets on the horizontal axis.**

