# Peer review of "Time-variability and uncertainty of the fraction of young water in a small headwater catchment"

_Hydrology and Earth System Sciences, 2018_

## Referee Comment (RC1) · Anonymous Referee #1 · 28 Jan 2019

This paper presents a numerical experiment to estimate the relative influence of the different sampling periods in the estimate of the fraction of young water (fyw). The authors used 1-year long subsets of precipitation and stream tracer data sampled sequentially over a 4.5-year long record. This resulted in 189 different to estimate fyw based on sine function fits. I find the paper interesting as this approach to estimate the event water fraction is becoming popular among hydrologist. However I dough this paper provides information useful outside the catchment where the data was collected. The authors made no case on how these findings would be relevant to other locations. As such, it reads like a case study. Therefore, I suggest this paper not be consider for publication in HESS in its present form. In addition, I found the study lacks proper justification for the used of 2% difference in fyw as indicative of a significant difference.

The statistical approach is also somehow vague. For example, it would be important to know how does the r2 fits of the input compare to the r2 fits of the output. This would allow understanding what is driving to low mean r2 values that were observed for some of the results. Other specific comments:

Line 21 (P1): Sentence in poorly worded.

Line 23 (P1): The abstract indicates that they recommend an r2 threshold for future studies. However nowhere in the text, the authors offer any justification for the limit.

Line 6 (P2), Line 15 (P13) and elsewhere: Better to refer "water stable isotopes" rather than "stable isotopes of water"

Line 16 (P3): Indicates that the hypotheses were tested against rules of acceptance that were based on whether differences in Fyw exceeded a threshold value of $\pm 2\%$. A more comprehensive justification for the 2% threshold should be included.

Line 17-18 (P4): Please explain how this precision was estimated. Did you collect duplicate samples?

Line 21 (P4): Did you consider using deuterium instead of 18O?

Line14-16 (P5): It would be interesting to see the distributions or R2 of both fits independently.

Line 1-5 (P6): Since the 2% threshold is mentioned in the introduction this explanation belongs there.

Line 22 (P7): These values are very low. An r2 =-.08 would indicate that a sine wave function is weak to describe the variability of the data.

Line 24-25 (P7): Can you provide information about the range of the r2 of these fits

Line 4-9 (P8): Would this indicate that the sine fit method is not appropriate for much of 2014? How confident can one be of the Fyw estimates when the r2 are below 0.2?

Line 12-14 (P8): It is not clear what is the significance of this clustering of points .

Lie 19 (P8): Considering how skew the data is would it be better to use the median? Also I suggesting some standard deviation or standard error .

Line 20 (P8) Please consider some measure of error or uncertainty in the fyw estimates.

Line 23-345 (P8): Please elaborate, that is indicate how many of the 189 were between this ranges.

Line 29 (P8): Please provide some statistical information about the strength of the correlation.

Line 10 (P9): This is confusing about figure 9. Are these the 189 fits? That is, are these fits over a one-year duration time series?

Line17 (P9): Where is the value of d18O for ground water coming from?

Line 19-20 (P9): please elaborate some more in the parallel to the Weigand et al. [2017] study

Line 18-19 P13): How do we know this conclusion is relevant to other catchments?

Line 24-25 (P 13): This sentence is vague. Please explain.

Line 26-28 (P13) It would be important to understand if the variability observed here would be relevant to understand difference across catchments.

Figure 1: The markers for the precipitation and runoff gauges are too similar. Add latitude and longitude grids to the map. The contour should be in the legend indicating the units. In addition, the font next to the contours is difficult to read.

Figure 4: In the legend, please clarify that "mean" refers to

Figure 7: Please add a legend. Figure 8: The caption should include an explanation of what is hypothesis 2

---

## Referee Comment (RC2) · Anonymous Referee #2 · 30 Jan 2019

This manuscript investigates the temporal variability of the young water fractions (Fyw) based on a 4.5-year time series of $\delta$18O in precipitation and streamwater in the German Weiherbach catchment. For this, the authors fit sine curves to the entire 4.5-year data set to estimate the long-term average Fyw. Then, they cut out 189 individual 1-year $\delta$18O time series from the 4.5-year data set (i.e., shifting each 1-year period by 1 week) and fit individual sine curves to these 1-year periods to estimate 189 Fyw-values. The goodness-of-fit of the sine curves to the $\delta$18O data was quantified through adjusted R2 values. Three hypotheses were tested: "(1) Fyw estimates do not change over time (time-invariance) (2) Short-term changes in the start of a tracer sampling campaign do not influence the Fyw estimate (sampling-invariance) (3) Fyw estimates are similar for a given calendar month of different years (seasonal-invariance)" (P3L13-15). By

applying a Fyw-threshold value of 2%, the authors reject all hypotheses and conclude that Fyw-values based on 1-year isotope data sets can be highly variable over time. This time-variably of Fyw hampers catchment-comparison studies that utilize tracer data of different time series lengths or time periods.

I find the critical evaluation of new metrics (young water fraction Fyw) interesting and useful as this allows to better plan sampling campaigns or to use existing data sets more efficiently for a robust estimation of Fyw. In that regard, I consider the testing of hypothesis 2 the most useful scientific contribution of this study (Sect. 3.4) because it quantifies how much a 4-weeks delay can change the Fyw-values that are estimated from 1-year data sets. Unfortunately, these changes in Fyw do not correlate with any of the tested hydro-meteorological variables (Sect. 4.4) so that the authors cannot provide any suggestions about the optimal sampling strategy.

Besides the testing of hypothesis 2, I find it difficult to identify a clear motivation and the novel scientific contribution of this study. The fact that Fyw responds to changes in precipitation, discharge and/or catchment wetness has already been established (e.g., Kirchner, 2016b; Lutz et al., 2018; von Freyberg et al., 2018; Wilusz et al., 2017), and thus it can be expected that Fyw changes over time in a catchment with a variable hydro-climatic regime such as the Weiherbach. Thus, the temporal changes in Fyw that have been identified in the present study are likely related to the hydro-climatic conditions at the site, however, the scientific analysis of these relationships often remains too superficial (which is surprising, given that the Weiherbach catchment is an intensively-studied research site). As such, this study does not teach us something new about the catchment but rather shows that different tracer time series provide different young water fractions. Although it is interesting to quantify these temporal differences in Fyw, it remains to be tested how these findings for the 0.385km2 Weiherbach catchment are transferable to other landscapes and climates.

Major comments:

1. One of my largest concern is that the presented analysis did not provide any information on the uncertainties of the individual Fyw estimates. Only the adjusted R2 values of the sine fits are presented. Previous studies that calculated young water fractions for several other catchment reported uncertainties in Fyw between 1% and 41% (e.g., Jasechko et al., 2016; Stockinger et al., 2016; von Freyberg et al., 2018). Thus, it should be tested whether the individual young water fractions that were calculated from the 1-year time series are indeed statistically significantly different from each other when their uncertainties are considered. Looking at the low adjusted R2 values for the July 2014-October 2015 period (e.g. Figure 4), I would expect the uncertainties of the 1-year Fyw values to be rather large. However, instead of analyzing the uncertainties in Fyw, the authors mainly focus on the time-variability of the individual 1-year Fyw-values and conclude rather boldly (P12L22) "The obtained Fyw could be a potential outlier, a larger value or part of the Fyw baseline". I would argue that the uncertainty in Fyw (e.g., expressed as standard error) would allow us to objectively judge whether we can believe our Fyw estimates or not. Such an analysis is, however, missing here. In fact, knowing the uncertainties of the individual 1-year Fyw values would allow a more informative analysis of how the Fyw-uncertainty (not Fyw itself) is controlled by hydro-climatic conditions. Such an analysis might provide concrete guidelines for planning targeted sampling campaigns to robustly estimate Fyw.

2. Furthermore, given that the uncertainties in Fyw values can potentially be much larger than 2% (as it was shown in the previous studies cited above), to me the 2% threshold seems too low and the authors' justification for that 2% threshold is not convincing.

3. The hydro-meteorological conditions in the Weiherbach catchment were highly variable during the 4.5-year study period. For instance, only the winter 2013/2014 was snow-free in contrast to the other winters when a snowpack built up (Sect. 3.1). In addition, 21% ". . .of the forest were clear-cut in August/September 2013. . ." (P3L27), which significantly altered the streamflow regime of the Weiherbach creek (Wiekenkamp et

al., 2016). In Fig. 8d (P24) we find that the runoff coefficients for the Weiherbach catchment ranged between roughly 0.8 and 1.25, suggesting that hydro-climatic conditions at the site varied considerably over time. The authors do not, however, provide any data or figures that present the hydro-climatic conditions during the study period except for the scatter plots in Figure 8, which contrasts 1-year averages of four hydro-climatic metrics with the respective Fyw-values. Despite the highly variable streamflow regime of the catchment and the authors citing another study where flow weighting of the streamwater isotope values resulted in "...significant changes in Fyw..." (P2L4), the authors should more thoroughly investigate how catchment wetness might control Fyw. Why was streamflow-weighting not done here? Why was there no further analysis of potential factors that may control the large variability in 1-year Fyw values, particularly in the period July 2014-October 2015? It seems likely, that individual storm events may have had strong effects on the discharge of young water, so it may be useful to investigate extreme events rather than average behavior.

4. Sect. 3.5 and Figure 9: It is not clear to me how the Fyw values for testing hypothesis 3 (seasonal invariance) were determined. As far as I understood, Fyw-values were calculated for 189 1-year periods (Sect. 2.3). How were month-specific Fyw-values extracted from these annual Fyw-values? Wouldn't each 1-year Fyw-value be affected by the isotope values of all 12 months that comprise this 1-year period? If so, I doubt that the analysis presented in Sect. 3.5 and Figure 9 provides useful information.

5. Part of the analysis presented in "4.2 Fraction of young water" is not valid. First, the authors calculated Fyw from the entire 4.5-year data set (Fyw,4.5=10.8%) and compared this to the average of the 186 1-year Fyw values (9.3%), concluding that both values are similar with regard to their 2% threshold. A second comparison was carried out with Fyw,4.5 and the average of a much smaller number of 1-year Fyw values that neglects the Fyw values from the period July 2014-October 2015 (7.5%). This second comparison should, however, use another Fyw value as a reference based on the same isotope data set (i.e., 4.5 years minus the period July 2014-October 2015)

- otherwise the authors compare apples with oranges.

6. At the very end of the Discussion section the authors state that a previous analysis has been carried out that used a 3-year isotope times series from the Weiherbach catchment. This previous study already showed that the Fyw values differed substantially between three 1-year periods (Stockinger et al., 2017, data in the supplement). In the present study, the authors simply repeat this analysis with a 4.5-year isotope data set from the same site knowing that their hypothesis "(1) Fyw does not deviate more than ±2% from the mean of all Fyw results indicating long-term invariance [. . .]" will likely be rejected. I was surprised to read about a very similar previous study at the very end of the current manuscript and wondered why is it necessary to repeat the analysis when the result (rejection of hypothesis 1) is already known?

Minor comments:

P3L8-9: "However, it remains to be tested how sensitive the Fyw method is towards the timing and the length of the available data." Why does this need to be tested? Can you provide an example of where the length and the timing of the isotope data resulted in different Fyw values? Otherwise, a clear motivation for your analysis is missing.

P4L25: "Because of this on average 43 isotope values were available for precipitation compared to 53 values for streamflow." Does this average refer to a 1-year period? Please clarify. It would also be nice to provide the total number of streamwater and precipitation samples of the entire 4.5-year period.

P5L22-23: I would suggest to move these two sentences to the beginning of the chapter to make clear where the number "189" comes from.

P5L13: 24*365.25 is 8766 not 1/8766

P5L31-32: What do you mean with "the timing of peaks and the individual amplitudes"? Do you refer to the isotope time series or to the fitted sine functions?

P6L3: Here you switch units of Fyw (0.02 and 2%). Also, in the text you express Fyw

in percent, whereas in the figures you use the scale from zero to one. Please be consistent throughout the manuscript.

P7L13-14: Please be more specific about what water isotopes you are talking about, e.g. add $\delta$18O.

P8L28-30: Please provide some metrics for the strength of these correlations (e.g., Pearson correlation coefficients).

P8L29: Was the runoff coefficient calculated with catchment-average precipitation or throughfall? I would suggest to add the runoff coefficients to Fig. 3 since the relationship between Q/P and the sine wave fits to the isotope data are discussed in Sect. 4.1.

P9L16-20: You suddenly present groundwater isotope data without providing information about the source (location, sampling procedure, number of samples etc.) of these data. Please include this information into Sect. 2.2.

P10L13: "The double-peak in precipitation of autumn 2015 was not found in streamflow (Figure 3)." Do you refer to the $\delta$18O in precipitation and streamflow or to the sine fits to the isotope data?

P11L33: "Thus, during the 4.5-years Fyw never fell below the baseline of 5% [. . .]" This statement is incorrect. Figures 6 and 7 clearly show that Fyw fell below 5% on several occasions, such as around June 2014 and September 2016.

P12L5: "The variability in Fyw of this study could not be explained by most meteorological or hydrometric variables". Could a lack of correlation be explained by the large distance (3km) of the meteorological station to the study site? What about median values of the hydro-climatic variables or metrics that describe extreme events?

P12L9: ". . . the different sampling periods of all mentioned studies. . .". This contradicts a previous statement: ". . . Lutz et al. [2018] used the same sampling period for precipitation and streamflow for all 24 investigated catchments." (P11L25).

P12L23: "As the violation of hypothesis 2 did not correlate with any meteorological or hydrometric data ...". How can a violation correlate with anything? Please clarify.

Figures: The date formats in all figures are confusing. Does 4/10/13 mean 4th October 2013 or 10 April 2013? Also, I would suggest to have each tick mark at the first of the month and to have consistent date axes in all figures.

Figure 4: This figure misses a proper legend (e.g., What does "Mean" stand for?). The unit and numbers of Fyw on the right vertical axes don't match. Do panels a and b share the same legend? Why are the shown time series much shorter than 4.5 years?

References:

Jasechko, S., Kirchner, J. W., Welker, J. M., and McDonnell, J. J.: Substantial proportion of global streamflow less than three months old, Nature Geoscience, 9, 126-129, 10.1038/Ngeo2636, 2016.

Kirchner, J. W.: Aggregation in environmental systems-Part 1: Seasonal tracer cycles quantify young water fractions, but not mean transit times, in spatially heterogeneous catchments, Hydrol. Earth Syst. Sci., 20, 279-297, 10.5194/hess-20-279-2016, 2016a.

Kirchner, J. W.: Aggregation in environmental systems-Part 2: Catchment mean transit times and young water fractions under hydrologic nonstationarity, Hydrol. Earth Syst. Sci., 20, 299-328, 10.5194/hess-20-299-2016, 2016b.

Lutz, S. R., Krieg, R., Müller, C., Zink, M., Knöller, K., Samaniego, L., and Merz, R.: Spatial Patterns of Water Age: Using Young Water Fractions to Improve the Characterization of Transit Times in Contrasting Catchments, Water Resour. Res., 54, 4767-4784, 10.1029/2017WR022216, 2018.

Stockinger, M. P., Bogena, H. R., Lücke, A., Diekkrüger, B., Cornelissen, T., and Vereecken, H.: Tracer sampling frequency influences estimates of young water fraction and streamwater transit time distribution, Journal of Hydrology, 541, Part B, 952-964, http://dx.doi.org/10.1016/j.jhydrol.2016.08.007, 2016.

[Figure]

Stockinger, M. P., Lücke, A., Vereecken, H., and Bogena, H. R.: Accounting for seasonal isotopic patterns of forest canopy intercepted precipitation in streamflow modeling, Journal of Hydrology, 555, 31-40, https://doi.org/10.1016/j.jhydrol.2017.10.003, 2017.

von Freyberg, J., Allen, S. T., Seeger, S., Weiler, M., and Kirchner, J. W.: Sensitivity of young water fractions to hydro-climatic forcing and landscape properties across 22 Swiss catchments, Hydrol. Earth Syst. Sci., 22, 3841-3861, https://doi.org/10.5194/hess-22-3841-2018, 2018.

Wilusz, D. C., Harman, C. J., and Ball, W. P.: Sensitivity of Catchment Transit Times to Rainfall Variability Under Present and Future Climates, Water Resour. Res., 53, 10231-10256, 10.1002/2017WR020894, 2017.

Wiekenkamp, I., Huisman, J.A., Bogena, H., Graf, A., Lin, H., Drüe, C., and Vereecken, H.: Changes in Spatiotemporal Patterns of Hydrological Response after Partial Deforestation. J. Hydrol. 542: 648-661, doi:10.1016/j.jhydrol.2016.09.037, 2016.

---

## Referee Comment (RC3) · Anonymous Referee #3 · 9 Feb 2019

Stockinger et al. presents a study to evaluate the temporal variability of young water fraction (Fyw) based on 189 sine curve fits of 1-year subsets of a 4.5-year rainwater and streamwater 18-O isotope dataset. The Fyw, developed by Kirchner 2016, has become a powerful descriptor of streamwater flow path as the substitute for mean transit time. It is important to test the how Fyw change with different timing and sampling time coverage of water isotopes. The results showed "high" temporal variability of Fyw but no seasonality of Fyw based on the criterion defined by the author. The variability due to sampling time chosen is very useful for the isotope hydrology community. This study sheds new light on the development and application of Fyw, which is interesting and suitable for HESS. I find this paper is generally well-written but not strong enough. One of my concern is that how and why the 2% difference was defined as significant for

the three hypotheses? This threshold value is introduced in the paper but not clearly explained. The discussion section lacks the discussion of the importance of the results. It would be a stronger paper if the author can explain the cause of the Fyw time-variability, which is ambiguous in the current form. Other specific comments on this paper are listed below.

P3-L27: change "8" to Eight

P5-L14: Add "reciprocal of" or similar phrase before "24 hours..." since frequency (f) should be 1/T.

Figure 1: It would be nice to add latitude and longitude to the map. An alternative way is giving the latitude and longitude of the sampling location in the text. Square brackets with "-" can be removed, it may be misread as minus.

Figure 4a: Which line represent the R2?

Figure 7, 8, and 9: The hypotheses should be explained in the captions.

---

## Author Comment (AC1) · 18 Feb 2019

This paper presents a numerical experiment to estimate the relative influence of the different sampling periods in the estimate of the fraction of young water (fyw). The authors used 1-year long subsets of precipitation and stream tracer data sampled sequentially over a 4.5-year long record. This resulted in 189 different to estimate fyw based on sine function fits. I find the paper interesting as this approach to estimate the event water fraction is becoming popular among hydrologist. However I dough this paper provides information useful outside the catchment where the data was collected. The authors made no case on how these findings would be relevant to other locations. As such, it reads like a case study. Therefore, I suggest this paper not be consider for publication in HESS in its present form. In addition, I found the study lacks proper justification for the used of 2% difference in fyw as indicative of a significant difference.

We thank reviewer #1 for the helpful comments.

Usefulness to other catchments
The main aim of this study is to present a *generic method* to analyze the time-variance of the fraction of young water. The reviewer already mentioned that Fyw is becoming more popular. Still, we lack information on when to best use this method and how sensitive it is to different datasets (e.g. frequency of sampling and length of observation time). Thus, investigating its use, limits and pitfalls is very important before we apply it to any catchment and particularly when comparing results from different catchments. Many catchment studies showed that the transit time of water strongly varies (e.g. Harman, 2015; Heidbüchel et al., 2013), and it is thus very likely that Fyw also varies in other locations than ours.
While previous studies focused on hydroclimatic and methodological influences on Fyw, this study is the first to focus on the influence of the sampling period and length. This is a first step, and it is highly recommended that this is repeated in other catchments to assess if this is a general situation or only a few catchments have time-varying Fyw (which we doubt because of strongly varying transit times in general).
Ultimately, catchment comparison studies that rely on Fyw should be based on comparable Fyw results. For example, Stockinger et al. 2016 already showed that only changing the sampling frequency of isotopes data led to drastically changed Fyw results. The present study goes further and shows that also the sampling period can influence Fyw. Based on reviewer comments, we now also present the associated uncertainties. Information on the various influences on Fyw results is critical for catchment comparison studies. We encourage hydrologists to use our generic method to test the existence of strong time-variances of Fyw in other catchments. However, the application of this method to a large set of catchments is beyond the scope of this study.

Additionally, we now conducted a preliminary extended analysis on possible hydro-climatological influences on Fyw and its uncertainty to further foster transferability to other catchments. The (hydrologic, meteorological and isotopic) data suggests that unique meteorological conditions might have influenced the Fyw estimate.

We will expand the revised manuscript with these discussion points.

Using 2% as difference
We now applied Gauß error propagation to estimate uncertainty in Fyw and will use it to derive a data-driven threshold instead of 2% difference (preliminary Figure R1):

[Figure]

**Figure R1**. 189 Fyw results (black) and uncertainty (grey) compared to Fyw for all data (red, solid line) and respective uncertainty (red, dashed line). Additionally plotted is the adjusted R² (blue).

The following can be said from this result:

a) with a drop in R² below approx. 0.2 the uncertainty increases drastically. This, together with the strongly fluctuating Fyw results (page 11, lines 6-8), indicates that in the Wüstebach an R² of at least 0.2 should be reached. We highly recommend conducting similar studies in different catchments to test whether different R² threshold values exist in other catchments.
b) Fyw of all data ("Fyw all" in Figure R1) had an uncertainty of appr. ±4%. We will use this new data-driven value instead of the ±2% for re-evaluating our hypotheses.

Thus, we will use a threshold value based on the information contained in our data instead of a value chosen from studies of different catchments (i.e., ±2% taken from Lutz et al., 2018).

The statistical approach is also somehow vague. For example, it would be important to know how does the r2 fits of the input compare to the r2 fits of the output. This would allow understanding what is driving to low mean r2 values that were observed for some of the results.
A comparison of input and output R² fits is shown in Figure 4: the mean R², as well as R² of throughfall (input to the catchment, TF in Figure 4a) and R² of streamflow (output from the catchment, Q in Figure 4a). The sentence of page 8, line 4 might have been misleading. We suggest the following new sentence:
"The mean adjusted R² is the arithmetic mean of the precipitation and the streamflow adjusted R² values of the respective sine waves. It showed a marked decrease[…]"

As both R² for TF and Q drop significantly (Figure 4a), we assumed that the largest influence on the low R² values were the low amplitudes of the sine waves. A comparison of TF and Q amplitudes is shown in Figure 4b, where low TF amplitudes seem to drive low R² values.

In the manuscript we only compared the mean amplitude. We will add a discussion about the suggested influence of TF and Q amplitudes on low $R^2$.

Other specific comments:
Line 21 (P1): Sentence in poorly worded.
Based on the new uncertainty analysis, we suggest:

"Our results showed a high short-term variability and increase in uncertainty of Fyw when the mean adjusted $R^2$ was below 0.2. Consequently, a low $R^2$ indicated highly uncertain Fyw results"

Line 23 (P1): The abstract indicates that they recommend an r2 threshold for future studies. However nowhere in the text, the authors offer any justification for the limit.
The added uncertainty estimate of Fyw supports a certain threshold of quality for the fit. Very low goodness-of-fit values increase the uncertainty (as expected). Otherwise, $R^2$ values close to 0 would also be accepted and the respective Fyw results accepted as they are.

In a revised version we would expand on the currently existing discussion (page 11, line 11. Added text in **bold**):

"[…], we assumed that **in our case** the Fyw calculation method reached its limit below an average adjusted $R^2$ = 0.2. Fyw became highly sensitive to a small change in input data and highly uncertain. Additional investigations on the sensitivity of Fyw to the goodness-of-fit (not necessarily only measured with adjusted $R^2$) are subject to future studies. **It remains to be seen if a value of 0.2 for adjusted $R^2$ is a critical threshold value for Fyw or if other studies in different catchments show varying results.**"

Line 6 (P2), Line 15 (P13) and elsewhere: Better to refer "water stable isotopes" rather than "stable isotopes of water"
We will change this.

Line 16 (P3): Indicates that the hypotheses were tested against rules of acceptance that were based on whether differences in Fyw exceeded a threshold value of ± 2%. A more comprehensive justification for the 2% threshold should be included.
We will change this to the data-based 4% (see answer above)

Line 17-18 (P4): Please explain how this precision was estimated. Did you collect duplicate samples?
This is the long-term precision derived from the uncertainty of 10,000s of measurements of various water samples conducted during the last years. Each unique sample is measured 6 times.

Line 21 (P4): Did you consider using deuterium instead of 18O?
D and 18O are strongly correlated ($R^2$ = 0.98) so we did not consider using it.

Line14-16 (P5): It would be interesting to see the distributions or R2 of both fits independently.
$R^2$ is shown in Figure 4a for both throughfall and streamflow (orange lines labeled TF and Q).

Line 1-5 (P6): Since the 2% threshold is mentioned in the introduction this explanation belongs there.
We will move the suggested sentences and adapt them to the new threshold value.

Line 22 (P7): These values are very low. An r2 =-.08 would indicate that a sine wave function is weak to describe the variability of the data.
The low $R^2$ mentioned are values for the single sine wave fit to the full 4.5-year time series and are 0.08 for TF and 0.2 for Q. We fully agree that those sine wave functions are gross simplification of the

inter-annual variability of isotopes; which is the point of this study: a single sine wave fit oversimplifies naturally occurring, annual variations. However, even if a sine wave is weak to describe the data, the Fyw calculation method is based on using the amplitudes of sine wave functions (Kirchner, 2016). We will add this discussion in the revised version.

A quick way of assessing inter-annual variability is to use the moving average over isotope values. The following two figures show the moving average (12 week interval, not weighed, thick black line) plotted over the single sine wave fit and the 189 fits. The 189 time-variable sine waves are better suited (but not perfect) to capture the inter-annual variability than the single sine wave. We do not intend of using these figures in a revised manuscript version.

[Figure]

**Figure R2**. 12-week, non-weighed moving average over the throughfall isotope data, compared to the 189 1-year sine waves (red) and the single 4.5-years sine wave (green).

[Figure]

**Figure R3**. 12-week, non-weighed moving average over the streamflow isotope data, compared to the 189 1-year sine waves (red) and the single 4.5-years sine wave (green).

Line 24-25 (P7): Can you provide information about the range of the r2 of these fits

We will add:
Precipitation min, max and mean R²:  -0.03, 0.63, 0.21
Streamflow min, max and mean R²:  0.00, 0.55, 0.25

Line 4-9 (P8): Would this indicate that the sine fit method is not appropriate for much of 2014? How confident can one be of the Fyw estimates when the r2 are below 0.2?
We added the uncertainty estimate (see above, Figure R1) and Fyw indeed becomes highly unreliable in this period. We encourage studies of the reliability of Fyw based on goodness-of-fit measures of the sine waves.

Line 12-14 (P8): It is not clear what is the significance of this clustering of points.
We will remove this sentence, as this was just an observation on our part.

Line 19 (P8): Considering how skew the data is would it be better to use the median? Also I suggesting some standard deviation or standard error.
The median value is approximately 8.0% Fyw (complete data set, including the low R² period). We will add the median as another measure in the text.

We will incorporate the error estimates obtained by Gauß error propagation.

Line 20 (P8) Please consider some measure of error or uncertainty in the fyw estimates.
Please see the answer and new Figure R1 above.

Line 23-345 (P8): Please elaborate, that is indicate how many of the 189 were between this ranges.
Line 20 mentions that 63 results from 189 were within the range. The reduced data set of Fyw (leaving out the low R²) has a total of 124 Fyw values, with 66 being in the range (53%). We will mention those numbers in the manuscript.

Line 29 (P8): Please provide some statistical information about the strength of the correlation.
The equation for the whole data set (including the low R²) is
Runoff Coefficient = -1.24 * Fyw + 1.13, with an adjusted R² of 0.30 and p-value of 3E-16

The equation for the limited data set (low R² excluded) is
Runoff Coefficient = -2.11 * Fyw + 1.19, with an adjusted R² of 0.23 and p-value of 2E-7

We will add this information to the revised text.

Line 10 (P9): This is confusing about figure 9. Are these the 189 fits? That is, are these fits over a one-year duration time series?
We agree that the sentence is confusing. These are indeed the 189 fits.
We suggest adapting the sentence "As mentioned in the methods, the Fyw results were put in the middle of the one-year calculation period (calculating from February 2016 to February 2017, the result would be displayed as a data point in August 2016). We grouped together all Fyw results that were assigned to a specific calendar month to detect possible seasonality."

Line17 (P9): Where is the value of d18O for ground water coming from?
The Wüstebach catchment is extensively monitored and groundwater d18O data is available. We did not use this data at first but will add it to the revised version.

Line 19-20 (P9): please elaborate some more in the parallel to the Weigand et al. [2017] study
We extended the sentence:

"The study by Weigand et al. [2017] came to the same conclusion for the Wüstebach catchment using wavelet analysis of nitrate and DOC data collected at mainstream and tributary locations. While lower altitude locations of the catchment near the outlet were dominated by groundwater, higher altitude areas were less affected. This finding was additionally supported by field observations of shallow groundwater."

Line 18-19 P13): How do we know this conclusion is relevant to other catchments?
We expect that changes in flow paths over time alter the Fyw result of a catchment. A one-year long time series of tracer data might lead to very different Fyw results depending on when the tracer data was sampled. Thus, the results of the comparison study could vary greatly simply by shifting the one-year sampling window by a couple of weeks. We suggest using the ensemble of Fyw values and their distribution to get an additional uncertainty estimate of Fyw. We'll add this information to the revised manuscript.

Please see also our answer to the major comment regarding usefulness of our results for other catchments.

Line 24-25 (P 13): This sentence is vague. Please explain.
We adapted the sentence to make it clearer:

"If feasible, we recommend investigating a multi-year time series of tracer data with the method suggested in this study. That is, to use a one-year moving time window and estimate an ensemble of Fyw results to derive its uncertainty."

Line 26-28 (P13) It would be important to understand if the variability observed here would be relevant to understand difference across catchments.
In a revised version we will phrase this more clearly.
As discussed above, different sampling periods yield (occasionally very) different Fyw result that would influence the interpretation of catchment comparison studies. Please also refer to the discussion points above regarding the usefulness to other catchments.

Figure 1: The markers for the precipitation and runoff gauges are too similar. Add latitude and longitude grids to the map. The contour should be in the legend indicating the units. In addition, the font next to the contours is difficult to read.
We will adapt the figure accordingly.

Figure 4: In the legend, please clarify that "mean" refers to
We will change it to "Mean R²", "TF R²" and "Q R²" to clarify.

Figure 7: Please add a legend.
We will do this.

Figure 8: The caption should include an explanation of what is hypothesis 2
We will add the explanation.

References
Harman, C. J. (2015), Time-variable transit time distributions and transport: Theory and application to storage-dependent transport of chloride in a watershed, Water Resour. Res., 51, doi:10.1002/2014WR015707.

Heidbüchel, I., P. A. Troch, and S. W. Lyon (2013), Separating physical and meteorological controls of variable transit times in zero-order catchments, Water Resour. Res., 49, 7644–7657, doi:10.1002/2012WR013149.

Kirchner, J. W.: Aggregation in environmental systems - Part 1: Seasonal tracer cycles quantify young water fractions, but not mean transit times, in spatially heterogeneous catchments, Hydrol Earth Syst Sc, 20(1), 279-297, 2016.

Lutz, S. R., Krieg, R., Müller, C., Zink, M., Knöller, K., Samaniego, L., and Merz, R.: Spatial patterns of water age: Using young water fractions to improve the characterization of transit times in contrasting catchments. Water Resources Research,54, 4767–4784.https://doi.org/10.1029/2017WR022216, 2018.

Stockinger, M.P., H.R. Bogena, A. Lücke, B. Diekkrüger, T. Cornelissen and H. Vereecken (2016): Tracer sampling frequency influences estimates of young water fraction and streamwater transit time distribution. J. Hydrol. 541: 952-964, doi:10.1016/j.jhydrol.2016.08.007.

Weigand, S., R. Bol, B. Reichert, A. Graf, I. Wiekenkamp, M. Stockinger, A. Lücke, W. Tappe, H. Bogena, T. Pütz, W. Amelung and H. Vereecken (2017): Spatiotemporal dependency of dissolved organic carbon to nitrate in stream- and groundwater of a humid forested catchment – a wavelet transform coherence analysis. Vadose Zone J. 16(3), doi:10.2136/vzj2016.09.0077.

---

## Author Comment (AC2) · 18 Feb 2019

This manuscript investigates the temporal variability of the young water fractions (Fyw) based on a 4.5-year time series of $\delta18O$ in precipitation and streamwater in the German Weiherbach catchment. For this, the authors fit sine curves to the entire 4.5-year data set to estimate the long-term average Fyw. Then, they cut out 189 individual 1-year 18O time series from the 4.5-year data set (i.e., shifting each 1-year period by 1 week) and fit individual sine curves to these 1-year periods to estimate 189 Fyw-values. The goodness-of-fit of the sine curves to the $\delta18O$ data was quantified through adjusted R2 values. Three hypotheses were tested: "(1) Fyw estimates do not change over time (time-invariance) (2) Short-term changes in the start of a tracer sampling campaign do not influence the Fyw estimate (sampling-invariance) (3) Fyw estimates are similar for a given calendar month of different years (seasonal-invariance)" (P3L13-15). By applying a Fyw-threshold value of 2%, the authors reject all hypotheses and conclude that Fyw-values based on 1-year isotope data sets can be highly variable over time. This time-variably of Fyw hampers catchment-comparison studies that utilize tracer data of different time series lengths or time periods. I find the critical evaluation of new metrics (young water fraction Fyw) interesting and useful as this allows to better plan sampling campaigns or to use existing data sets more efficiently for a robust estimation of Fyw. In that regard, I consider the testing of hypothesis 2 the most useful scientific contribution of this study (Sect. 3.4) because it quantifies how much a 4-weeks delay can change the Fyw-values that are estimated from 1-year data sets. Unfortunately, these changes in Fyw do not correlate with any of the tested hydro-meteorological variables (Sect. 4.4) so that the authors cannot provide any suggestions about the optimal sampling strategy.

We thank reviewer #2 for evaluating our study and the helpful comments.

We now conducted an extended analysis of hydro-meteorological data and it points to possible unique meteorological conditions during summer 2015 that might have influenced Fyw. Please also refer to our answer given to the next reviewer comment:

Besides the testing of hypothesis 2, I find it difficult to identify a clear motivation and the novel scientific contribution of this study. The fact that Fyw responds to changes in precipitation, discharge and/or catchment wetness has already been established (e.g., Kirchner, 2016b; Lutz et al., 2018; von Freyberg et al., 2018; Wilusz et al., 2017), and thus it can be expected that Fyw changes over time in a catchment with a variable hydro-climatic regime such as the Weiherbach. Thus, the temporal changes in Fyw that have been identified in the present study are likely related to the hydro-climatic conditions at the site, however, the scientific analysis of these relationships often remains too superficial (which is surprising, given that the Weiherbach catchment is an intensively studied research site). As such, this study does not teach us something new about the catchment but rather shows that different tracer time series provide different young water fractions. Although it is interesting to quantify these temporal differences in Fyw, it remains to be tested how these findings for the 0.385km2 Weiherbach catchment are transferable to other landscapes and climates.

First, to avoid any misunderstandings we would like to make clear that our study area is the Wüstebach catchment and not the Weiherbach catchment as indicated by the reviewer. The primary focus of this study was not to identify catchment influences on Fyw but to investigate the extent and significance of temporal Fyw changes using variable 1-year time series of isotope tracer data. We primarily aim to improve the robustness of the Fyw method and not to analyze specific Wüstebach influences on Fyw. This is essential information for planning future sampling strategies in other catchments and for the layout of catchment comparison studies.

We extended the analysis of hydro-meteorological data and found evidence of a possible unique meteorological situation in summer 2015. Without going into too much detail, presently several data (hydrologic, meteorological and isotopic) indicate this special situation. The analysis is not yet finalized. The finished version will be incorporated into the manuscript and if possible, we will give guidelines for a more robust Fyw estimation.

Even if in the end no clear recommendation for a sampling time can be given, a recommendation was given: estimating Fyw with data from a single year is not enough (page 12, 25f); the time-variant Fyw for a catchment should be calculated to understand the behavior and uncertainty for a given location (page 13, lines 2ff).

The reviewer already mentioned that previous studies found Fyw reacts to changes in precipitation, discharge, and other factors. Thus it is safe to assume that other catchments also have a time-varying Fyw and applying our method would yield more information about the Fyw behavior and uncertainty in each catchment. We highly suggest conducting the same study for other catchments in other landscape or climatic units to be able to generalize the findings.

We will add these discussion points to a revised version of the manuscript.

Major comments:
We first answer major comment #6 as it is of great importance:

6. At the very end of the Discussion section the authors state that a previous analysis has been carried out that used a 3-year isotope times series from the Weiherbach catchment. This previous study already showed that the Fyw values differed substantially between three 1-year periods (Stockinger et al., 2017, data in the supplement). In the present study, the authors simply repeat this analysis with a 4.5-year isotope data set from the same site knowing that their hypothesis "(1) Fyw does not deviate more than _2% from the mean of all Fyw results indicating long-term invariance [: : :]" will likely be rejected. I was surprised to read about a very similar previous study at the very end of the current manuscript and wondered why is it necessary to repeat the analysis when the result (rejection of hypothesis 1) is already known?

The focus of Stockinger et al., 2017 was to correct canopy-induced isotope changes in throughfall for the complete time series. 3 individual years were cut out as a test without going into further detail. We extended this test into its own study to focus on investigating the extent and dynamics of the time-variance of Fyw, which was not the focus of Stockinger et al., 2017 and would have been out of scope for the previous study.

We had no possibility to know the results beforehand since for hypothesis 1 more than 90% of Fyw results must be within ±2% (page 6, line 10). It is not possible to estimate from 3 out of 189 results if approximately 19 results (10%) would be outside the ±2%.

1. One of my largest concern is that the presented analysis did not provide any information on the uncertainties of the individual Fyw estimates. Only the adjusted R2 values of the sine fits are presented. Previous studies that calculated young water fractions for several other catchment reported uncertainties in Fyw between 1% and 41% (e.g., Jasechko et al., 2016; Stockinger et al., 2016; von Freyberg et al., 2018). Thus, it should be tested whether the individual young water fractions that were calculated from the 1-year time series are indeed statistically significantly different from each other when their uncertainties are considered. Looking at the low adjusted R2 values for the July 2014-October 2015 period (e.g. Figure 4), I would expect the uncertainties of the 1-year Fyw values to be rather large. However, instead of analyzing the uncertainties in Fyw, the authors mainly focus on the time-variability of the individual 1-year Fywvalues and conclude rather boldly (P12L22) "The obtained Fyw could be a potential outlier, a larger value or part of the Fyw baseline". I would argue that the uncertainty in Fyw (e.g., expressed as standard error) would allow us to objectively judge whether we can believe our Fyw estimates or not. Such an analysis is, however, missing here. In fact, knowing the uncertainties of the individual 1-year Fyw values would allow a more informative analysis of how the Fyw-uncertainty (not Fyw itself) is controlled by hydroclimatic conditions. Such an analysis might provide concrete guidelines for planning targeted sampling campaigns to robustly estimate Fyw.

We thank the reviewer for pointing out this very important issue. It is the aim of this study to present a generic method to analyze Fyw for time-variance and thus improve the robustness of the Fyw method. We added uncertainty estimates of Fyw using Gauß error propagation (preliminary Figure R1):

[Figure]

**Figure R1**. 189 Fyw results (black) and uncertainty (grey) compared to Fyw for all data (red, solid line) and respective uncertainty (red, dashed line). Additionally plotted is the adjusted $R^2$ (blue).

The following can be said from this result:

a) with a drop in $R^2$ below approx. 0.2 the uncertainty increases drastically. This, together with the strongly fluctuating Fyw results (page 11, lines 6-8), indicates that in the Wüstebach an $R^2$ of at least 0.2 should be reached. We highly recommend conducting similar studies in different catchments to test whether different $R^2$ threshold values exist in other catchments.
b) Fyw of all data (Fyw all in Figure R1) had an uncertainty of appr. ±4%. We will use this new data-driven value instead of the ±2% for re-evaluating our hypotheses.
c) the Fyw results become highly uncertain during 2014/2015. From our preliminary analysis mentioned above it seems that special meteorological conditions during summer 2015 are responsible.

2. Furthermore, given that the uncertainties in Fyw values can potentially be much larger than 2% (as it was shown in the previous studies cited above), to me the 2% threshold seems too low and the authors' justification for that 2% threshold is not convincing.

We will use 4% based on the uncertainty estimation of using all data, see comment above.

3. The hydro-meteorological conditions in the Weiherbach catchment were highly variable during the 4.5-year study period. For instance, only the winter 2013/2014 was snow-free in contrast to the other

winters when a snowpack built up (Sect. 3.1). In addition, 21% "…of the forest were clear-cut in August/September 2013: : :" (P3L27), which significantly altered the streamflow regime of the Weiherbach creek (Wiekenkamp et al., 2016). In Fig. 8d (P24) we find that the runoff coefficients for the Weiherbach catchment ranged between roughly 0.8 and 1.25, suggesting that hydro-climatic conditions at the site varied considerably over time. The authors do not, however, provide any data or figures that present the hydro-climatic conditions during the study period except for the scatter plots in Figure 8, which contrasts 1-year averages of four hydroclimatic metrics with the respective Fyw-values. Despite the highly variable streamflow regime of the catchment and the authors citing another study where flow weighting of the streamwater isotope values resulted in "…significant changes in Fyw…" (P2L4), the authors should more thoroughly investigate how catchment wetness might control Fyw. Why was streamflow-weighting not done here? Why was there no further analysis of potential factors that may control the large variability in 1-year Fyw values, particularly in the period July 2014-October 2015? It seems likely, that individual storm events may have had strong effects on the discharge of young water, so it may be useful to investigate extreme events rather than average behavior.

In a revised version we will present the hydroclimatic data, including catchment wetness conditions expressed as a mean soil water content, either as a figure or as a supplementary. As mentioned above, we already started a more extensive analysis of hydrometeorological conditions and their possible impact on Fyw and its uncertainty. It is likely that special conditions during summer 2015 influenced Fyw. The investigation is still ongoing.

Precipitation and streamflow weighing were both done for Fyw calculation, but unfortunately not clearly mentioned in the submission.

4. Sect. 3.5 and Figure 9: It is not clear to me how the Fyw values for testing hypothesis 3 (seasonal invariance) were determined. As far as I understood, Fyw-values were calculated for 189 1-year periods (Sect. 2.3). How were month-specific Fyw-values extracted from these annual Fyw-values? Wouldn't each 1-year Fyw-value be affected by the isotope values of all 12 months that comprise this 1-year period? If so, I doubt that the analysis presented in Sect. 3.5 and Figure 9 provides useful information.

No monthly Fyw were extracted from the data. Each of the 189 Fyw results was assigned to the date that lies in the middle of the calculation period. For example, Fyw was calculated from 1 January to 31 December and the corresponding result was assigned to 1 July.

We then grouped all Fyw results according to the month they were assigned to. All results in Fig 9 are still 1-year calculation results. Should a seasonal trend be observable, one could argue that e.g., a 1-year sampling campaign centered around July would lead to higher/lower Fyw estimates compared to when it is centered on March.

5. Part of the analysis presented in "4.2 Fraction of young water" is not valid. First, the authors calculated Fyw from the entire 4.5-year data set (Fyw,4.5=10.8%) and compared this to the average of the 186 1-year Fyw values (9.3%), concluding that both values are similar with regard to their 2% threshold. A second comparison was carried out with Fyw,4.5 and the average of a much smaller number of 1-year Fyw values that neglects the Fyw values from the period July 2014-October 2015 (7.5%). This second comparison should, however, use another Fyw value as a reference based on the same isotope data set (i.e., 4.5 years minus the period July 2014-October 2015)- otherwise the authors compare apples with oranges.

We thank the reviewer for pointing this out. This is absolutely correct and we will change it accordingly.

Minor comments:

P3L8-9: "However, it remains to be tested how sensitive the Fyw method is towards the timing and the length of the available data." Why does this need to be tested? Can you provide an example of where the length and the timing of the isotope data resulted in different Fyw values? Otherwise, a clear motivation for your analysis is missing.

We will cite Stockinger et al., 2017 here (see also answer to major comment #6) where a multi-year Fyw as well as three individual years Fyw were calculated and differences found.

P4L25: "Because of this on average 43 isotope values were available for precipitation compared to 53 values for streamflow." Does this average refer to a 1-year period? Please clarify. It would also be nice to provide the total number of streamwater and precipitation samples of the entire 4.5-year period.

Yes, this refers to the 1-year calculation periods. The total number of P and Q samples (156 and 195, respectively) is mentioned on page 7, line 21 but we will move it here.

P5L22-23: I would suggest to move these two sentences to the beginning of the chapter to make clear where the number "189" comes from.

We agree.

P5L13: 24*365.25 is 8766 not 1/8766

We will rephrase this "i.e., if CP(t) and CS(t) are calculated in hourly time steps then the frequency f is 1/8766; once per 24 x 365.25 hours)."

P5L31-32: What do you mean with "the timing of peaks and the individual amplitudes"? Do you refer to the isotope time series or to the fitted sine functions?

We referred to the fitted sine functions and suggest the following change to the sentence

"We also calculated Fyw for the whole time series with one sine wave and compared its peak timing and amplitude to the timing of peaks and amplitudes of the 189 sine waves."

P6L3: Here you switch units of Fyw (0.02 and 2%). Also, in the text you express Fyw in percent, whereas in the figures you use the scale from zero to one. Please be consistent throughout the manuscript.

We will consistently use e.g., 0.02 instead of 2%.

P7L13-14: Please be more specific about what water isotopes you are talking about, e.g. add $\delta$18O.

Page 7 line 13 already featured "$\delta$18O" at the end of the first sentence:

"Precipitation isotope ratios ranged from -3.04 to -17.80‰, spanning a range of 14.76‰ in $\delta$18O values."

P8L28-30: Please provide some metrics for the strength of these correlations (e.g., Pearson correlation coefficients).

We will add statistical information to the text, e.g.:

The equation for the whole data set (including the low $R^2$) is
Runoff Coefficient = -1.24 * Fyw + 1.13, with an adjusted $R^2$ of 0.30 and p-value of 3E-16

The equation for the limited data set (low R² excluded) is
Runoff Coefficient = -2.11 * Fyw + 1.19, with an adjusted $R^2$ of 0.23 and p-value of 2E-7

P8L29: Was the runoff coefficient calculated with catchment-average precipitation or throughfall? I would suggest to add the runoff coefficients to Fig. 3 since the relationship between Q/P and the sine wave fits to the isotope data are discussed in Sect. 4.1.

The runoff coefficient was calculated with throughfall but we plan to change it to open precipitation to enable comparability to other studies. Since we started an extensive analysis of hydrometeorological data that features the runoff coefficient but is not yet finished (see also mentions above), we will present the runoff coefficient more prominently in the revised manuscript, but not necessarily as part of Figure 3.

P9L16-20: You suddenly present groundwater isotope data without providing information about the source (location, sampling procedure, number of samples etc.) of these data. Please include this information into Sect. 2.2.

This was on oversight on our part and we will add it in a revised manuscript version.

P10L13: "The double-peak in precipitation of autumn 2015 was not found in streamflow (Figure 3)." Do you refer to the $\delta18O$ in precipitation and streamflow or to the sine fits to the isotope data?

To the sine fits, we adapted the sentence:

"The sine wave double-peak in precipitation of autumn 2015 was not found in streamflow (Figure 3)."

P11L33: "Thus, during the 4.5-years Fyw never fell below the baseline of 5% […]" This statement is incorrect. Figures 6 and 7 clearly show that Fyw fell below 5% on several occasions, such as around June 2014 and September 2016.

We will adapt the sentence "Thus, during the 4.5-years Fyw seldom fell below the baseline of 5% […]"

P12L5: "The variability in Fyw of this study could not be explained by most meteorological or hydrometric variables". Could a lack of correlation be explained by the large distance (3km) of the meteorological station to the study site? What about median values of the hydro-climatic variables or metrics that describe extreme events?

Correlations of precipitation amounts ($R^2$ = 0.95), temperature ($R^2$ = 0.99) and relative humidity ($R^2$ = 0.94) of the 3 km distant climate station with the respective climate data from the clear-cut area of the Wüstebach catchment showed good $R^2$. We did not use the on-site climate station for our study as its data does not cover the full study period.

We are currently in the process of re-evaluating and extending our analysis of hydrometeorological data (as mentioned above). Several newly analyzed data point to influencing Fyw during summer 2015. We will incorporate and test median values and extreme event metrics.

P12L9: "…the different sampling periods of all mentioned studies…". This contradicts a previous statement: "…Lutz et al. [2018] used the same sampling period for precipitation and streamflow for all 24 investigated catchments." (P11L25).

We will correct this to "most of the mentioned studies".

P12L23: "As the violation of hypothesis 2 did not correlate with any meteorological or hydrometric data : : :". How can a violation correlate with anything? Please clarify.

We referred here to the timing of the violation of hypothesis 2: if the timing could be connected to hydrometric or meteorological data.

We adapted:

"As the timing of the violation of hypothesis 2 did not[…]"

Figures: The date formats in all figures are confusing. Does 4/10/13 mean 4th October 2013 or 10 April 2013? Also, I would suggest to have each tick mark at the first of the month and to have consistent date axes in all figures.

4/10/13 refers to April 10[th], 2013. We will adapt the figures to uniformly start on 4/1/13 with the exception of Figure 2 (just a theoretical example) and Figure 3 (showing the input data and starting on a different date than the Fyw result figures; see also explanation below).

Figure 4: This figure misses a proper legend (e.g., What does "Mean" stand for?). The unit and numbers of Fyw on the right vertical axes don't match. Do panels a and b share the same legend? Why are the shown time series much shorter than 4.5 years?

We will change the legend entries to "Mean $R^2$", "TF $R^2$" and "Q $R^2$".
We are not sure about the reviewer comment regarding the right axis having mismatching units and numbers for Fyw. The only right-hand axis is in Figure 4b and shows Amplitudes (in permille) and not Fyw (in percent). In a revised version we would remove [%] from the Fyw-axis to avoid the misleading conclusion of "0.3% Fyw".

Panels a and b share the same legend.
The time series are shorter than 4.5 years since each Fyw result was placed in the middle of the year it was calculated for. The time series starts on 10/10/12, thus the first Fyw result is placed on 4/10/13. Doing this cuts off the first half year and the last half year of the complete time series, explaining the shortening.

---

## Author Comment (AC3) · 18 Feb 2019

Stockinger et al. presents a study to evaluate the temporal variability of young water fraction (Fyw) based on 189 sine curve fits of 1-year subsets of a 4.5-year rainwater and streamwater 18-O isotope dataset. The Fyw, developed by Kirchner 2016, has become a powerful descriptor of streamwater flow path as the substitute for mean transit time. It is important to test the how Fyw change with different timing and sampling time coverage of water isotopes. The results showed "high" temporal variability of Fyw but no seasonality of Fyw based on the criterion defined by the author. The variability due to sampling time chosen is very useful for the isotope hydrology community. This study sheds new light on the development and application of Fyw, which is interesting and suitable for HESS. I find this paper is generally well-written but not strong enough. One of my concern is that how and why the 2% difference was defined as significant for the three hypotheses? This threshold value is introduced in the paper but not clearly explained. The discussion section lacks the discussion of the importance of the results. It would be a stronger paper if the author can explain the cause of the Fyw timevariability, which is ambiguous in the current form. Other specific comments on this paper are listed below.

We thank reviewer #3 for the helpful comments.

We will replace the 2% difference:
Based on other reviewer comments, we now estimated the uncertainty of Fyw by Gauß error propagation (preliminary Figure R1). Fyw of all data (the single sine wave fit) had an uncertainty of ±4%. We will use this new data-driven value instead of the ±2% for re-evaluating our hypotheses:

[Figure]

**Figure R1**. 189 Fyw results (black) and uncertainty (grey) compared to Fyw for all data (red, solid line) and respective uncertainty (red, dashed line). Additionally plotted is the adjusted $R^2$ (blue).

The following can be said from this result:

a) with a drop in $R^2$ below approx. 0.2 the uncertainty increases drastically. This, together with the strongly fluctuating Fyw results (page 11, lines 6-8), indicates that in the Wüstebach an $R^2$ of at least

0.2 should be reached. We highly recommend conducting similar studies in different catchments to test whether different $R^2$ threshold values exist in other catchments.

b) Fyw of all data ("Fyw all" in Figure R1) had an uncertainty of appr. ±4%. We will use this new data-driven value instead of the ±2% for re-evaluating our hypotheses.

The importance of our results will be emphasized in the discussion:

We will first add the new Fyw uncertainties and discuss these. Additionally, an extended analysis of hydro-meteorological data that points to special climatic conditions during summer 2015 will be added. These special conditions might have influenced Fyw and its uncertainty (analysis not yet finalized). The finished analysis may lead to sampling recommendations for other studies.

Previous studies (e.g., Lutz et al., 2018; von Freyberg et al., 2018) showed that Fyw reacts to changes in e.g., precipitation and discharge. Thus, it is safe to assume that other catchments also have a time-varying Fyw. Applying our method would yield information about the Fyw behavior and its uncertainty which is important before applying the method to a catchment and especially when comparing results of different catchments. We will emphasize this in the discussion.

P3-L27: change "8" to Eight

We will do this.

P5-L14: Add "reciprocal of" or similar phrase before "24 hours…" since frequency (f) should be 1/T.

We will rephrase this "i.e., if CP(t) and CS(t) are calculated in hourly time steps then the frequency f is 1/8766; once per 24 x 365.25 hours)."

Figure 1: It would be nice to add latitude and longitude to the map. An alternative way is giving the latitude and longitude of the sampling location in the text. Square brackets with "-" can be removed, it may be misread as minus.

Latitude and longitude were added. We will remove [-] were applicable.

Figure 4a: Which line represent the R2?

The red and orange lines are $R^2$ (orange = $R^2$ of TF and Q, red = mean). We will change the legend entries to "Mean $R^2$", "TF $R^2$" and "Q $R^2$" to clarify.

Figure 7, 8, and 9: The hypotheses should be explained in the captions.

We will add explanations.

References

Lutz, S. R., Krieg, R., Müller, C., Zink, M., Knöller, K., Samaniego, L., and Merz, R.: Spatial patterns of water age: Using young water fractions to improve the characterization of transit times in contrasting catchments. Water Resources Research,54, 4767–4784.https://doi.org/10.1029/2017WR022216, 2018.

von Freyberg, J., Allen, S. T., Seeger, S., Weiler, M., and Kirchner, J. W.: Sensitivity of young water fractions to hydro-climatic forcing and landscape properties across 22 Swiss catchments. Hydrol.Earth Syst. Sci., 22, 3841–3861, https://doi.org/10.5194/hess-22-3841-2018, 2018.

---

## Author Response (AR1)

This paper presents a numerical experiment to estimate the relative influence of the different sampling periods in the estimate of the fraction of young water (fyw). The authors used 1-year long subsets of precipitation and stream tracer data sampled sequentially over a 4.5-year long record. This resulted in 189 different to estimate fyw based on sine function fits. I find the paper interesting as this approach to estimate the event water fraction is becoming popular among hydrologist. However I dough this paper provides information useful outside the catchment where the data was collected. The authors made no case on how these findings would be relevant to other locations. As such, it reads like a case study. Therefore, I suggest this paper not be consider for publication in HESS in its present form. In addition, I found the study lacks proper justification for the used of 2% difference in fyw as indicative of a significant difference.

We thank reviewer #1 for the helpful comments.

Usefulness to other catchments
The manuscript was adapted to account for the following points and supplementary material was added:

The main aim of this study is to present a *generic method* to analyze the time-variance of the fraction of young water. The reviewer already mentioned that Fyw is becoming more popular. Still, we lack information on when to best use this method and how sensitive it is to different datasets (e.g. frequency of sampling and length of observation time). Thus, investigating its use, limits and pitfalls is very important before we apply it to any catchment and particularly when comparing results from different catchments.

Many catchment studies showed that the transit time of water strongly varies (e.g. Harman, 2015; Heidbüchel et al., 2013), and it is thus very likely that Fyw also varies in other locations than ours. While previous studies focused on hydroclimatic and methodological influences on Fyw, this study is the first to focus on the influence of the sampling period and length. This is a first step, and it is highly recommended that this is repeated in other catchments to assess if this is a general situation or only a few catchments have time-varying Fyw (which we doubt because of strongly varying transit times in general).

Ultimately, catchment comparison studies that rely on Fyw should be based on comparable Fyw results. For example, Stockinger et al. 2016 already showed that only changing the sampling frequency of isotopes data led to drastically changed Fyw results. The present study goes further and shows that also the sampling period can influence Fyw. Based on reviewer comments, we now also present the associated uncertainties that are also varying in time.

Information on the various influences on Fyw results is critical for catchment comparison studies. We encourage hydrologists to use our generic method to test the existence of strong time-variances of Fyw in other catchments. However, the application of this method to a large set of catchments is beyond the scope of this study.

Additionally, we found strong evidence of the 2015 European heat wave significantly increasing Fyw uncertainty. With this knowledge we were able to reduce uncertainty. Additionally, we found indications that snow also potentially influenced Fyw uncertainty. Thus, we now present suggestions on how to reduce uncertainty in estimating Fyw.

Using 2% as difference
We now applied Gauß error propagation to estimate uncertainty in Fyw and used it to derive a data-driven threshold. This threshold was based on the uncertainty when estimating Fyw with a single sine wave to the complete data set. The threshold is now 4%. Figure R1 shows the uncertainty bands of the initial 189 Fyw results (influenced by the 2015 European heat wave) and was taken from the supplementary material.

[Figure]

**Figure R1**. Fyw and its uncertainty using all data (black solid and dashed lines) compared with the average of streamflow and precipitation adjusted $R^2$ values of the respective sine wave fits (mean $R^2$).

The following can be said from this result:

a) with a drop in $R^2$ below approx. 0.2 the uncertainty increases drastically. This, together with the strongly fluctuating Fyw results, indicates that in the Wüstebach an $R^2$ of at least 0.2 should be reached. We highly recommend conducting similar studies in different catchments to test whether different $R^2$ threshold values exist in other catchments.
b) Fyw using a single sine wave had an uncertainty of appr. ±4%. This is the new threshold that was used for re-evaluating our hypotheses.
c) the Fyw results become highly uncertain during 2014/2015. This was partly due to the 2015 European heat wave and we managed to correct for its influence.

The statistical approach is also somehow vague. For example, it would be important to know how does the r2 fits of the input compare to the r2 fits of the output. This would allow understanding what is driving to low mean r2 values that were observed for some of the results.
A comparison of input and output $R^2$ is shown in Figure 4: the mean $R^2$, $R^2$ of throughfall (input to the catchment) and $R^2$ of streamflow (output from the catchment). Both the throughfall and streamflow $R^2$ values became low during this period too. In a further analysis we found that the main driving force for low $R^2$ (and thus high Fyw uncertainty) was the magnitude of the throughfall amplitude (Figure 4b). This is featured in the manuscript now.

Other specific comments:
Line 21 (P1): Sentence in poorly worded.
Based on the new uncertainty analysis, we suggest:

"Based on an increase of Fyw uncertainty when the mean adjusted R² was below 0.2 we recommend further investigations into the dependence of Fyw and its uncertainty to goodness-of-fit measures."

Line 23 (P1): The abstract indicates that they recommend an r2 threshold for future studies. However nowhere in the text, the authors offer any justification for the limit.
The added uncertainty of Fyw supports a certain threshold of quality for the fit. Very low goodness-of-fit values increased the uncertainty (as expected). Otherwise, R² values close to 0 would also be accepted and the respective Fyw results accepted as they are.

We changed the text (page 13, line 6):

"[…], we assumed that in our case the Fyw calculation method reached its limit below an average R²adj = 0.2. Fyw became highly sensitive to a small change in input data and in consequence highly uncertain. We recommend further investigations of the sensitivity of Fyw to the goodness-of-fit (not necessarily only measured with R²adj) for future studies. It remains to be seen if a value of 0.2 for R²adj is a general critical threshold for Fyw or if different catchments show varying results."

Line 6 (P2), Line 15 (P13) and elsewhere: Better to refer "water stable isotopes" rather than "stable isotopes of water"
We changed it.

Line 16 (P3): Indicates that the hypotheses were tested against rules of acceptance that were based on whether differences in Fyw exceeded a threshold value of ± 2%. A more comprehensive justification for the 2% threshold should be included.
This was changed to the data-driven 4%.

Line 17-18 (P4): Please explain how this precision was estimated. Did you collect duplicate samples?
This is the long-term precision derived from the uncertainty of 10,000s of measurements of various water samples conducted during the last years. Each unique sample is measured 6 times.

Line 21 (P4): Did you consider using deuterium instead of 18O?
D and 18O are strongly correlated (R² = 0.97 throughfall, 0.87 streamflow) so we did not consider using it. We added this information to the manuscript.

Line14-16 (P5): It would be interesting to see the distributions or R2 of both fits independently.
R² is shown in Figure 4a for both throughfall and streamflow (orange lines labeled TF R² and Q R²).

Line 1-5 (P6): Since the 2% threshold is mentioned in the introduction this explanation belongs there.
We moved the suggested sentences and adapted them to the new threshold value.

Line 22 (P7): These values are very low. An r2 =-.08 would indicate that a sine wave function is weak to describe the variability of the data.
The low R² mentioned are values for the single sine wave fit to the full 4.5-year time series and are 0.09 for TF and 0.23 for Q. We fully agree that those sine wave functions are gross simplification of the inter-annual variability of isotopes; which is the point of this study: a single sine wave fit oversimplifies naturally occurring, annual variations. However, even if a sine wave is weak to describe the data, the Fyw calculation method is based on using the amplitudes of sine wave functions (Kirchner, 2016). We added this discussion in the revised version.

Line 24-25 (P7): Can you provide information about the range of the r2 of these fits
We added:
"The 189 fitted sine waves had a wide range of R²adj values: precipitation ranged from -0.02 to 0.63 with a mean of 0.22 and streamflow ranged from 0.00 to 0.55 with a mean of 0.25."

Line 4-9 (P8): Would this indicate that the sine fit method is not appropriate for much of 2014? How confident can one be of the Fyw estimates when the r2 are below 0.2?

We added the uncertainty estimate and Fyw indeed becomes highly unreliable in this period. Hydrometeorological evidence pointed to a strong influence of the 2015 European heat wave, and part of the uncertainty could be corrected. Using only a single sine wave would not have revealed this. In the discussion we encourage studies of the reliability of Fyw based on goodness-of-fit measures of the sine waves.

Line 12-14 (P8): It is not clear what is the significance of this clustering of points.

We removed the sentence, as this was just an observation on our part.

Line 19 (P8): Considering how skew the data is would it be better to use the median? Also I suggesting some standard deviation or standard error.

We ultimately decided against using the median versus the mean value in hypotheses testing as we assume that a single sine wave rather averages the data (mean result) instead of finding the median. Furthermore, the mean Fyw (0.09) did not deviate much from the median Fyw (0.08) and both lie within the uncertainty bounds of the Fyw derived by the single sine wave approach (0.12 ± 0.04).

Line 20 (P8) Please consider some measure of error or uncertainty in the fyw estimates.

We added Fyw uncertainty.

Line 23-345 (P8): Please elaborate, that is indicate how many of the 189 were between this ranges.

The manuscript now reads:

"Out of the 189 Fyw results 159, i.e. 84%, were within those boundaries (Figure 6a). It could be possible that the period between July 2014 and October 2015 with low $R^2$adj values and erratic Fyw behavior significantly influenced the rejection of the hypothesis. Therefore, in a second step we excluded this period, calculated the mean for those values and evaluated Fyw results again (Figure 6b). The new mean Fyw was 0.07 with 93% of results found between 0.03 to 0.11."

Line 29 (P8): Please provide some statistical information about the strength of the correlation.

We added:

"[…]the runoff coefficient (Q/P) was negatively correlated with $R^2$ = 0.25 and p-value = 1.7E-11 (Figure 7d). Leaving out again the period from July 2014 to October 2015 reduced the correlation to $R^2$ = 0.08 and p-value = 9.8E-4."

Line 10 (P9): This is confusing about figure 9. Are these the 189 fits? That is, are these fits over a one-year duration time series?

We agree that the sentence was confusing. We changed it to:

"As mentioned in the methods, the Fyw results were put in the middle of the one-year calculation period (calculating from February 2016 to February 2017, the result would be displayed as a data point in August 2016). We grouped together all Fyw results that were assigned to a specific calendar month and used a box plot to detect possible seasonality (Figure 8)."

Line17 (P9): Where is the value of d18O for ground water coming from?

The information about groundwater sampling was added to the methods:

"Isotope data was complemented by δ18O values of groundwater sampled in four different locations in weekly intervals since 2009."

Line 19-20 (P9): please elaborate some more in the parallel to the Weigand et al. [2017] study

We extended the sentence:

"The study by Weigand et al. [2017] came to the same conclusion for the Wüstebach catchment using wavelet analysis of nitrate and DOC data collected at mainstream and tributary locations. While lower altitude locations of the catchment near the outlet were dominated by groundwater, higher altitude areas were less affected. This finding was additionally supported by field observations of shallow groundwater."

Line 18-19 P13): How do we know this conclusion is relevant to other catchments?
We expect that changes in flow paths over time alter the Fyw result of a catchment. Because of this, a one-year long time series of tracer data might lead to very different Fyw results depending on when the tracer data was sampled. Therefore, the results of comparison study could vary greatly simply by shifting the one-year sampling window by a couple of weeks.

Please see also our answer to the major comment regarding usefulness of our results for other catchments.

Line 24-25 (P 13): This sentence is vague. Please explain.
We adapted the sentence to make it clearer:

"If feasible, we recommend investigating a multi-year time series of tracer data with the method suggested in this study to enhance our knowledge of the sensitivity of Fyw to the chosen time frame in different catchment situations and the behavior of its uncertainty."

Line 26-28 (P13) It would be important to understand if the variability observed here would be relevant to understand difference across catchments.
As discussed above, different sampling periods yield (occasionally very) different Fyw result that would influence the interpretation of catchment comparison studies. Please also refer to the discussion points above regarding the usefulness to other catchments.

Figure 1: The markers for the precipitation and runoff gauges are too similar. Add latitude and longitude grids to the map. The contour should be in the legend indicating the units. In addition, the font next to the contours is difficult to read.
We added it.

Figure 4: In the legend, please clarify that "mean" refers to
We changed it to "Mean $R^2$", "TF $R^2$" and "Q $R^2$".

Figure 7: Please add a legend.
Done.

Figure 8: The caption should include an explanation of what is hypothesis 2
Explanations were added to all figures.

References
Harman, C. J. (2015), Time-variable transit time distributions and transport: Theory and application to storage-dependent transport of chloride in a watershed, Water Resour. Res., 51, doi:10.1002/2014WR015707.

Heidbüchel, I., P. A. Troch, and S. W. Lyon (2013), Separating physical and meteorological controls of variable transit times in zero-order catchments, Water Resour. Res., 49, 7644–7657, doi:10.1002/2012WR013149.

Kirchner, J. W.: Aggregation in environmental systems - Part 1: Seasonal tracer cycles quantify young water fractions, but not mean transit times, in spatially heterogeneous catchments, Hydrol Earth Syst Sc, 20(1), 279-297, 2016.

Lutz, S. R., Krieg, R., Müller, C., Zink, M., Knöller, K., Samaniego, L., and Merz, R.: Spatial patterns of water age: Using young water fractions to improve the characterization of transit times in contrasting catchments. Water Resources Research,54, 4767–4784.https://doi.org/10.1029/2017WR022216, 2018.

Stockinger, M.P., H.R. Bogena, A. Lücke, B. Diekkrüger, T. Cornelissen and H. Vereecken (2016): Tracer sampling frequency influences estimates of young water fraction and streamwater transit time distribution. J. Hydrol. 541: 952-964, doi:10.1016/j.jhydrol.2016.08.007.

Weigand, S., R. Bol, B. Reichert, A. Graf, I. Wiekenkamp, M. Stockinger, A. Lücke, W. Tappe, H. Bogena, T. Pütz, W. Amelung and H. Vereecken (2017): Spatiotemporal dependency of dissolved organic carbon to nitrate in stream- and groundwater of a humid forested catchment – a wavelet transform coherence analysis. Vadose Zone J. 16(3), doi:10.2136/vzj2016.09.0077.

This manuscript investigates the temporal variability of the young water fractions (Fyw) based on a 4.5-year time series of δ18O in precipitation and streamwater in the German Weiherbach catchment. For this, the authors fit sine curves to the entire 4.5-year data set to estimate the long-term average Fyw. Then, they cut out 189 individual 1-year 18O time series from the 4.5-year data set (i.e., shifting each 1-year period by 1 week) and fit individual sine curves to these 1-year periods to estimate 189 Fyw-values. The goodness-of-fit of the sine curves to the δ18O data was quantified through adjusted R2 values. Three hypotheses were tested: "(1) Fyw estimates do not change over time (time-invariance) (2) Short-term changes in the start of a tracer sampling campaign do not influence the Fyw estimate (sampling-invariance) (3) Fyw estimates are similar for a given calendar month of different years (seasonal-invariance)" (P3L13-15). By applying a Fyw-threshold value of 2%, the authors reject all hypotheses and conclude that Fyw-values based on 1-year isotope data sets can be highly variable over time. This time-variably of Fyw hampers catchment-comparison studies that utilize tracer data of different time series lengths or time periods. I find the critical evaluation of new metrics (young water fraction Fyw) interesting and useful as this allows to better plan sampling campaigns or to use existing data sets more efficiently for a robust estimation of Fyw. In that regard, I consider the testing of hypothesis 2 the most useful scientific contribution of this study (Sect. 3.4) because it quantifies how much a 4-weeks delay can change the Fyw-values that are estimated from 1-year data sets. Unfortunately, these changes in Fyw do not correlate with any of the tested hydro-meteorological variables (Sect. 4.4) so that the authors cannot provide any suggestions about the optimal sampling strategy.

We thank reviewer #2 for evaluating our study and the helpful comments.

We now conducted an extended analysis of hydro-meteorological data and found that the 2015 European heat wave significantly influenced Fyw estimates and uncertainty, as well as indications of snow influencing it. From this, we derived the recommendation to sample only one year's winter in a one-year long sampling campaign.

Besides the testing of hypothesis 2, I find it difficult to identify a clear motivation and the novel scientific contribution of this study. The fact that Fyw responds to changes in precipitation, discharge and/or catchment wetness has already been established (e.g., Kirchner, 2016b; Lutz et al., 2018; von Freyberg et al., 2018; Wilusz et al., 2017), and thus it can be expected that Fyw changes over time in a catchment with a variable hydro-climatic regime such as the Weiherbach. Thus, the temporal changes in Fyw that have been identified in the present study are likely related to the hydro-climatic conditions at the site, however, the scientific analysis of these relationships often remains too superficial (which is surprising, given that the Weiherbach catchment is an intensively studied research site). As such, this study does not teach us something new about the catchment but rather shows that different tracer time series provide different young water fractions. Although it is interesting to quantify these temporal differences in Fyw, it remains to be tested how these findings for the 0.385km2 Weiherbach catchment are transferable to other landscapes and climates.

First, to avoid any misunderstandings we would like to make clear that our study area is the Wüstebach catchment and not the Weiherbach catchment as indicated by the reviewer. The primary focus of this study was not to identify catchment influences on Fyw but to investigate the extent and significance of temporal Fyw changes using variable 1-year time series of isotope tracer data. We primarily aim to improve the robustness of the Fyw method and not to analyze specific Wüstebach influences on Fyw. This is essential information for planning future sampling strategies in other catchments and for the layout of catchment comparison studies.

As mentioned above, we found evidence of the 2015 European heat wave negatively influencing Fyw uncertainty and accounted for it. Furthermore, a sampling recommendation could be derived.

Even before that, a recommendation was given in the original submission: estimating Fyw with data from a single year is not enough (page 12, 25f); the time-variant Fyw for a catchment should be calculated to understand the behavior and uncertainty for a given location (page 13, lines 2ff).

The reviewer already mentioned that previous studies found Fyw reacts to changes in precipitation, discharge, and other factors. Thus it is safe to assume that other catchments also have a time-varying Fyw and applying our method would yield more information about the Fyw behavior and uncertainty in each catchment. We highly suggest conducting the same study for other catchments in other landscape or climatic units to be able to generalize the findings.

These discussion points were added to the manuscript and the new supplementary material.

Major comments:
We first answer major comment #6 as it is of great importance:

6. At the very end of the Discussion section the authors state that a previous analysis has been carried out that used a 3-year isotope times series from the Weiherbach catchment. This previous study already showed that the Fyw values differed substantially between three 1-year periods (Stockinger et al., 2017, data in the supplement). In the present study, the authors simply repeat this analysis with a 4.5-year isotope data set from the same site knowing that their hypothesis "(1) Fyw does not deviate more than _2% from the mean of all Fyw results indicating long-term invariance [: : :]" will likely be rejected. I was surprised to read about a very similar previous study at the very end of the current manuscript and wondered why is it necessary to repeat the analysis when the result (rejection of hypothesis 1) is already known?

The focus of Stockinger et al., 2017 was to correct canopy-induced isotope changes in throughfall for the complete time series. 3 individual years were cut out as a test without going into further detail. We extended this test into its own study to focus on investigating the extent and dynamics of the time-variance of Fyw, which was not the focus of Stockinger et al., 2017 and would have been out of scope for the previous study.

We had no possibility to know the results beforehand since for hypothesis 1 more than 90% of Fyw results must be within ±4%. It is not possible to estimate from 3 out of 189 results if 19 results (10%) would be outside the ±4%.

1. One of my largest concern is that the presented analysis did not provide any information on the uncertainties of the individual Fyw estimates. Only the adjusted $R^2$ values of the sine fits are presented. Previous studies that calculated young water fractions for several other catchment reported uncertainties in Fyw between 1% and 41% (e.g., Jasechko et al., 2016; Stockinger et al., 2016; von Freyberg et al., 2018). Thus, it should be tested whether the individual young water fractions that were calculated from the 1-year time series are indeed statistically significantly different from each other when their uncertainties are considered. Looking at the low adjusted $R^2$ values for the July 2014- October 2015 period (e.g. Figure 4), I would expect the uncertainties of the 1-year Fyw values to be rather large. However, instead of analyzing the uncertainties in Fyw, the authors mainly focus on the time-variability of the individual 1-year Fywvalues and conclude rather boldly (P12L22) "The obtained Fyw could be a potential outlier, a larger value or part of the Fyw baseline". I would argue that the uncertainty in Fyw (e.g., expressed as standard error) would allow us to objectively judge whether we can believe our Fyw estimates or not. Such an analysis is, however, missing here. In fact, knowing the uncertainties of the individual 1-year Fyw values would allow a more informative analysis of how the Fyw-uncertainty (not Fyw itself) is controlled by hydroclimatic conditions. Such an analysis might provide concrete guidelines for planning targeted sampling campaigns to robustly estimate Fyw.

We thank the reviewer for pointing out this very important issue. It is the aim of this study to present a generic method to analyze Fyw for time-variance and thus improve the robustness of the Fyw method. We added Fyw uncertainty estimates using Gauß error propagation (Figure R1, taken from the supplementary material):

[Figure]

**Figure R1**. Fyw and its uncertainty using all data (black solid and dashed lines) compared with the average of streamflow and precipitation adjusted $R^2$ values of the respective sine wave fits (mean $R^2$).

The following can be said from this result:

a) with a drop in $R^2$ below approx. 0.2 the uncertainty increases drastically. This, together with the strongly fluctuating Fyw results, indicates that in the Wüstebach an $R^2$ of at least 0.2 should be reached. We highly recommend conducting similar studies in different catchments to test whether different $R^2$ threshold values exist in other catchments.
b) Fyw using a single sine wave had an uncertainty of appr. ±4%. We used this new data-driven value for evaluating our hypotheses.
c) the Fyw results become highly uncertain during 2014/2015. This was partly due to the 2015 European heat wave and we managed to correct for its influence.

Furthermore, we found indications of snow influence on Fyw uncertainty and now recommend sampling only one year's winter.

2. Furthermore, given that the uncertainties in Fyw values can potentially be much larger than 2% (as it was shown in the previous studies cited above), to me the 2% threshold seems too low and the authors' justification for that 2% threshold is not convincing.

We now used data-driven 4% based on the uncertainty estimation of using all data.

3. The hydro-meteorological conditions in the Weiherbach catchment were highly variable during the 4.5-year study period. For instance, only the winter 2013/2014 was snow-free in contrast to the other winters when a snowpack built up (Sect. 3.1). In addition, 21% "…of the forest were clear-cut in August/September 2013: : :" (P3L27), which significantly altered the streamflow regime of the Weiherbach creek (Wiekenkamp et al., 2016). In Fig. 8d (P24) we find that the runoff coefficients for the Weiherbach catchment ranged between roughly 0.8 and 1.25, suggesting that hydro-climatic conditions at the site varied considerably over time. The authors do not, however, provide any data or figures that present the hydro-climatic conditions during the study period except for the scatter plots in Figure 8, which contrasts 1-year averages of four hydroclimatic metrics with the respective Fyw-values. Despite the highly variable streamflow regime of the catchment and the authors citing another study where flow weighting of the streamwater isotope values resulted in "…significant changes in Fyw…" (P2L4), the authors should more thoroughly investigate how catchment wetness might control Fyw. Why was streamflow-weighting not done here? Why was there no further analysis of potential factors that may control the large variability in 1-year Fyw values, particularly in the period July 2014-October 2015? It seems likely, that individual storm events may have had strong effects on the discharge of young water, so it may be useful to investigate extreme events rather than average behavior.

Precipitation and streamflow weighing were both done for Fyw calculation, but unfortunately not clearly mentioned in the submission. We clarified this now
"Precipitation isotope values were weighed using collected precipitation volumes, while streamflow was weighed using runoff volumes."

As mentioned above, we found influences of the 2015 European heat wave and snow. This is now extensively discussed in the new manuscript version and in the supplementary material.

4. Sect. 3.5 and Figure 9: It is not clear to me how the Fyw values for testing hypothesis 3 (seasonal invariance) were determined. As far as I understood, Fyw-values were calculated for 189 1-year periods (Sect. 2.3). How were month-specific Fyw-values extracted from these annual Fyw-values? Wouldn't each 1-year Fyw-value be affected by the isotope values of all 12 months that comprise this 1-year period? If so, I doubt that the analysis presented in Sect. 3.5 and Figure 9 provides useful information.

No monthly Fyw were extracted from the data. Each of the 189 Fyw results was assigned to the date that lies in the middle of the calculation period. For example, Fyw was calculated from 1 January to 31 December and the corresponding result was assigned to 1 July.

We then grouped all Fyw results according to the month they were assigned to. All results in the boxplot (now Figure 8) are still 1-year calculation results. Should a seasonal trend be observable, one could argue that e.g., a 1-year sampling campaign centered around July would lead to higher/lower Fyw estimates compared to when it is centered on March. Ultimately, we found indications of snow potentially increasing Fyw uncertainty if the winters of two different calendar years are featured in a one-year sampling campaign. This is now discussed in the revised version of the manuscript.

5. Part of the analysis presented in "4.2 Fraction of young water" is not valid. First, the authors calculated Fyw from the entire 4.5-year data set (Fyw,4.5=10.8%) and compared this to the average of the 186 1-year Fyw values (9.3%), concluding that both values are similar with regard to their 2% threshold. A second comparison was carried out with Fyw,4.5 and the average of a much smaller number of 1-year Fyw values that neglects the Fyw values from the period July 2014-October 2015 (7.5%). This second comparison should, however, use another Fyw value as a reference based on the same isotope data set (i.e., 4.5 years minus the period July 2014-October 2015)- otherwise the authors compare apples with oranges.

We thank the reviewer for pointing this out. We accounted for this now.
"If we use the averages of the 189 sine wave amplitudes to calculate Fyw, the result would be 0.08 instead of 0.12 of the single sine wave. This is less than the 0.04 difference in Fyw defined by this study as the data-driven threshold value for significant differences. Leaving out the period of low R²adj values the single sine wave and the average of 189 amplitudes would both yield approximately 0.07."

Minor comments:
P3L8-9: "However, it remains to be tested how sensitive the Fyw method is towards the timing and the length of the available data." Why does this need to be tested? Can you provide an example of where the length and the timing of the isotope data resulted in different Fyw values? Otherwise, a clear motivation for your analysis is missing.

We rephrased the introduction
"The mentioned studies highlight the current research interest in the new measure of Fyw. For this reason, it is necessary to investigate the sensitivity of Fyw and its uncertainty to different datasets. This is especially important for catchment comparison studies where the conceptualization of calculating Fyw might vary between catchments or datasets of different catchments may vary in quality. The question to answer is how much of the difference between individual Fyw estimates stems from actual, catchment-borne differences in flow path distributions and which part is merely based on e.g., different data quality or quantity."

P4L25: "Because of this on average 43 isotope values were available for precipitation compared to 53 values for streamflow." Does this average refer to a 1-year period? Please clarify. It would also be nice to provide the total number of streamwater and precipitation samples of the entire 4.5-year period.

Yes, this refers to the one-year calculation periods. The total number of P and Q samples (156 and 195, respectively) is now mentioned on page 5, line 8.

P5L22-23: I would suggest to move these two sentences to the beginning of the chapter to make clear where the number "189" comes from.

We agree.

P5L13: 24*365.25 is 8766 not 1/8766

We changed it
"(i.e., if CP(t) and CS(t) are calculated in hourly time steps then the frequency f is 1/8766; once per 24 x 365.25 hours)"

P5L31-32: What do you mean with "the timing of peaks and the individual amplitudes"? Do you refer to the isotope time series or to the fitted sine functions?

We referred to the fitted sine functions and changed it:

"Apart from the 189 Fyw results we also calculated Fyw for the whole time series with one sine wave as was the standard of previous studies. We compared its peak timing and amplitude to the timing of peaks and amplitudes of the 189 sine waves."

P6L3: Here you switch units of Fyw (0.02 and 2%). Also, in the text you express Fyw in percent, whereas in the figures you use the scale from zero to one. Please be consistent throughout the manuscript.

We now consistently use e.g., 0.02 instead of 2%.

P7L13-14: Please be more specific about what water isotopes you are talking about, e.g. add $\delta$18O.

The sentence is now:

"Precipitation isotope ratios ranged from -3.04 to -17.80‰, spanning a range of 14.76‰ in $\delta$18O values."

P8L28-30: Please provide some metrics for the strength of these correlations (e.g., Pearson correlation coefficients).

We added statistical information to the text:

""[…]the runoff coefficient (Q/P) was negatively correlated with $R^2$ = 0.25 and p-value = 1.7E-11 (Figure 7d). Leaving out again the period from July 2014 to October 2015 reduced the correlation to $R^2$ = 0.08 and p-value = 9.8E-4."

P8L29: Was the runoff coefficient calculated with catchment-average precipitation or throughfall? I would suggest to add the runoff coefficients to Fig. 3 since the relationship between Q/P and the sine wave fits to the isotope data are discussed in Sect. 4.1.

The runoff coefficient was calculated with throughfall, but we changed to open precipitation to enable comparability to other studies. It is now prominently featured in the supplementary material.

P9L16-20: You suddenly present groundwater isotope data without providing information about the source (location, sampling procedure, number of samples etc.) of these data. Please include this information into Sect. 2.2.

This was on oversight on our part. We added:

"Isotope data was complemented by $\delta$18O values of groundwater sampled in four different locations in weekly intervals since 2009."

P10L13: "The double-peak in precipitation of autumn 2015 was not found in streamflow (Figure 3)." Do you refer to the $\delta$18O in precipitation and streamflow or to the sine fits to the isotope data?

To the sine fits, we adapted the sentence to also account for the 2015 European heat wave:

"However, the relationship between precipitation and streamflow considerably changed due to the influence of the 2015 European heat wave: while the sine wave double-peak of precipitation in summer 2015 was not transferred to streamflow (Figure 3), the amplitudes of both lost their close relationship at the same time (supplementary Figure S2a)."

P11L33: "Thus, during the 4.5-years Fyw never fell below the baseline of 5% […]" This statement is incorrect. Figures 6 and 7 clearly show that Fyw fell below 5% on several occasions, such as around June 2014 and September 2016.

We changed it
"Thus, during the 4.5-years Fyw seldom fell below the baseline of 0.05 and […]"

P12L5: "The variability in Fyw of this study could not be explained by most meteorological or hydrometric variables". Could a lack of correlation be explained by the large distance (3km) of the meteorological station to the study site? What about median values of the hydro-climatic variables or metrics that describe extreme events?

Correlations of precipitation amounts ($R^2$ = 0.95), temperature ($R^2$ = 0.99) and relative humidity ($R^2$ = 0.94) of the 3 km distant climate station with the respective climate data from the clear-cut area of the Wüstebach catchment showed good $R^2$. We did not use the on-site climate station for our study as its data does not cover the full study period.

While most hydrometeorological data still did not have a strong correlation with the time-variable Fyw, we found evidence of the 2015 European heat wave increasing Fyw uncertainty.

P12L9: "…the different sampling periods of all mentioned studies…". This contradicts a previous statement: "…Lutz et al. [2018] used the same sampling period for precipitation and streamflow for all 24 investigated catchments." (P11L25).

We changed it
"Such contradictions could be explained by the different sampling periods of our study and the mentioned studies but also by differing catchment characteristics.".

P12L23: "As the violation of hypothesis 2 did not correlate with any meteorological or hydrometric data : : :". How can a violation correlate with anything? Please clarify.

We referred here to the timing of the violation of hypothesis 2: if the timing could be connected to hydrometric or meteorological data.

We adapted:

"As the timing of the violation of hypothesis 2 did not[…]"

Figures: The date formats in all figures are confusing. Does 4/10/13 mean 4th October 2013 or 10 April 2013? Also, I would suggest to have each tick mark at the first of the month and to have consistent date axes in all figures.
4/10/13 refers to April 10[th], 2013. We will adapt the figures to uniformly start on 4/1/13 with the exception of Figure 2 (just a theoretical example) and Figure 3 (showing the input data and starting on a different date than the Fyw result figures; see also explanation below).

Figure 4: This figure misses a proper legend (e.g., What does "Mean" stand for?). The unit and numbers of Fyw on the right vertical axes don't match. Do panels a and b share the same legend? Why are the shown time series much shorter than 4.5 years?

We will change the legend entries to "Mean $R^2$", "TF $R^2$" and "Q $R^2$".
We changed part b of the figure drastically to avoid any confusion of units.

The time series are shorter than 4.5 years since each Fyw result was placed in the middle of the year it was calculated for. The time series starts on 10/10/12, thus the first Fyw result is placed on 4/10/13. Doing this cuts off the first half year and the last half year of the complete time series, explaining the shortening.

Stockinger et al. presents a study to evaluate the temporal variability of young water fraction (Fyw) based on 189 sine curve fits of 1-year subsets of a 4.5-year rainwater and streamwater 18-O isotope dataset. The Fyw, developed by Kirchner 2016, has become a powerful descriptor of streamwater flow path as the substitute for mean transit time. It is important to test the how Fyw change with different timing and sampling time coverage of water isotopes. The results showed "high" temporal variability of Fyw but no seasonality of Fyw based on the criterion defined by the author. The variability due to sampling time chosen is very useful for the isotope hydrology community. This study sheds new light on the development and application of Fyw, which is interesting and suitable for HESS. I find this paper is generally well-written but not strong enough. One of my concern is that how and why the 2% difference was defined as significant for the three hypotheses? This threshold value is introduced in the paper but not clearly explained. The discussion section lacks the discussion of the importance of the results. It would be a stronger paper if the author can explain the cause of the Fyw timevariability, which is ambiguous in the current form. Other specific comments on this paper are listed below.

We thank reviewer #3 for the helpful comments.

We now estimated Fyw uncertainty by Gauß error propagation (Figure R1 taken from the supplementary material). Fyw of the single sine wave fit had an uncertainty of ±4% (not shown in Figure R1). We used this new data-driven value instead of the ±2% for evaluating our hypotheses.

[Figure]

**Figure R1**. Fyw and its uncertainty using all data (black solid and dashed lines) compared with the average of streamflow and precipitation adjusted R² values of the respective sine wave fits (mean R²).

The following can be said from this result:

a) with a drop in R² below approx. 0.2 the uncertainty increases drastically. This, together with the strongly fluctuating Fyw results, indicates that in the Wüstebach an R² of at least 0.2 should be reached.

We highly recommend conducting similar studies in different catchments to test whether different $R^2$ threshold values exist in other catchments.

b) Fyw using a single sine wave had an uncertainty of appr. ±4%. We used this new data-driven value for evaluating our hypotheses.

c) the Fyw results become highly uncertain during 2014/2015. This was partly due to the 2015 European heat wave and we managed to correct for its influence.

Furthermore, we found indications of snow influence on Fyw uncertainty and now recommend sampling only one year's winter.

The importance of our results is now emphasized in the discussion:

We will first add the new Fyw uncertainties and discuss these. Additionally, an extended analysis of hydro-meteorological data was added in the supplementary material. The discussion was expanded by the 2015 European heat wave and possible influence of snow.

Previous studies (e.g., Lutz et al., 2018; von Freyberg et al., 2018) showed that Fyw reacts to changes in e.g., precipitation and discharge. Thus, it is safe to assume that catchments other than the Wüstebach also have a time-varying Fyw. Applying our method would yield information about the Fyw behavior and its uncertainty which is important before applying the method to a catchment and especially when comparing results of different catchments. We emphasize this now more in the manuscript.

P3-L27: change "8" to Eight

Done.

P5-L14: Add "reciprocal of" or similar phrase before "24 hours…" since frequency (f) should be 1/T.

We rephrased this "(i.e., if CP(t) and CS(t) are calculated in hourly time steps then the frequency f is 1/8766; once per 24 x 365.25 hours)"

Figure 1: It would be nice to add latitude and longitude to the map. An alternative way is giving the latitude and longitude of the sampling location in the text. Square brackets with "-" can be removed, it may be misread as minus.

Latitude and longitude were added. [-] were removed in all Figures.

Figure 4a: Which line represent the R2?

The red and orange lines are $R^2$ (orange = $R^2$ of TF and Q, red = mean). We changed the legend entries to "Mean $R^2$", "TF $R^2$" and "Q $R^2$" to clarify.

Figure 7, 8, and 9: The hypotheses should be explained in the captions.

We added explanations.

References

[revised manuscript text omitted]

---

## Referee Report (RR1)

**2nd review of hess-2018-604**
**"Time-variability of the fraction of young water in a small headwater catchment"**
**by Michael P. Stockinger et al.**

The authors responded to all of my comments in detail and addressed my major concerns, which were
- Provide uncertainties of the individual 1-year Fyw estimates
- Justify the choice of the threshold value
- Account to hydro-climatic variations in the data set
- Explain how month-specific Fyw values were extracted from 1-year Fyw values
- Be consistent in the inter-comparison of Fyw values
- Better relate the current study to a previous one where 1-year Fyw have already been calculated and compared

Unfortunately, the track-changed version of the manuscript did not show deletions, and thus it was quite difficult to reconstruct all changes made by the authors during this round of revisions. My comments below refer to the version of the manuscript that shows track-changes.

The authors added a more detailed analysis of the 2015 summer heat wave, which has contributed largely to increased uncertainties in 1-year Fyw estimates around the period. As a result, the authors decided for a part of their analysis to remove 4 moths of heat wave-affected isotope data from the 4.5-year time series. For this new 4.1-year isotope time series, the authors obtained generally more consistent Fyw values. In addition, the authors discussed the potential effects of winter precipitation isotope values in the 1-year data set, e.g., sampling parts of two winter seasons instead of only one likely result in different Fyw estimates.

Despite the additional analyses of the 2015 summer heat wave and winter precipitation effects on Fyw, most of my major concerns with this study remain:

The authors introduced a new threshold value of 0.04 to test their three hypotheses. Although this threshold value is two times larger than the previous one, the authors still reject all three hypotheses; thus, the overall outcome of the manuscript did not change. However, because the revised Figure 4a now shows the uncertainties (i.e., standard errors) of the individual 1-year Fyw values, I doubt that hypothesis 1 can be rejected so readily. The authors decided to not show the Fyw-uncertainties in Figure 6, which they use to illustrate that >10% of the individual 1-year Fyw values fall outside the 0.04-threshold. If the uncertainties of Fyw would be considered here, I suppose that >90% of all Fyw values would lie within the boundary conditions, i.e. hypothesis 1) can be accepted.

Similarly, hypothesis 2) needs to be re-evaluated considering the uncertainties of the individual 1-year Fyw values. In P15L27-28, the authors refer again to Figure 6 to point out that some Fyw-values that are 4 weeks apart differ by more than 0.04. I doubt that these differences are statistically significant given the large standard errors of the 1-year Fyw values around summer 2015 and at the end of the 4.5-year time series.

The analysis around hypothesis "3) Fyw estimates are similar for calculation years that are centered around a given calendar month (seasonal-invariance)" is still not satisfying to me since I am still puzzled about the question of what is actually tested here. For instance, in P8L5-9 the authors state: "If the hypothesis is accepted, it would indicate seasonal changes in the Fyw result as a function of the start date of a one-year sampling campaign." I do not follow this train of thought. If all 1-year Fyw values values centered around e.g. August would be similar within 0.04, how can we conclude that 1-year Fyw values varies seasonally? Please be more specific about the goal of this analysis and why it is important.

As far as I understood, the main objective of the study was to show that individual 1-year Fyw values might not be representative for the long-term Fyw (P16L2-3). As I have pointed out in my first review, this finding seems rather trivial since a catchment's hydro-climatic conditions and flow pathways can change substantially between seasons and years so that the age of streamwater is likely to change as well. A specific 1-year Fyw value might therefore estimate the average fraction of young water for this particular year, whereas a multi-year Fyw-value will be representative for the average fraction of young water for these multiple years. In case of hydro-climatic conditions and isotope values being highly variable during these multiple years, the uncertainty in the multi-year Fyw value would accordingly be large. As a consequence, comparing 1-year Fyw values between catchments is not per se a useless analysis as long as the same time periods are used and uncertainties are considered.

Overall, I find that the reviewed version of the manuscript is more challenging to read due to confusing wording and too general or contradicting statements. Below I provide some examples, however, my list is not complete and I suggest a thorough review of the language in a revised version of the manuscript.

In order to improve readability, I would suggest to be more specific about and more consistent in addressing the different Fyw values. In the manuscript, the distinction between the different Fyw values is often not clear and causes confusion, e.g. what is a "single Fyw" in P13L6-7: "If the isotope data and Fyw results of the period of low $R^2_{adj}$ values was left out, the average Fyw of the 189 sine waves compared even better to the single Fyw (approximately 0.07 in both cases)." It might be easier to refer to the 4.5-year Fyw (often referred to as "single Fyw" in the manuscript) as $F_{yw,4.5}$, to the individual 1-year Fyw values as $F_{yw,i}$ (with $i$ denoting the $i$-th 1-year time series), and to the average of all 189 $F_{yw,i}$ values as $\overline{F_{yw,186}}$.

The phrase "data-inherent uncertainty of the complete timeseries" is repeatedly used (e.g., P7L13: "In doing so, the time-variable Fyw results were tested against the data-inherent uncertainty of the complete timeseries"). The phrase "data-inherent uncertainty of the complete timeseries" is inaccurate as it is not clear what time series are referred to (temperature, streamflow, isotopes?) and what "data-inherent" means; I would suggest to be more specific and simply say "the standard error of $F_{yw,4.5}$".

P8L15-16: "Therefore, these results would represent a runoff with a fraction of young water that systematically varies with the start of the sampling campaign, from a catchment with stable environmental conditions and water transport properties, and low sampling uncertainties." Can you elaborate on this? It would be good to add a statement about whether this condition would be good/bad for estimating Fyw or whether these conditions would result in small/large uncertainties in Fyw.

P8L10-14: This paragraph is very confusing due to poor wording. E.g., "Despite it having a time-variant young water fraction, all three hypotheses are accepted." This statement is contradicting hypothesis 1) "Fyw estimates do not change over time (time-invariance)"! Please re-phrase. Also, the statement "On a long term basis, the young water fraction does not deviate significantly from its overall mean value (time-invariance)" is not easy to understand. Please use more specific wording to make clear what Fyw-values you are referring to.

What do you mean by "short-term changes in the start of a one-year sampling campaign" (P10L30)? How can a "start" exhibit short-term changes? What does "short-term" mean here? I suggest that you actually refer to the shift of the starting and end time of the time series by 1 week. Also, the following sentence (P10L30-31) "The hypothesis is accepted if during any consecutive four weeks Fyw did not differ more than 0.04", reads as if you have calculated Fyw based on 4-week isotope data sets. Please be more specific.

Your justification of why streamflow at Wüstebach is mainly comprised of groundwater is not convincing (P11L21-27). You infer from the similar $\delta^{18}O$ values in precipitation (-8.53‰) and groundwater (-8.43±0.17‰), that groundwater is mainly fed by rainwater (which is a somewhat trivial observation since precipitation is usually the main source of recharge); however, streamwater has a mean $\delta^{18}O$ signature (-8.40±??‰) that is very similar to that of precipitation, too. Instead, aren't the Fyw values (both, $F_{yw,4.5}$ and $F_{yw,i}$) actually more informative here as they suggest that roughly >90% of streamwater is older than 3 months?

Please be more specific when you talk about "peaks" and "amplitudes". E.g., (P12L12-15): "However, the relationship between precipitation and streamflow considerably changed due to the influence of the 2015 European heat wave: while the double-peak of precipitation in summer 2015 was not transferred to streamflow (Figure 3), the amplitudes of both lost their close relationship at the same time (supplementary Figure S2a)." The "double-peak of precipitation" is actually the "double-peak of the sine fits to the precipitation isotopes"; the "amplitudes of both" are the "seasonal cycle amplitudes of the isotopes in streamwater and precipitation".

The authors claim in P13L1-2 that "… this hydrological information about the Wüstebach catchment [that precipitation mixes with a quasi-constant $\delta^{18}O$ source] would have been impossible to detect with a single sine wave fit." I feel that the authors over-sell their results here. The sine fits to the entire 4.5-year streamwater and precipitation isotope data set would lead to the same conclusion (that is, the sine fit to streamwater isotopes resembles very similar patterns to the sine fit to precipitation isotopes).

Similarly, the authors conclude that the baseline Fyw value of 0.05 could only be found because of calculating Fyw for individual 1-year periods (P15L8-11). However, their estimate of Fyw based on the entire 4.5-year isotope record was 0.12±0.04 (or 0.11±0.04, see my other comment), so that a minimum Fyw value of 0.8 can be obtained. Thus, I disagree with the authors' statement that "Using a single sine wave would not have revealed this lower boundary" (P15L10), because it would still have revealed a very similar lower boundary (0.8).

Specific comments (referring to the version of the manuscript that shows track-changes):

P1L19: "For a given calendar month …" sounds like as if you have calculated Fyw for one month only. Similarly inaccurate expressions are used throughout the manuscript and should be corrected everywhere.

P1L23: Define "adjusted $R^2$", since the reader won't know where this value comes from without reading the rest of the manuscript.

P9L17: From your numbers of As and Ap, I obtain a 4.5-year Fyw value of 0.11... Can you please provide the standard errors of As and Ap?

P10L19: On P9, you stated that the average of all 189 1-year Fyw values was 0.08, not 0.09!

P10L6-7: You could actually calculate the effects of Ap on the standard error of Fyw through the Gauss error propagation approach. Through this, you might find that Ap has a much larger influence than As simply because Ap>As and standard error(Ap)> standard error(As).

P12L17-20: "Thus, considering the general hydrological observations obtained from the isotope data discussed above, we conclude that a certain percentage of precipitation became groundwater while another

percentage that might or might not be Fyw quickly generated runoff, conserving the precipitation d18O signal in streamflow and resulting in the similar shapes of the 189 sine wave pairs." This sentence is very general ("… a certain percentage of precipitation…") and does not tell us anything specific or interesting. Please rephrase.

P1416-30: What about catchments in Mediterranean climates that receive highly seasonal precipitation inputs? This case has already been discussed in Kirchner (2016), e.g. Figure 3.

Kirchner, J. W.: Aggregation in environmental systems-Part 2: Catchment mean transit times and young water fractions under hydrologic nonstationarity, Hydrol. Earth Syst. Sci., 20, 299-328, 10.5194/hess-20-299-2016, 2016.

P16L15-16: The statement "Furthermore, snow blankets also change the isotopic signal potentially to a degree that obscures seasonal isotope patterns [Cooper, 2006]" is seems arbitrary. Why snow blankets and not snow cover? What do you mean by "obscures"? What seasonal isotope pattern is obscured? How much can the seasonal snow cover change the streamwater (?) isotope signal, and would this be significant in case of the Wüstebach catchment?

P16L27-26: "Only two Fyw were calculated in contrast to the 189 results of the present study (approximately 1%), making insights into the possible causes and a judgement if varying Fyw results are an isolated result or the rule impossible." Poor language, please rephrase.

P17L7-8: "(1) a potential strong influence of the 2015 European heat wave on Fyw estimates and uncertainties was discovered, which is a problem which could magnify in the future considering global warming;" Why is this a problem? It could very much be true that the 1-year Fyw values between June 2014 and October 2015 are representative for this particular period.

P17L9-10: "(2) precipitation and groundwater seemed to be the only end-members in streamflow which is information that isotope hydrograph separation studies can greatly benefit from;" As far as I can tell, these were the only two endmembers measured. So how can you be sure that there are not more endmembers, such as soil moisture or deep groundwater? What about isotopic fractionation effects due to evaporation?

P17L11-12: "Testing three hypotheses about the time-variability of Fyw we found that both in the long and short term Fyw is time-variable …" I do not understand how the long-term variability of Fyw was tested. Please clarify.

Figure 2, 4 and 5: Unit for Fyw is missing.
Figure 6: Please include the uncertainty bounds for the Fyw values, similar to Figure 4.

Figure 7: Tick marks are missing, unit for Fyw is missing.

---

## Author Response (AR2)

**Comments by the editor**

Dear authors,

We have now received the comments of the two reviewers on your revised manuscript. While one of the reviewers thinks that the revised manuscript has been improved and all comments on the previous version satisfactory addressed. However, the other reviewer has still many concerns on the methodology and results that need to be appropriately addressed before considering publication. I think that the report shows that the reviewer has put significant effort to provide extremely valuable feedback that will give you the opportunity to further improve your work and its value for the hydrology community. I hope you find these comments useful.

Dear editor,

The reviewers feedback indeed helped us again. We took care in improving the manuscript accordingly and believe that it is now stronger than before. Please find below our replies to the reviewer comments and the changes we implemented in the manuscript, which can also be found in the track-change version. We are looking forward to a positive answer and publication.

Thank you,

Dr. Michael Stockinger, on behalf of all authors

**Comments by anonymous referee #3**

In this revision the authors made adequate improvements of the paper. This paper can be accepted with the following change:

P2L13: The accurate definition of $F_{yw}$ is the streamflow fraction that is younger than a specific threshold ($\tau_{yw}$), which is not always exactly three months. This threshold may vary from catchment to catchment. I suggest add something like 'approximately' or 'roughly' before 'three months'.

We thank the reviewer for evaluating our manuscript again and changed the manuscript accordingly.

**Comments by anonymous referee #2**

The authors responded to all of my comments in detail and addressed my major concerns, which were

- Provide uncertainties of the individual 1-year $F_{yw}$ estimates

- Justify the choice of the threshold value

- Account to hydro-climatic variations in the data set

- Explain how month-specific $F_{yw}$ values were extracted from 1-year $F_{yw}$ values

- Be consistent in the inter-comparison of $F_{yw}$ values

- Better relate the current study to a previous one where 1-year $F_{yw}$ have already been calculated

and compared

Unfortunately, the track-changed version of the manuscript did not show deletions, and thus it was quite difficult to reconstruct all changes made by the authors during this round of revisions. My comments below refer to the version of the manuscript that shows track-changes.

We appreciate the time and effort the reviewer put into reviewing our manuscript again. We opted for hiding deletions in the track-changed version, as this improved the readability immensely. It was not our intention to hide changes and we apologize for the difficulty the reviewer experienced.

The authors added a more detailed analysis of the 2015 summer heat wave, which has contributed largely to increased uncertainties in 1-year $F_{yw}$ estimates around the period. As a result, the authors decided for a part of their analysis to remove 4 moths of heat wave-affected isotope data from the 4.5-year time series. For this new 4.1-year isotope time series, the authors obtained generally more consistent $F_{yw}$ values. In addition, the authors discussed the potential effects of winter precipitation

isotope values in the 1-year data set, e.g., sampling parts of two winter seasons instead of only one likely result in different Fyw estimates.

Despite the additional analyses of the 2015 summer heat wave and winter precipitation effects on Fyw, most of my major concerns with this study remain:

5  The authors introduced a new threshold value of 0.04 to test their three hypotheses. Although this

threshold value is two times larger than the previous one, the authors still reject all three hypotheses; thus, the overall outcome of the manuscript did not change. However, because the revised Figure 4a now shows the uncertainties (i.e., standard errors) of the individual 1-year Fyw values, I doubt that hypothesis 1 can be rejected so readily. The authors decided to not show the Fyw-uncertainties in Figure 6, which they use to illustrate that >10% of the individual 1-year Fyw values fall outside the 0.04-
10  threshold. If the uncertainties of Fyw would be considered here, I suppose that >90% of all Fyw values would lie within the boundary conditions, i.e. hypothesis 1) can be accepted.

These are important considerations. We agree that hypothesis 1 can't neither be "absolutely" rejected nor accepted due to the size of the uncertainty bands. While we agree on this point, we also want to point out that the suggestion of the reviewer in the last sentence (>90% of Fyw values would lie within our boundaries) can also not be absolutely stated.

15  In a thought-experiment, let us assume for Figure 4a that the uncertainty bands are almost non-existent: a case of very well constrained Fyw results. We would still need some rule to judge the significance of differences between Fyw results. This is currently a subjective choice, as there are no agreed upon rules yet. Below in the comments, just before the specific comments, the reviewer mentions that 5% Fyw is similar to 8% Fyw. Stating similarity here was also a subjective choice. We simply want to highlight the problem of deciding on differences/similarities in Fyw and not highlight "mistakes" of the reviewer because
20  in fact, no one currently knows if this is significant or not.

Due to these difficulties we chose from the start of this study to define rules and state specific hypotheses that are solely based on the calculated Fyw. We now avoid using the term "significant" as it is strongly connected to the concept of "statistical significance". Our rules can and should be discussed in hydrological sciences as the definition of differences/similarities in Fyw is an important issue to be solved. We do not claim to have found the final definition of different Fyw results but wanted
25  to show one possible way and foster research into this issue. Choosing different rules for the acceptance/rejection of the null hypotheses can of course lead to outcomes different to the ones of our study. We provide all information on uncertainties and thresholds and the reader can judge himself/herself on how similar or different the results are.

It was our fault for writing this not clearly in the manuscript and we adapted the text accordingly.

Similarly, hypothesis 2) needs to be re-evaluated considering the uncertainties of the individual 1-year Fyw values. In P15L27-
30  28, the authors refer again to Figure 6 to point out that some Fyw-values that are 4 weeks apart differ by more than 0.04. I doubt that these differences are statistically significant given the large standard errors of the 1-year Fyw values around summer 2015 and at the end of the 4.5-year time series.

We adapted the manuscript to phrase this more carefully (P8 L12f)

"This study does not claim to have found the final rules for judging differences in Fyw but presents one possible way of doing
35  this by using the threshold value of 0.04."

The analysis around hypothesis "3) Fyw estimates are similar for calculation years that are centered around a given calendar month (seasonal-invariance)" is still not satisfying to me since I am still puzzled about the question of what is actually tested here. For instance, in P8L5-9 the authors state: "If the hypothesis is accepted, it would indicate seasonal changes in the Fyw result as a function of the start date of a one-year sampling campaign." I do not follow this train of thought. If all 1-year Fyw
40  values values centered around e.g. August would be similar within 0.04, how can we conclude that 1-year Fyw values varies seasonally? Please be more specific about the goal of this analysis and why it is important.

We changed the manuscript and explained the intention of hypothesis 3 and the results more clearly. We made the following revision (P8 L1-4):

"This hypothesis tests if the Fyw(189) results centered around a specific month do not differ more than ±0.04 within this month. With this we test (1) if the starting month of a one-year sampling campaign can influence Fyw(189) variability and (2) if a "seasonal pattern" can be detected with e.g., larger Fyw(189) results during one-year periods centered around specific months."

5  As far as I understood, the main objective of the study was to show that individual 1-year Fyw values might not be representative for the long-term Fyw (P16L2-3). As I have pointed out in my first review, this finding seems rather trivial since a catchment's hydro-climatic conditions and flow pathways can change substantially between seasons and years so that the age of streamwater is likely to change as well. A specific 1-year Fyw value might therefore estimate the average fraction of young water for this particular year, whereas a multi-year Fyw-value will be representative for the average fraction of young
10  water for these multiple years. In case of hydro-climatic conditions and isotope values being highly variable during these multiple years, the uncertainty in the multi-year Fyw value would accordingly be large. As a consequence, comparing 1-year Fyw values between catchments is not per se a useless analysis as long as the same time periods are used and uncertainties are considered.

The main objective of the study was to investigate the variability of one-year Fyw results in comparison to the multi-year Fyw
15  with no preference to showing a representativeness or the lack thereof. We already stated in the manuscript that we aim at testing the sensitivity of one-year Fyw results (P3 L19-21):

"The question to answer is how much of the difference between individual Fyw estimates stems from actual, catchment-borne differences in flow path distributions and which part is merely based on e.g., different data quality or quantity."

Thus, we fully agree with the reviewer and had these thoughts already in the manuscript (P13 L7-9):

20  "The fact that the average of Fyw(189) was within the ±0.04 boundary to Fyw calculated with a single sine wave (0.09 vs. 0.12) indicated that the single sine wave generally averaged the behavior of the 189 ones."

(P15 L5-9):

"As far as we can tell, the recent Fyw catchment comparison study of Lutz et al. [2018] used the same sampling period for precipitation and streamflow for all 24 investigated catchments. In contrast, the studies of Jasechko et al. [2016] and von
25  Freyberg et al. [2018] had isotope sampling periods varying in start date and overall length for the 254 and 22 investigated catchments, respectively, potentially influencing the uncertainty for the inter-catchment comparison according to the results of our study."

From the conclusions (P17 L17f):

"The long-term variability has implications for catchment comparison studies when different time periods are investigated."

30  Overall, I find that the reviewed version of the manuscript is more challenging to read due to confusing wording and too general or contradicting statements. Below I provide some examples, however, my list is not complete and I suggest a thorough review of the language in a revised version of the manuscript.

We thank the reviewer for the suggestions and incorporated them.

In order to improve readability, I would suggest to be more specific about and more consistent in addressing the different Fyw
35  values. In the manuscript, the distinction between the different Fyw values is often not clear and causes confusion, e.g. what is a "single Fyw" in P13L6-7: "If the isotope data and Fyw results of the period of low R2 adj values was left out, the average Fyw of the 189 sine waves compared even better to the single Fyw (approximately 0.07 in both cases)." It might be easier to refer to the 4.5-year Fyw (often referred to as "single Fyw" in the manuscript) as $F_{yw,4.5}$, to the individual 1-year Fyw values as $F_{yw,i}$ (with i denoting the i-th 1-year time series), and to the average of all 189 $F_{yw,i}$ values as $F_{yw,186}$.

40  We revised several terms to improve clarity.

The phrase "data-inherent uncertainty of the complete timeseries" is repeatedly used (e.g., P7L13: "In doing so, the time-variable Fyw results were tested against the data-inherent uncertainty of the complete timeseries"). The phrase "data-inherent

uncertainty of the complete timeseries" is inaccurate as it is not clear what time series are referred to (temperature, streamflow, isotopes?) and what "data-inherent" means; I would suggest to be more specific and simply say "the standard error of $F_{yw,4.5}$".

We removed the term "data-inherent".

P8L15-16: "Therefore, these results would represent a runoff with a fraction of young water that systematically varies with the start of the sampling campaign, from a catchment with stable environmental conditions and water transport properties, and low sampling uncertainties." Can you elaborate on this? It would be good to add a statement about whether this condition would be good/bad for estimating $F_{yw}$ or whether these conditions would result in small/large uncertainties in $F_{yw}$.

We added:

"Under these conditions, starting a one-year sampling campaign in different seasons will lead to different $F_{yw}$ results and one needs to take this into consideration when comparing results from different time periods. However, deciding to wait up to four weeks with the start of the campaign will have no impact on $F_{yw}$, while in the long-term the $F_{yw}$ can be considered stable."

However, it is difficult for us to estimate *a priori* the uncertainty of individual $F_{yw}$ results just by using the data in the theoretical results of Figure 2. As we suggest in the study, heat waves and snow could influence uncertainty, but we think that mentioning this here already would be out-of-nowhere and confuse the reader.

P8L10-14: This paragraph is very confusing due to poor wording. E.g., "Despite it having a time-variant young water fraction, all three hypotheses are accepted." This statement is contradicting hypothesis 1) "$F_{yw}$ estimates do not change over time (time-invariance)"! Please re-phrase. Also, the statement "On a long term basis, the young water fraction does not deviate significantly from its overall mean value (timeinvariance)" is not easy to understand. Please use more specific wording to make clear what $F_{yw}$-values you are referring to.

We changed this paragraph to:

"All three hypotheses are accepted in this case: the $F_{yw}$ results are (1) time-invariant as all are within the average $F_{yw} \pm$ its uncertainty (0.04 in this example); (2) sampling-invariant as within any four weeks the maximum difference of $F_{yw}$ results is less than 0.04; and (3) seasonally varying as they show a stable seasonal behavior."

What do you mean by "short-term changes in the start of a one-year sampling campaign" (P10L30)? How can a "start" exhibit short-term changes? What does "short-term" mean here? I suggest that you actually refer to the shift of the starting and end time of the time series by 1 week. Also, the following sentence (P10L30-31) "The hypothesis is accepted if during any consecutive four weeks $F_{yw}$ did not differ more than 0.04", reads as if you have calculated $F_{yw}$ based on 4-week isotope data sets. Please be more specific.

We rephrased the two sentences:

"Here we tested if deciding to delay the start of a one-year sampling campaign up to four weeks could influence $F_{yw}$".

"The hypothesis is accepted if any four consecutive $F_{yw}$ results did not differ more than 0.04".

Your justification of why streamflow at Wüstebach is mainly comprised of groundwater is not convincing (P11L21-27). You infer from the similar $\delta^{18}O$ values in precipitation (-8.53‰) and groundwater (- 8.43±0.17‰), that groundwater is mainly fed by rainwater (which is a somewhat trivial observation since precipitation is usually the main source of recharge); however, streamwater has a mean $\delta^{18}O$ signature (- 8.40±??‰) that is very similar to that of precipitation, too. Instead, aren't the $F_{yw}$ values (both, $F_{yw,4.5}$ and $F_{yw,i}$) actually more informative here as they suggest that roughly >90% of streamwater is older than 3 months?

We agree that the $F_{yw}$ values indicate groundwater dominance while theoretically, the similar precipitation and streamflow values could mean that direct precipitation runoff is the main source of runoff. However, the strong attenuation of the seasonal isotopic signal in the stream, as well as the study of Weigand et al. cited in the text, indicate that groundwater is the main source. We added the strong isotope attenuation in streamflow and the $F_{yw}$ information to this paragraph.

Please be more specific when you talk about "peaks" and "amplitudes". E.g., (P12L12-15): "However, the relationship between precipitation and streamflow considerably changed due to the influence of the 2015 European heat wave: while the doublepeak of precipitation in summer 2015 was not transferred to streamflow (Figure 3), the amplitudes of both lost their close relationship at the same time (supplementary Figure S2a)." The "double-peak of precipitation" is actually the "double-peak of the sine fits to the precipitation isotopes"; the "amplitudes of both" are the "seasonal cycle amplitudes of the isotopes in streamwater and precipitation".

We now wrote it more carefully throughout the manuscript.

The authors claim in P13L1-2 that "… this hydrological information about the Wüstebach catchment [that precipitation mixes with a quasi-constant δ18O source] would have been impossible to detect with a single sine wave fit." I feel that the authors over-sell their results here. The sine fits to the entire 4.5-year streamwater and precipitation isotope data set would lead to the same conclusion (that is, the sine fit to streamwater isotopes resembles very similar patterns to the sine fit to precipitation isotopes).

We rephrased this sentence:

"The 189 sine wave fits to precipitation and streamflow isotope data facilitated finding this hydrological information about the Wüstebach catchment."

Similarly, the authors conclude that the baseline Fyw value of 0.05 could only be found because of calculating Fyw for individual 1-year periods (P15L8-11). However, their estimate of Fyw based on the entire 4.5-year isotope record was 0.12±0.04 (or 0.11±0.04, see my other comment), so that a minimum Fyw value of 0.8 can be obtained. Thus, I disagree with the authors' statement that "Using a single sine wave would not have revealed this lower boundary" (P15L10), because it would still have revealed a very similar lower boundary (0.8).

Our analysis using one-year Fyw results improved the knowledge about the minimum possible Fyw in the Wüstebach catchment (5%). This can also be seen in the histogram of Figure 5 where the class "4 to 6%" features the majority of Fyw results. We would also like to emphasize that this lower boundary of 5% Fyw is only valid for the Wüstebach catchment of our study and we do not claim to have found a general lower boundary.

Specific comments (referring to the version of the manuscript that shows track-changes):

P1L19: "For a given calendar month …" sounds like as if you have calculated Fyw for one month only. Similarly inaccurate expressions are used throughout the manuscript and should be corrected everywhere.

We adapted it

"(3) the Fyw results of one-year sampling campaigns started in a given calendar month[…]"

P1L23: Define "adjusted R2", since the reader won't know where this value comes from without reading the rest of the manuscript.

The adjusted $R^2$ is a term from statistics and accounts for the number of predictors in the model. We mentioned it in the main text, but not the abstract.

P9L17: From your numbers of As and Ap, I obtain a 4.5-year Fyw value of 0.11... Can you please provide the standard errors of As and Ap?

This is due to the mathematical rounding to two decimal places. We used the original values (Ap = 0.71741±0.17956, As = 0.08261±0.01085, Fyw = 0.11515).

P10L19: On P9, you stated that the average of all 189 1-year Fyw values was 0.08, not 0.09!

These are two different calculations. The result of 0.08 came from averaging all 189 amplitudes and then using this averaged amplitude to calculate Fyw. The result 0.09 stems from averaging all Fyw values. We agree that this was confusing and we only used the average of all individual Fyw in the revised version.

P10L6-7: You could actually calculate the effects of Ap on the standard error of Fyw through the Gauss error propagation approach. Through this, you might find that Ap has a much larger influence than As simply because Ap>As and standard error(Ap)> standard error(As).

This is a good suggestion for follow-up studies. We mentioned the possibility of error propagation approaches in the discussion section of the manuscript.

P12L17-20: "Thus, considering the general hydrological observations obtained from the isotope data discussed above, we conclude that a certain percentage of precipitation became groundwater while another percentage that might or might not be Fyw quickly generated runoff, conserving the precipitation d18O signal in streamflow and resulting in the similar shapes of the 189 sine wave pairs." This sentence is very general ("… a certain percentage of precipitation…") and does not tell us anything specific or interesting. Please rephrase.

We deleted this sentence.

P1416-30: What about catchments in Mediterranean climates that receive highly seasonal precipitation inputs? This case has already been discussed in Kirchner (2016), e.g. Figure 3.

Kirchner, J. W.: Aggregation in environmental systems-Part 2: Catchment mean transit times and young water fractions under hydrologic nonstationarity, Hydrol. Earth Syst. Sci., 20, 299-328, 10.5194/hess-20-299-2016, 2016.

The Mediterranean climate of the Smith River catchment in Kirchner (2016) is described as having wet winters and dry summers. We believe that this situation is different to what we describe here. We did not describe a lack of precipitation during higher temperatures, but a longer time frame of precipitation falling on the catchment but never reaching the outlet. Leaving out the precipitation isotope data during the 2015 European heat wave improved results as we eliminated an input signal that had no connection to runoff. This is different to no input signal during summer. If, however, in a catchment in Mediterranean climate large portions of precipitation that fell do not reach runoff for longer times, we can imagine issues with Fyw uncertainty too. Please note that interception losses where accounted for in our study, as we directly used throughfall isotopes and amounts in calculating Fyw.

P16L15-16: The statement "Furthermore, snow blankets also change the isotopic signal potentially to a degree that obscures seasonal isotope patterns [Cooper, 2006]" is seems arbitrary. Why snow blankets and not snow cover? What do you mean by "obscures"? What seasonal isotope pattern is obscured? How much can the seasonal snow cover change the streamwater (?) isotope signal, and would this be significant in case of the Wüstebach catchment?

We are no English native speakers and were not aware of any differences in snow blanket or snow cover. With "obscure" we meant "not conserve the seasonal isotope signal of precipitation in the snow blanket/cover". The studies cited in Cooper, 2006 were done in glaciated areas and might not be 100% transferable to the Wüstebach with its snow accumulation/snow melt cycle. As the sentence is of minor importance to this section, we removed it and the following sentence.

P16L27-26: "Only two Fyw were calculated in contrast to the 189 results of the present study (approximately 1%), making insights into the possible causes and a judgement if varying Fyw results are an isolated result or the rule impossible." Poor language, please rephrase.

"Only two Fyw were calculated in contrast to the 189 results of the present study. This low number of results made it impossible to investigate possible causes of varying Fyw results and to judge if those results were the rule or an exception."

P17L7-8: "(1) a potential strong influence of the 2015 European heat wave on Fyw estimates and uncertainties was discovered, which is a problem which could magnify in the future considering global warming;" Why is this a problem? It could very much be true that the 1-year Fyw values between June 2014 and October 2015 are representative for this particular period.

It could be true yes, but that would neglect the uncertainty. The uncertainty bands were large with the influence of the heat wave, making it impossible to say what the real Fyw value is. Removing the heat wave influence decreased uncertainty.

P17L9-10: "(2) precipitation and groundwater seemed to be the only end-members in streamflow which is information that isotope hydrograph separation studies can greatly benefit from;" As far as I can tell, these were the only two endmembers measured. So how can you be sure that there are not more endmembers, such as soil moisture or deep groundwater? What about isotopic fractionation effects due to evaporation?

We now more clearly write that additional endmembers could be possible in the Wüstebach as we cannot with 100% certainty exclude additional ones, but that the data does not point to missing endmembers.

"Thus, the 189 sine waves strongly indicated that streamflow in the Wüstebach consisted of precipitation and groundwater with no additional, unaccounted sources of runoff such as subsurface flows from outside the catchment boundaries, although additional sources are still theoretically possible."

P17L11-12: "Testing three hypotheses about the time-variability of $F_{yw}$ we found that both in the long and short term $F_{yw}$ is time-variable …" I do not understand how the long-term variability of $F_{yw}$ was tested. Please clarify.

From the abstract:

"(1) At least 90% of $F_{yw}$ results do not deviate more than ±0.04 from the mean of all $F_{yw}$ results indicating long-term invariance."

Hypothesis 1 tests long-term variability of $F_{yw}$ which is mentioned several times throughout the manuscript.

Figure 2, 4 and 5: Unit for $F_{yw}$ is missing.

$F_{yw}$ is a ratio and therefore without unit.

This is also mentioned in the manuscript (P7 L1-7)

"For example, a $F_{yw}$ result of 0.2 on 6th August 2013 means that between 5th February 2013 to 4th February 2014 on average 20% of runoff consisted of water younger than three months"

Figure 6: Please include the uncertainty bounds for the $F_{yw}$ values, similar to Figure 4.

We included the uncertainty in Figure 6.

Figure 7: Tick marks are missing, unit for $F_{yw}$ is missing.

See above for $F_{yw}$ unit, tick marks were added.

**Time-variability and uncertainty of the fraction of young water in a small headwater catchment**

Michael Paul Stockinger[1,2], Heye Reemt Bogena[1], Andreas Lücke[1], Christine Stumpp[2], Harry Vereecken[1]

[1]Forschungszentrum Jülich GmbH, Agrosphere Institute (IBG-3), Wilhelm-Johnen-Straße, 52425 Jülich, Germany
[2]University of Natural Resources and Life Sciences Vienna, Institute of Hydraulics and Rural Water Management, Muthgasse 18, 1190 Vienna, Austria.

*Correspondence to*: Michael Paul Stockinger (michael_stockinger@boku.ac.at)

**Abstract.** The time precipitation needs to travel through a catchment to its outlet is an important descriptor of a catchment's susceptibility to pollutant contamination, nutrient loss and hydrological functioning. The fast component of total water flow can be estimated by the fraction of young water (Fyw) which is the percentage of streamflow younger than three months. Fyw is calculated by comparing the amplitudes of sine waves fitted to seasonal precipitation and streamflow tracer signals. This is usually done for the complete tracer time series available neglecting annual differences in the amplitudes of longer time series. Considering inter-annual amplitude differences, we employed a moving time window of one-year length in weekly time steps over a 4.5-years $\delta^{18}O$ tracer time series to calculate 189 Fyw estimates and their uncertainty. They were then tested against the following null hypotheses: (1) At least 90% of Fyw results do not deviate more than ±0.04 (4%) from the mean of all Fyw results indicating long-term invariance. Larger deviations would indicate changes in the relative contribution of different flow paths; (2) for any four-week window Fyw does not change more than ±0.04 indicating short-term invariance. Larger deviations would indicate a high sensitivity of Fyw to a 1-4 weeks shift in the start of a one-year sampling campaign; (3) the Fyw results of one-year sampling campaigns started in a given calendar month do not change more than ±0.04 indicating seasonal invariance. 
[revised manuscript text omitted]

This study will use "Fyw(all)" to refer to the Fyw calculated by using one sine wave each for the complete 4.5-year time series of precipitation and streamflow isotope data and "Fyw(189)" for the 189 individual Fyw results calculated using a one-year calculation window which was moved in 7-days steps. A minimum time window length of one year was chosen to fully capture the annual isotope signal. Fyw is calculated by fitting sine waves to both the seasonally-varying precipitation and streamflow isotope signals, respectively. We used the multiple regression algorithm IRLS (iteratively reweighted least squares, available in the software R) to minimize the influence of outliers:

$$C_P(t) = a_P \cos(2\pi ft) + b_P \sin(2\pi ft) + k_P,$$
$$C_S(t) = a_S \cos(2\pi ft) + b_S \sin(2\pi ft) + k_S \qquad (1)$$

with $C_P(t)$ and $C_S(t)$ the simulated precipitation and streamflow isotope values of time $t$, $a$ and $b$ regression coefficients, and $k$ and $f$ the vertical shift and frequency of the sine wave. The difference of $C_P(t)$ and $C_S(t)$ to the measured isotope time series in precipitation and streamflow is minimized to fit the parameters a, b and k, while the frequency $f$ of the sine wave is known due to its annual character (i.e., if $C_P(t)$ and $C_S(t)$ are calculated in hourly time steps then the frequency $f$ is 1/8766; once per 24 x 365.25 hours). Precipitation isotope values were weighed using collected precipitation volumes, while streamflow was weighed using runoff volumes. The goodness-of-fit of the sine waves are expressed as the adjusted coefficient of determination $R^2$ ($R^2_{adj}$) which accounts for the number of predictors in the regression model. If not otherwise stated we will use the mean of the streamflow and precipitation $R^2_{adj}$, as both sine waves are needed to estimate the fraction of young water. After fitting the multiple regression equations, the amplitudes $A_P$ and $A_S$ and Fyw can be calculated:

[revised manuscript text omitted]

This hypothesis tests if the Fyw(189) results centered around a specific month do not differ more than ±0.04 within this month. With this we test (1) if the starting month of a one-year sampling campaign can influence Fyw(189) variability and (2) if a "seasonal pattern" can be detected with e.g., larger Fyw(189) results during one-year periods centered around specific months. To clarify, we did not calculate Fyw on a monthly basis but simply sorted the Fyw(189) results by the month they were assigned to (midpoint of the calculation year, see also explanation above). If the hypothesis is accepted

it would indicate seasonal changes in the Fyw(189) as a function of the start date of a one-year sampling campaign. This would allow the pre-planning of sampling campaigns to establish comparable Fyw results. However, it is also possible that the hypothesis is accepted if Fyw(189) is constant for all 189 results, as only the intra-month variance matters with this hypothesis. Contrary to the acceptance of the hypothesis, rejecting it for most months would indicate that there are no distinct seasonal patterns imprinted on Fyw(189).

This study does not claim to have found the final rules for judging differences in Fyw but presents one possible way of doing this by using the threshold value of 0.04. An example of a theoretical Fyw time series is given in Figure 2. All three hypotheses are accepted in this case: the Fyw results are (1) time-invariant as all are within the average Fyw ± its uncertainty (0.04 in this example); (2) sampling-invariant as within any four weeks the maximum difference of Fyw results is less than 0.04; and (3) seasonally varying as they show a stable seasonal behavior.. Therefore, these results would represent a runoff with a fraction of young water that systematically varies with the start of the sampling campaign, from a catchment with stable environmental conditions and water transport properties, and low sampling uncertainties. Under these conditions, starting a one-year sampling campaign in different seasons will lead to different Fyw results and one needs to take this into consideration when comparing results from different time periods. However, deciding to wait up to four weeks with the start of the campaign will have no impact on Fyw, while in the long-term the Fyw can be considered stable.

[revised manuscript text omitted]

the results of an, e.g., risk assessment, then these Fyw changes are negligible for the practical purpose at hand. The present study did not aim to answer any specific question related to Fyw that would justify setting a threshold value a priori but investigated the time-variability of Fyw and used the uncertainty as its threshold value. Thus, the results of the hypothesis tests might change completely if we would answer practical questions about the Wüstebach such as the vulnerability to pollutant

5    loads of a certain chemical substance. Choosing different rules for the acceptance or rejection of our hypotheses has a large impact on the results. The hypotheses and rules of acceptance should be fitted to the task at hand and we urge further studies to investigate appropriate rules for the practical usage of Fyw as we do not claim to have found the absolute answer in deciding which Fyw results are different and which are not.

[revised manuscript text omitted]